# MMR-V: *What's Left Unsaid?* A BENCHMARK FOR MULTIMODAL DEEP REASONING IN VIDEOS

**Kejian Zhu**[1,2], **Zhuoran Jin**[1,2],[†] **Hongbang Yuan**[1,2], **Jiachun Li**[1,2], **Shangqing Tu**[3]
**Pengfei Cao**[1,2], **Yubo Chen**[1,2], **Kang Liu**[1,2], **Jun Zhao**[1,2],[†]
[1]The Key Laboratory of Cognition and Decision Intelligence for Complex Systems,
Institute of Automation, Chinese Academy of Sciences, Beijing, China
[2]School of Artificial Intelligence, University of Chinese Academy of Sciences [3]Tsinghua University
zhukejian2025@ia.ac.cn {zhuoran.jin, hongbang.yuan} @nlpr.ia.ac.cn
{pengfei.cao, yubo.chen, kliu, jzhao} @nlpr.ia.ac.cn

## ABSTRACT

The sequential structure of videos poses a challenge to the ability of multimodal large language models (MLLMs) to locate multi-frame evidence and conduct multimodal reasoning. However, existing video benchmarks mainly focus on understanding tasks, which only require models to match frames mentioned in the question (hereafter referred to as "question frame") and perceive a few adjacent frames. To address this gap, we propose **MMR-V: A Benchmark for Multimodal Deep Reasoning in Videos**. The benchmark is characterized by the following features. **(1) Long-range, multi-frame reasoning**: Models are required to infer and analyze evidence frames that may be far from the question frame. **(2) Beyond perception**: Questions cannot be answered through direct perception alone but require reasoning over hidden information. **(3) Reliability**: All tasks are manually annotated, referencing extensive real-world user understanding to align with common perceptions. **(4) Confusability**: Carefully designed distractor annotation strategies to reduce model shortcuts. MMR-V consists of 317 videos and 1,257 tasks. Our experiments reveal that current models still struggle with multi-modal reasoning; even the best-performing model, Gemini-2.5-pro, achieves only 64.3% accuracy. Additionally, current reasoning enhancement strategies (Chain-of-Thought and scaling test-time compute) bring limited gains. Error analysis indicates that the CoT demanded for multi-modal reasoning differs from it in textual reasoning, which partly explains the limited performance gains. We hope that MMR-V can inspire further research into enhancing multi-modal reasoning capabilities.

🤗 **Benchmark**    https://huggingface.co/datasets/JokerJan/MMR-VBench
⭕ **Project**    https://mmr-v.github.io/

## 1 INTRODUCTION

Recent models like OpenAI's o1 (Jaech et al., 2024) and Deepseek-R1 (Guo et al., 2025) have significantly improved text reasoning ability through reinforcement learning. This has sparked growing interest in multimodal reasoning (Wang et al., 2025). Models like o3 (OpenAI, 2025b) and GPT-5 (OpenAI, 2025a) have achieved impressive results on image reasoning tasks through tool use, integrating visual information into the reasoning process to enable deep reflection and evidence mining. However, most of these studies focus on images, with limited exploration of more challenging video reasoning tasks. Video involves sequential and richer multimodal information, requiring models to reason and mine evidence over long-range, multi-frame. Since this capability is essential for real-world applications such as embodied intelligence and intelligent security monitoring (Hou et al., 2008; Yang et al., 2024b), it naturally raises an important question: *can current MLLMs perform deep multimodal reasoning and "think with videos" like o3 on image tasks?*

---

[†]Corresponding authors.

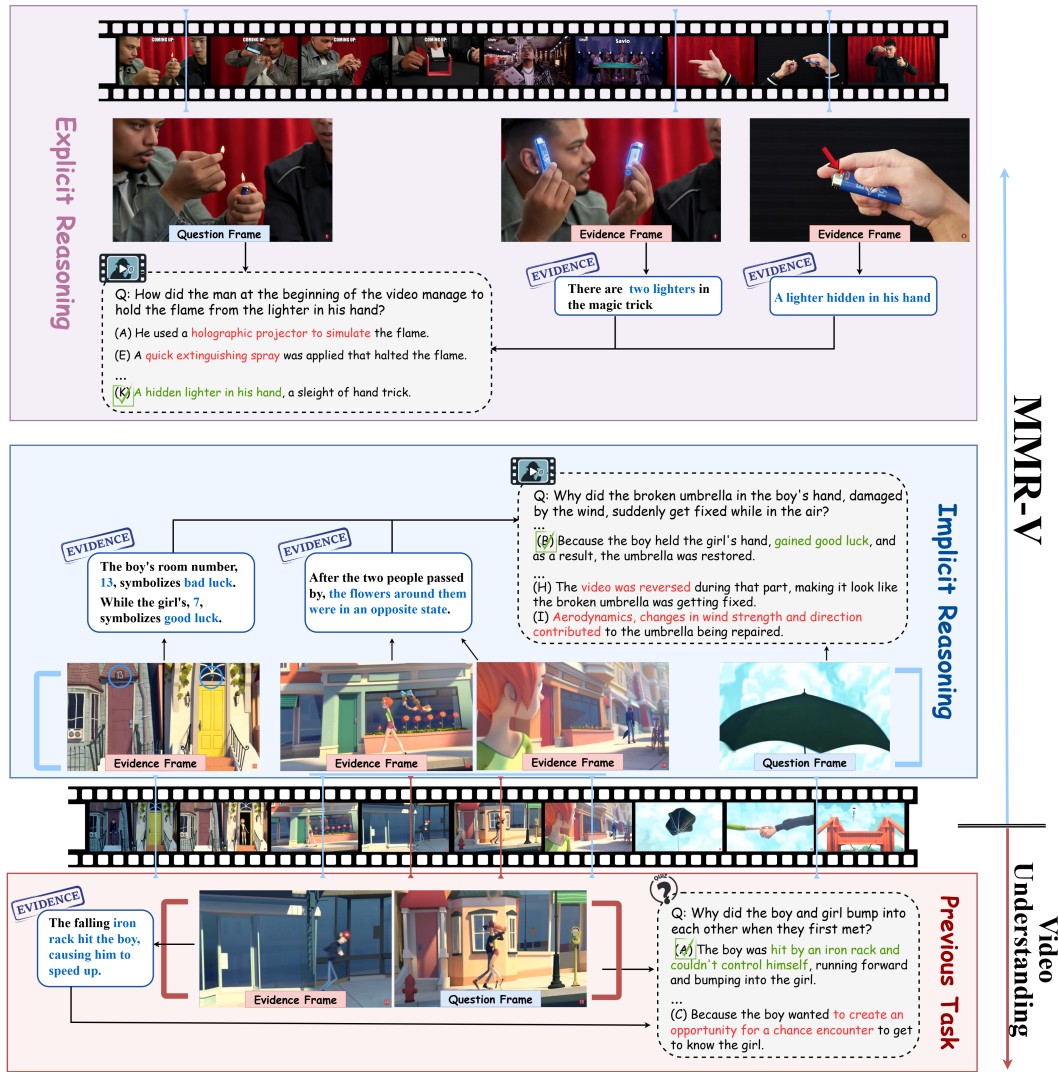

Figure 1: Examples showing the MMR-V tasks and the difference from previous video benchmarks.

However, existing video benchmarks primarily focus on perception and understanding tasks (Zhou et al., 2024; Fu et al., 2024). These tasks often only require locating frames mentioned in the question and understanding adjacent frames. For example, at the bottom of Figure 1, noticing the boy being hit by the metal frame is enough to understand why he ran into the girl. Such tasks fall short in evaluating multimodal reasoning abilities. We summarize their limitations as follows: **(1) Limited frame context**: Even for long videos, existing tasks often rely on just a few adjacent frames, failing to exploit the long-range sequential structure of the video. **(2) Lack of reasoning**: Many questions can be answered through direct perception. **(3) Unrealistic task**: Simple perception and adjacent-frame understanding tasks do not meet the real-world demands for AI system strong capabilities.

To address these shortcomings, we propose **MMR-V Bench: A Benchmark for Multi-modal Deep Reasoning in Videos**. We present two examples to illustrate the key differences with previous video understanding benchmarks in Figure 1. MMR-V offers the following features: **(1) Long-range, multi-frame reasoning:** tasks involve multimodal reasoning over non-adjacent video frames to locate and analyze multiple evidences; **(2) Beyond perception:** questions cannot be answered by direct perception of question frame directly, requiring reasoning and the extraction of implications; **(3) Reliability:** All tasks are annotated manually, and potential subjective bias is reduced by cross-referencing the most popular video comments. **(4) Confusability:** We employ carefully designed annotation strategies to craft model-aligned distractor options, thereby ensuring confusability.

Inspired by cognitive and psychological theories (Evans, 1984; Sun, 2006; Polanyi, 2012), such as Kahneman's Dual Process Theory (Kahneman, 2011), we categorize the tasks in MMR-V into **implicit reasoning** and **explicit reasoning**. The key distinction lies in *whether the question requires*

*reasoning beyond surface-level information to infer underlying implications*. Explicit reasoning is defined as questions that can be solved using perceivable information from the video. For example, the task shown in Figure 1 requires noticing the two lighters hidden in the hand. Implicit reasoning requires extracting and interpreting the underlying subtext behind visual information. For example, in the implicit reasoning case shown in Figure 1, it requires inferring the underlying implication that the girl's room number 7 symbolizes good luck. This is more of an assessment of *EQ*, testing whether the model can use its deep understanding of the world knowledge to make implicit and subconscious reasoning paths like humans.

**MMR-V comprises 317 videos and 1257 tasks**. The videos span six major categories, with lengths ranging from 7 to 3771 seconds, with an average of 277 seconds. Tasks are further divided into 10 categories and subcategories. Each task is in multiple-choice format with approximately ten options on average. Tasks typically require reasoning over average 12 video frames, covering about 60% of video duration. All questions and correct answers are human-annotated and reviewed. Distractors are generated using a carefully designed annotation strategy (Details in Section 3.2).

We evaluated 11 proprietary models and 10 open-source models on MMR-V. The results reveal that even the best-performing model, Gemini-2.5-pro, achieved **only 64.3% accuracy**, highlighting the significant challenge MMR-V poses to current multimodal large language models. Our key findings are as follows. **(1) Multimodal reasoning challenge:** Our findings in Section 4.2 show that reasoning enhancement strategies (e.g., CoT and scaling test-time compute) yield limited improvements, indicating that MMR-V presents a greater challenge to current multimodal reasoning models. Further error analysis in Section 4.5 shows that the CoT demanded in multimodal reasoning differs from those in textual reasoning. Current models tend to rely on textual reasoning based on visual information from the question frame and few adjacent frames, lacking the multimodal reasoning needed to locate and analyze evidence from long-range frames. This limitation hinders the overall reasoning performance. **(2) More modality will benefit:** We found that for models that support all modalities, adding additional audio modalities will improve the performance (Accuracy improved by 1.4%, 1.0%, and 1.0% for Gemini 2.0-Flash, Gemini 2.0-Flash-Thinking, and Phi-4-Multimodal-Instruct, respectively). **(3) Human-model gap:** In human experiments, we found that although models exhibit human-level performance on text reasoning tasks, there is still a significant gap between model and human on multimodal, especially video, reasoning tasks. We hope MMR-V will inspire further research into enhancing multimodal reasoning capabilities in AI systems.

## 2 TASK OVERVIEW

The tasks in MMR-V require deeper multimodal reasoning. Unlike previous tasks such as math and puzzle problems (Lu et al., 2023; Wang et al., 2024a; Zhang et al., 2024), we argue that the scope of multimodal reasoning should be more broadly defined. Previous work focuses more on text-oriented reasoning based on perceived visual information. In contrast, our task requires integrating the various forms of visual evidences, such as artistic style, lighting, and depth, into the reasoning process. Even more challenging, it involves **reasoning over long-range, multi-frame visual evidence**. Videos have a temporal dimension, which puts a greater challenge on the ability to find clues in different frames through multimodal reasoning.

### 2.1 DEFINITION FOR IMPLICIT AND EXPLICIT REASONING.

We categorize reasoning tasks in MMR-V into **Implicit Reasoning** and **Explicit Reasoning**, inspired by Kahneman's Dual Process Theory (Kahneman, 2011) and other cognitive theories (Evans, 1984; Sun, 2006; Polanyi, 2012). The most obvious difference is whether or not one needs to understand the subtext beneath the surface information. Secondly, implicit reasoning for human is often achieved by experience based on world knowledge, thus consuming little attention resources. Tasks are further divided into 10 categories and 33 subcategories. Figure 2 shows six categories (top row: implicit; bottom: explicit). Further explanations and examples can be found in Appendix E.

**Implicit Reasoning** focuses on incorporating **hidden meanings behind visual information** into reasoning. In these tasks, surface-level visual cues often conceal deeper layers of meaning, such as metaphor. Besides, for human, *"(implicit) operates automatically and quickly, with little or no effort and no sense of voluntary control." - Dual Process Theory.*

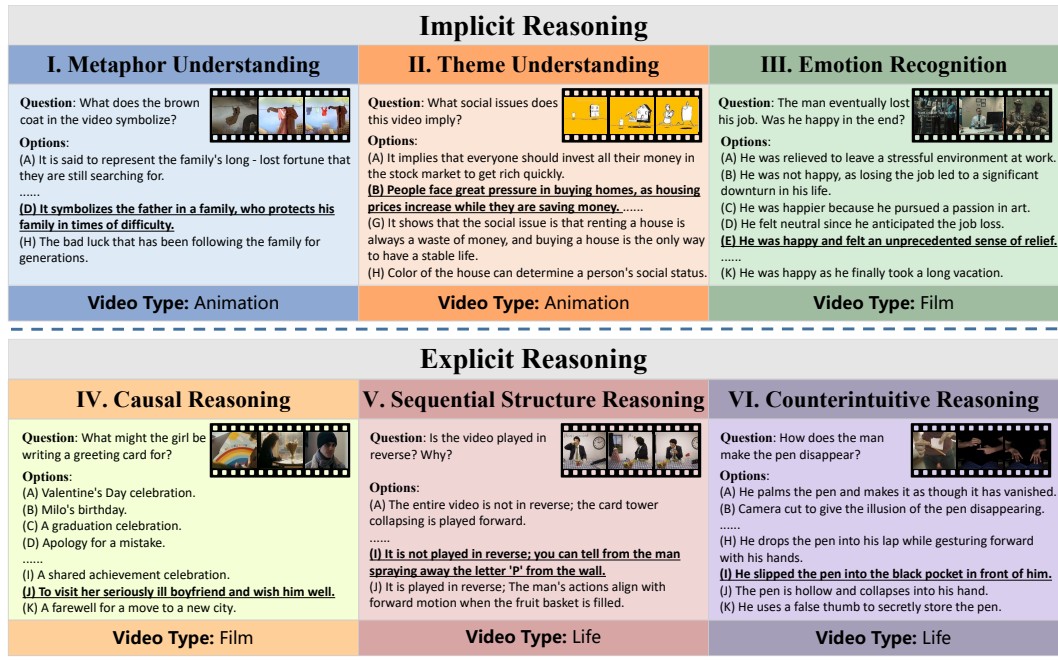

Figure 2: Overview of six tasks in MMR-V Bench.

**Explicit Reasoning** evaluates whether a model can perform reasoning based on multimodal details **explicitly presented** across long-range, multi-frame of a video. However, solving these tasks demands fine-grained perception and rigorous logical reasoning. *"(explicit) allocates attention to the effortful mental activities that demand it, including complex computations." - Dual Process Theory.*

## 2.2 IMPLICIT REASONING TASKS

**Metaphor Understanding (MU)**: MU tasks evaluate the ability to reason about metaphors for entities or environment. For example, the case in Figure 2 I interprets the metaphor of the brown coat.

**Theme Understanding (TU)**: TU assesses the ability to infer the main idea and attitude of the author through the full video. For example, the case in Figure 2 II asks what social issue the video reveals.

**Emotion Recognition (ER)**: ER tasks evaluate the ability to analyze character emotional states, as well as higher-level emotions such as the author's attitude and the audience's emotional response. For example, the case in Figure 2 III involves inferring whether the character feels happy at the end.

**Comment Matching (CM)**: CM task is to predict the most fitting audience comments for a video based on a criteria. For example, selecting which comment would be the most humorous after watching the video. Detailed example can be found in Appendix E.1.

**Implicit Symbol (IS)**: IS task is to interpret implicit symbols in the video, such as cultural elements. For example, inferring the ethnicity of the filming location. Details can be found in Appendix E.1.

## 2.3 EXPLICIT REASONING TASKS

**Causal Reasoning (CAR)**: CAR assesses the ability to reason about causal relationships in the video. For example, in Figure 2 IV, it involves inferring the reason why the girl is making a card.

**Sequential Structure Reasoning (SSR)**: SSR tasks assess reasoning about temporal structure in video editing and storytelling. In the example from Figure 2 V, the task is to infer if the video is reversed. However, the creator of this video explains the video is played normally.

**Counterintuitive Reasoning (CIR)**: CIR tasks evaluate the ability to analyze information that contradicts common sense, requiring detailed cross-frame analysis. In the example from Figure 2 VI, the task is to reason the principle behind the counterintuitive magic trick.

**Cross-modal Transfer Reasoning (CTR)**: To reason and match information out of the video that shares similar meaning. For example, find the quote with same theme of the video.

**Video Type and Intent (VTI)**: VTI tasks test the ability to infer key meta-level information such as the genre and communicative intent of the video from a global perspective. For example, the case in Appendix E.2 infers the release time by reasoning the video is set during COVID-19.

## 3 MMR-V BENCH

To ensure that MMR-V effectively evaluates multimodal reasoning abilities, we follow **three principles** during construction: **P1. Multi-frame:** Questions require reference to long-range, multi-frame information, prompting the model to reason across multiple visual cues. **P2. Deep reasoning:** Answers should not be directly perceivable from the video; instead, they should demand understanding of the subtext or multimodal reasoning, reflecting a deep comprehension of the content. **P3. Realistic:** Tasks should align with real-world question-answering needs, ensuring answers are consistent with common user understanding and free from individual cognitive biases or prejudices.

### 3.1 VIDEO COLLECTION

We manually curated diverse original videos from Youtube with the following checklist: (1) **Avoidance of linear, descriptive content**: We excluded videos with straightforward structures, such as daily recordings or sports broadcasts, in order to ensure that the tasks require deep reasoning over multi-frames (For Principle P1). (2) **Creative and thematically rich videos**: We selected videos that are intentional designed and edited by creators, often conveying well-crafted themes. This ensures that the questions require interpretation beyond surface-level visual content (For Principle P2). (3) **Alignment with real-world**: Highly Popular Videos were preferred, which are indicated by active comment sections and audience engagement. This helps avoid biases introduced by niche content and ensures alignment with general user cognition (For Principle P3). (4) **Diverse coverage**: To further promote generalizability, we ensured broad coverage across video types, topics, and durations, allowing MMR-V to reflect the diversity of real-world video content (For Principle P3). As a result, our final benchmark comprises **317 videos** spanning **six major categories**: Animation, Film, Philosophy, TV, Life, and Art. The specific categories are shown in the Appendix D. Furthermore, for problems where audio might be helpful, we ensure that the videos include audio.

### 3.2 DATA ANNOTATION & QUALITY ASSURANCE

All tasks in MMR-V Bench are designed in a multiple-choice format. There is one correct option and several wrong options. We make sure there are carefully crafted distractors among the wrong options. To ensure the quality and plausibility of these distractors, we designed three distinct distractors annotation strategies. (1) Str. 1: We prompt a strong model GPT-4o (Hurst et al., 2024) to directly answer the manually annotated question. If the model generates an incorrect answer (as verified by human annotators), that answer is retained as a high-quality distractor. If correct, we combine human-written distractors with incorrect options generated by GPT-4o as distractors. (2) Str. 2: Given the question and correct answer annotated manually, GPT-4o is prompted to generate distractors. (3) Str. 3: Human annotators construct distractors manually.

We conducted a test using 100 questions, using three strategies to form three test-set with 100 multiple-choice tasks. As shown in Table 1, distractors generated by strategy 1 are more confusing, significantly increasing the difficulty and quality of our tasks. It is worth noting that in the above test process, when GPT-4o directly answered 100 tasks, the accuracy rate verified by humans was only 17%. This reflects the limitations of the current model in multimodal reasoning capabilities.

Table 1: Performance on 100 questions annotated with different strategies (str.).

| Models | Str. 1 | Str. 2 | Str. 3 |
|---|---|---|---|
| GPT-4o | 59% | 70% | 62% |
| Qwen-VL-7B | 37% | 51% | 42% |

To ensure high quality, we also developed an checklist based on the construction principles and invited human annotators to verify the accuracy and difficulty of the tasks using this checklist. We invited five annotators with at least a bachelor's degree to participate in the annotation and review process. The checklist of MMR-V is shown in the Appendix C. The overall annotation process and the annotation platform can be found in Figure 7 and Figure 8 in the Appendix C.

## 3.3 DATA STATISTICS

MMR-V comprises a total of 317 videos spanning a wide range of content types, and includes 1,257 multiple-choice reasoning tasks. Each question is annotated with 7 to 11 candidate answers, with only one correct answer guaranteed. As illustrated in Figure 9a, the videos are categorized into six major domains, each encompassing fine-grained subcategories to ensure diversity in content, style, and semantics. The reasoning tasks in our benchmark are organized across three levels of granularity, reflecting different dimensions of reasoning complexity and modality. The distribution of task types across these levels is shown in Figure 9b. More information is shown in Table 2. Table 5 shows a statistic comparison with previous benchmarks.

Table 2: Dataset Statistic of MMR-V.

| Dataset | Statistic |
|---|---|
| *Task* | |
| Question Count | 1257 |
| Average Option Count | 10 |
| Average Question Words | 14 |
| Average Option Words | 10 |
| *Video* | |
| Video Count | 317 |
| Minimum Length (s) | 7 |
| Maximum Length (s) | 3771 |
| Average Length (s) | 277 |

## 4 EXPERIMENTS

### 4.1 SETTINGS

We conducted extensive evaluations on 11 proprietary and 10 open-source models as detailed in the Appendix F.1. Our main experiments were conducted under two settings: zero-shot and zero-shot + CoT (Wei et al., 2022), in order to examine whether reasoning enhances performance. For further analysis, we introduced the following categories of comparative models: (1) Models with different scales. (2) "Thinking" model and its base version. (e.g., Gemini-Flash and Gemini-Flash-Thinking).

**Multimodal Inputs**: For models supporting full-modal inputs (e.g., Gemini-2.0-flash), we further compare their performance with and without audio input to evaluate its influence on reasoning.

**Frame Selection**: Since some models only support multiple images or short video clips, we standardized the number of input frames. Details of frame sampling are provided in Appendix F.

**Human Experiment:** To provide a meaningful upper bound for MMR-V and to examine the human-model gap, we invited participants with at least bachelor degree to conduct human experiment. We sampled 100 tasks GPT-4o answered incorrectly and 100 tasks it answered correctly for experiment.

### 4.2 MAIN RESULTS

Results in Table 3 highlight the challenge of MMR-V. The highest score was achieved by Gemini-2.5-pro with 1 fps video sampling, reaching only 64.3%. Under the setting of fixed input frame number, GPT-5 obtained the best score of 60.9%. Among open-source models, Gemma-3-27b-it performs the best, demonstrating relatively strong performance. However, there remains a gap compared to proprietary models. The performance on videos of different lengths is shown in the Appendix G.1.

**Current reasoning enhancements have limitations on MMR-V.** Results in Table 3 show that current reasoning enhancement strategies, which are relatively effective in textual domains, such as CoT prompt reasoning and scaling test-time compute (i.e., "Thinking" models), offer only limited gains on MMR-V. CoT brings only a 0.88% average gain, and "Thinking" model improves just 2.4%. This indicates that MMR-V presents a significant challenge to the multimodal reasoning capabilities of existing models. Analysis of sampled model responses shows that visual analysis accounts for only about 10% of the CoTs. This reveals that reasoning process of current model is mostly text-based (reasoning on questions and options), relying on visual perception of question frame, instead of integrating visual reasoning and evidence mining into CoTs. Several examples are provided in Appendix J, and further analysis in Section 4.5 supports similar findings.

**Model performance on MMR-V Bench shows a clear scaling law effect.** Smaller models under the same architecture perform poorly on tasks that require complex reasoning. For instance, larger models like Qwen2.5-VL-72B (39.1%) and GPT-4o (52.8%) outperform their smaller versions Qwen2.5-VL-7B (30.1%) and GPT-4o-mini (34.8%), showing relative gains of 9% and 18%, respectively.

**Model performance across different tasks on MMR-V Bench.**

Table 3: Evaluation results (%) on MMR-V. Results under CoT prompting are highlighted in gray. The random accuracy on MMR-V Bench is approximately 10%. **Bold** and underlined values indicate the best performance among proprietary and open-source models, respectively.

| Model | Overall | | Implicit | | Explicit | | Art | Life | TV | Film | Ani. | Phi. |
|---|---|---|---|---|---|---|---|---|---|---|---|---|
| | | | Tasks | | | | Video Categories | | | | | |
| *Open-source models* | | | | | | | | | | | | |
| LLaVA-Video | 18.4 | 17.6 | 19.1 | 18.1 | 15.4 | 16.3 | 14.4 | 11.2 | 13.2 | 17.4 | 21.4 | 12.8 |
| NVILA-8B-Video | 25.5 | 25.3 | 26.2 | 24.2 | 23.9 | 25.9 | 17.3 | 21.3 | 23.5 | 21.6 | 38.0 | 21.8 |
| Phi-4-multimodal-instruct | 26.7 | 27.6 | 29.4 | 31.2 | 19.4 | 18.1 | 19.4 | 19.2 | 25.9 | 26.4 | 33.9 | 24.4 |
| Cogvlm2-video-llama3 | 25.6 | 26.1 | 25.4 | 26.2 | 26.1 | 25.7 | 15.5 | 18.3 | 24.7 | 19.1 | 43.2 | 20.8 |
| Qwen2.5-VL-7B | 30.1 | 32.4 | 33.7 | 36.2 | 20.8 | 22.5 | 20.9 | 18.1 | 29.6 | 21.2 | 48.4 | 19.8 |
| Intern3-8B | 33.6 | 32.9 | 35.5 | 33.4 | 28.6 | 31.4 | 23.0 | 22.6 | 31.7 | 24.3 | 52.9 | 23.2 |
| Gemma-3-12b-it | 34.0 | 34.2 | 37.8 | 37.6 | 24.0 | 25.4 | 19.4 | 24.9 | 25.9 | 31.3 | 51.9 | 24.4 |
| InternVL2.5-38B | 39.9 | 39.7 | 43.8 | 43.7 | 29.9 | 29.4 | 30.4 | 28.8 | 30.4 | 37.2 | 57.4 | 29.1 |
| Qwen2.5-VL-72B | 39.1 | 40.4 | 41.3 | 42.8 | 33.4 | 34.3 | 28.9 | 28.2 | 29.1 | 36.5 | 55.6 | 37.2 |
| Gemma-3-27b-it | 42.0 | 41.1 | 46.5 | 44.7 | 30.3 | 32.0 | 31.7 | 32.2 | 35.5 | 41.3 | 56.1 | 33.7 |
| *Proprietary models* | | | | | | | | | | | | |
| GPT-4o-mini-2024-07-18 | 34.8 | 35.2 | 38.0 | 38.6 | 26.3 | 26.3 | 29.5 | 25.4 | 29.6 | 33.0 | 48.7 | 18.6 |
| Gemini-2.0-Flash (16 frames) | 42.6 | 44.3 | 44.3 | 45.9 | 38.3 | 40.0 | 30.9 | 32.2 | 40.7 | 40.6 | 58.5 | 24.4 |
| Claude-3.5-Sonnet-20241022 | 43.3 | 44.2 | 45.0 | 46.1 | 38.9 | 39.1 | 33.8 | 31.1 | 41.3 | 41.3 | 55.8 | 4.4 |
| Gemini-2.0-Flash-thinking | 45.0 | 43.5 | 46.6 | 46.0 | 40.6 | 37.1 | 34.5 | 31.6 | 38.6 | 48.3 | 60.1 | 25.6 |
| GPT-4.1-2025-04-14 | 46.6 | 48.9 | 49.1 | 51.7 | 40.3 | 41.7 | 43.2 | 35.6 | 43.9 | 46.5 | 57.1 | 34.9 |
| Gemini-2.0-Flash (512 frames) | 48.0 | 49.9 | 50.5 | 52.6 | 41.6 | 42.9 | 36.7 | 36.7 | 39.7 | 46.2 | 66.7 | 31.4 |
| Gemini-2.5-Flash | 51.2 | 50.5 | 52.9 | 52.3 | 46.9 | 45.7 | 45.3 | 39.5 | 50.3 | 47.9 | 65.6 | 34.9 |
| o4-mini-2025-04-16 | 52.5 | 52.1 | 54.6 | 54.5 | 47.1 | 46.0 | 48.2 | 40.1 | 54.0 | 51.7 | 65.3 | 27.9 |
| GPT-4o-2024-11-20 | 52.8 | 55.0 | 54.7 | 56.3 | 48.1 | 51.4 | 46.0 | 42.5 | 50.0 | 49.0 | 67.7 | 38.4 |
| o3-2025-04-16 | 59.1 | 58.3 | 60.1 | 57.8 | 57.6 | 59.7 | 51.8 | 45.2 | 55.6 | 58.3 | 74.3 | 43.0 |
| GPT-5-2025-08-07 | 60.6 | 60.9 | 61.0 | 60.6 | 59.7 | **61.4** | 56.8 | 41.8 | 56.6 | 60.4 | **76.5** | 45.4 |
| Gemini-2.5-pro (1fps) | **64.3** | **63.2** | **65.9** | **64.4** | **60.2** | 60.3 | **57.6** | **58.1** | **65.6** | **62.5** | 73.0 | **54.6** |
| *Baseline* | | | | | | | | | | | | |
| *Best Performance of Models* | 64.3 | | 65.9 | | 60.3 | | 57.6 | 58.1 | 65.6 | 62.5 | 74.3 | 54.6 |
| *Human* | 86.0 | | 80.6 | | 91.2 | | 57.7 | 92.3 | 90.6 | 92.3 | 90.7 | 70.0 |

**Firstly, the models performed better on implicit tasks than on explicit tasks** (with an average gain of +7.9%). Through analysis of tasks and model responses, we found that in implicit tasks, video creators often embed implicit meanings throughout the entire video, resulting in abundant visual cues that can support reasoning. This reduces the requirements for multi-modal reasoning and clue localization. In contrast, explicit tasks demand finer-grained reasoning and the ability to identify specific evidence. For example, in the implicit task at the bottom of Figure 1, many frames provide clues suggesting that the girl symbolizes good luck (e.g., room number, flowers, lighting, weather, etc.). In contrast, the explicit task at the top contains only a few key frames where the hidden lighter in magician's hand can be seen.

**Secondly, the models performed particularly poorly on *Counterintuitive Reasoning (CIR)*, *Sequential Structure Reasoning (SSR)*, and *Comment Matching (CM)* tasks**. For CIR and SSR, poor performance mainly stems from the limited ability of models to perform multi-frame reasoning. These two tasks require the model to reason on long-range videos, rather than relying on internal knowledge. However, instead of analyzing to locate evidences in other frames, models often rely on surface-level visual perception of the question frame, followed by textual reasoning over question and options. For CM, the results highlight a significant gap between model and human capabilities in implicit reasoning. While humans can infer underlying information such as humor and emotion with minimal cognitive effort (Krishna et al., 2022), current models consistently fail to capture such subtleties.

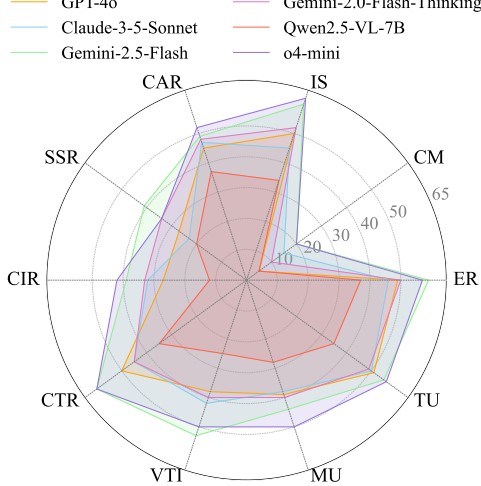

Figure 3: Performance on different tasks.

Table 4: The impact of adding audio modality on the performance (accuracy %) on different tasks.

| | Overall | Tasks | | Categories | | | | | |
| --- | --- | --- | --- | --- | --- | --- | --- | --- | --- |
| | | *Imp.* | *Exp.* | *Art* | *Life* | *TV* | *Film* | *Ani.* | *Phi.* |
| Gemini-2.0 | 42.6 | 44.3 | 38.3 | 30.9 | 32.2 | 40.7 | 40.6 | 58.5 | 24.4 |
| +audio | $44.0^{\uparrow 1.4}$ | $46.2^{\uparrow 1.9}$ | $38.3^{-0.0}$ | $31.0^{\uparrow 0.1}$ | $31.6^{\downarrow 0.6}$ | $42.3^{\uparrow 1.6}$ | $41.0^{\uparrow 0.4}$ | $61.1^{\uparrow 2.6}$ | $29.1^{\uparrow 4.7}$ |
| Gemini-2.0-thinking | 45.0 | 46.6 | 40.6 | 34.5 | 31.6 | 38.6 | 48.3 | 60.1 | 25.6 |
| +audio | $46.0^{\uparrow 1.0}$ | $48.4^{\uparrow 1.8}$ | $39.7^{\downarrow 0.9}$ | $31.7^{\downarrow 2.8}$ | $33.9^{\uparrow 2.3}$ | $44.4^{\uparrow 5.8}$ | $42.7^{\downarrow 5.6}$ | $62.4^{\uparrow 2.3}$ | $32.6^{\uparrow 7.0}$ |
| Phi-4-multimodal-instruct | 26.7 | 29.4 | 19.4 | 19.4 | 19.2 | 25.9 | 26.4 | 33.9 | 24.4 |
| +audio | $27.7^{\uparrow 1.0}$ | $31.3^{\uparrow 1.9}$ | $18.1^{\downarrow 1.3}$ | $15.4^{\downarrow 3.0}$ | $19.7^{\uparrow 0.5}$ | $24.5^{\downarrow 1.4}$ | $27.8^{\uparrow 1.4}$ | $37.3^{\uparrow 3.4}$ | $26.7^{\uparrow 2.3}$ |

**Human Performance.** Humans achieved an average score of 86%, highlighting a significant human-model gap. Although studies suggest models reached human-level performance on text tasks (Guo et al., 2025; OpenAI, 2023), they still lag on multimodal reasoning. Humans can identify clues in videos easily, while models tend to focus on question frames rather than exploring other evidence frames. Specially, unlike models, humans perform slightly worse on implicit tasks, which is mainly due to the challenges posed by highly abstract implicit understanding in art and philosophy.

## 4.3 INFLUENCE OF FRAMES COUNT

For Gemini-2.0-Flash, which supports long video inputs, we evaluated performance changes as the number of frames increases. As shown in Figure 4, accuracy improves with more frames, but the rate of improvement gradually slows. After sampling and observing the CoTs, it is found that the initial gains come from the addition of evidence frames, while the slowdown is mainly due to limited multi-frame reasoning ability of the model. Performance on implicit tasks continues to improve in later stages, as visual cues for such tasks are often dispersed throughout the video (as discussed in Section 4.2); more frames tend to provide more clues. In contrast, explicit clues are fewer and more localized.

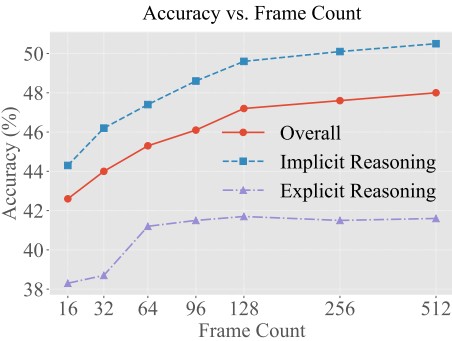

Figure 4: Accuracy with the increase of input frame counts.

## 4.4 INFLUENCE OF AUDIO INPUT

For models supporting full-modal input, we compared performance before and after incorporating audio. As shown in Table 4, overall performance improved with the addition of audio. Specifically, Gemini 2.0-Flash, Gemini 2.0-Flash-Thinking, and Phi-4-multimodal-instruct improved by 1.4%, 1.0%, and 1.0%, respectively. This suggests that advancing research on fully multimodal models is a promising direction. A case study illustrating how audio aids reasoning is provided in Appendix G.2.

## 4.5 ERROR ANALYSIS

We sampled 100 incorrect responses from GPT-4o for error analysis. The errors can be categorized as follows: **(1) Lack of Visual Reasoning:** the model often failed to locate the correct evidence frames and lack of long-range, multi-frame visual reasoning. **(2) Implicit Misinterpretation:** revealing a significant understanding gap between the model and human cognition. **(3) Knowledge Insufficiency:** the model lacks some intrinsic knowledge **(4) Reasoning Error:** during the multi-step deduction process. **(5) Hallucination:** the model introduced fake or unsupported information. **(6) Output Formatting Issue:** model refusals or formatting errors prevent answer extraction. Among error cases, *Lack of Visual Reasoning* accounts for the largest proportion. This indicates that

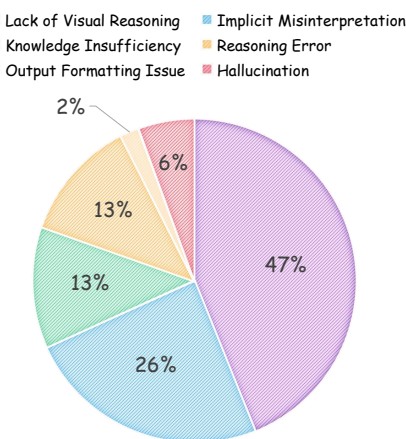

Figure 5: Error analysis of GPT-4o.

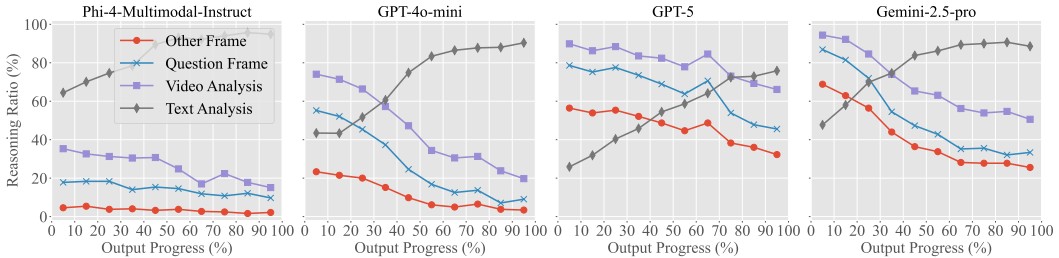

Figure 6: CoT content across different stages. The y-axis indicates the ratio of the 500 sampled CoTs that include analysis of these four types of content at each stage.

current models still lack genuine multimodal reasoning capabilities. They tend to rely on text-based reasoning after briefly perceiving frames adjacent to the question, rather than engaging in long-range, multi-frame video reasoning. Most existing reasoning models remain inadequate in integrating multimodal information into the reasoning process and performing thorough analysis. In contrast, models like GPT-5, o3 exhibits a better reasoning paradigm, as shown in Figure 13 for comparison. Details of the error category annotation pipeline are provided in Appendix C.3.

We further analyzed model CoTs to examine performance differences by categorizing each reasoning step as either video or text analysis (e.g., options), with video analysis further divided into question-frame and other-frame analysis (details in Appendix H). We sampled 500 CoTs from models, split each into 10 equal-length segments, and used GPT-4.1 to label each segment. As shown in Figure 6, where models further to the right perform better on MMR-V, models with better performance on MMR-V show more video analysis, especially on **other frames** (red line). Notably, GPT-5 and Gemini-2.5-pro stand out with strong analysis of non-question frames, highlighting the value of enhanced multi-frame visual reasoning in video reasoning tasks.

## 5    RELATED WORK

**Video Understanding Benchmark.** Existing video benchmarks mainly evaluate models' perception and understanding of video, such as action recognition (Khattak et al., 2024; Mangalam et al., 2023; Patraucean et al., 2023; Xiao et al., 2021) and description (Xu et al., 2017; 2016). Recent works like Video-MME (Fu et al., 2024), MVBench (Li et al., 2024b), TempCompass (Liu et al., 2024a) and MMBench-Video (Fang et al., 2024) extend video understanding to diverse video types, enabling more comprehensive assessments. Benchmarks such as LVBench (Wang et al., 2024b), LongVideoBench (Wu et al., 2024), and CG-Bench (Chen et al., 2024) further introduce long video QA. However, these largely test whether models can extract relevant information from long videos, with evaluation remaining perception-oriented. MMR-V instead assesses whether models can perform multi-frame, long-span, multimodal autonomous reasoning from questions.

**Multimodal Reasoning.** Recent advancements has greatly enhanced LLM reasoning (Guo et al., 2025; Jaech et al., 2024; Team et al., 2025; Zhao et al., 2024), yet the evaluations still focus on text-based reasoning (Hendrycks et al., 2021; Bai et al., 2024; Cobbe et al., 2021; Rein et al., 2024; Wang et al., 2024c). MLLMs still lack thorough assessment in this area (Qin et al., 2023; Wang et al., 2023). Current multimodal reasoning benchmarks mainly involve mathematical or coding tasks in image form (Wang et al., 2024a; Shi et al., 2024; Ying et al., 2024), which primarily test visual recognition followed by text reasoning. True multimodal reasoning requires richer cross-modal cues. To address this gap, MMR-V is designed to evaluate video-based reasoning tasks.

**Video Reasoning.** Recently, several benchmarks have emerged to access reasoning in videos. VideoEspresso (Han et al., 2025) utilizes a fully automated pipeline to generate scalable CoT reasoning data in videos. Video-MMMU (Hu et al., 2025) and MMVU (Zhao et al., 2025) focus on evaluating specialized knowledge reasoning within multi-disciplinary videos (e.g., physics, economics). VRBench (Yu et al., 2025) targets narrative reasoning in long narrative videos and designs a human-model collaborative stepwise annotation pipeline. VideoReasonBench (Liu et al., 2025) assesses vision-centric reasoning using synthetic videos generated by engines. MMR-V distinguishes itself by ensuring all QA pairs are purely manually annotated, evaluating general multimodal reasoning across diverse real-world open-domain videos.

## 6 CONCLUSION

This paper introduces **MMR-V: A Benchmark for Multimodal Deep Reasoning in Videos**. All tasks are human-annotated. MMR-V poses a significant challenge for current models, with the best performance still trailing human accuracy by 21.7%. This highlights a human-model gap in interpreting and reasoning about video information. Notably, we observe that models with higher accuracy on MMR-V tend to perform more extensive and in-depth analysis of videos, suggesting that incorporating multi-frame reasoning into CoT or leveraging tool use may be promising directions for advancing video reasoning. We hope MMR-V will serve as a reliable evaluation benchmark for the development of MLLMs and offer valuable insights into advancing multimodal reasoning research.

## ACKNOWLEDGEMENTS

This work is supported by the National Natural Science Foundation of China (No. U24A20335, No. 62406321). This work is also supported by Beijing Natural Science Foundation (L243006).

## ETHICS STATEMENT

All experimental procedures involving human participants were conducted in accordance with the relevant ethical guidelines. The MMR-V data was gathered and annotated by compensated internal annotators, from whom all necessary informed consent was obtained. The dataset was carefully reviewed for safety and potential biases. We have prepared the dataset for public release under the CC-BY 4.0 license.

## REPRODUCIBILITY STATEMENT

We have taken several steps to enhance the reproducibility of our research. The code for the experiments and analyses reported in the main text is provided in the supplementary materials. The experimental setup of the main experiments and the selection of video frames are detailed in Section 4.1 and Appendix F. Further details regarding dataset construction and preprocessing are described in Appendix C and Appendix E. Finally, our dataset has been prepared for open release to support future research. The JSON file containing the questions, answers, and related information for MMR-V can be found in the supplementary materials.

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

## A    LLM USAGE STATEMENT

In this paper, LLMs were used to check grammar and polish text, and we carefully verified that their use did not alter the original meaning. In addition, AI assistants were employed to improve the presentation of some tables and figures, but we ensured the experimental results remained unchanged.

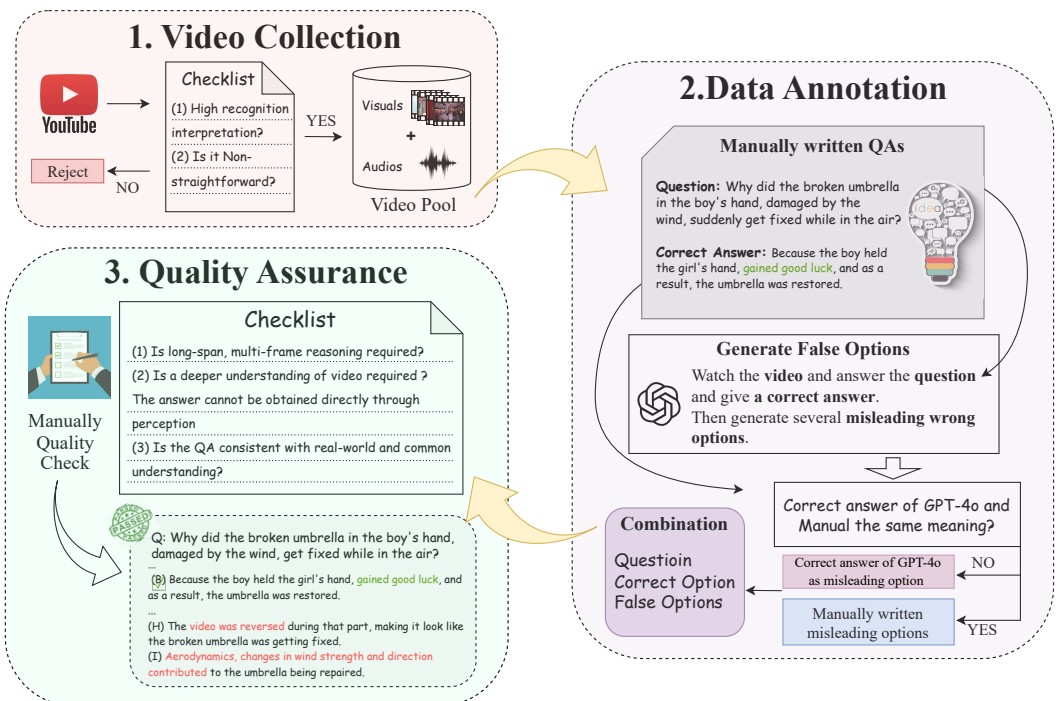

Figure 7: MMR-V Construction Pipeline.

## B  LIMITATIONS

Despite our efforts to improve our work, several limitations remain. (1) Scaling MMR-V is challenging due to the high cost of manual annotation and verification, as all tasks and correct answers are curated and reviewed by human annotators. (2) Although we strive to cover a wide range of video and task types, certain real-world categories (such as mystery, puzzle-solving, and gaming) are still underrepresented. (3) The majority of videos in MMR-V are in English, with only a small proportion in other languages such as Chinese, French, Thai, and German, which constrains its multilingual applicability. We will further study and try to solve this issue in the future.

## C  MMR-V CONSTRUCTION

### C.1  CHECKLIST

According to the MMR-V construction principles introduced in the main text Section 3 , we wrote the following annotation checklist:

> (1) You are expected to **watch the entire video** before formulating any questions or answers.
> (2) Each question must require **long-distance, multi-frame reasoning** and cannot be answered through direct perception (ensuring compliance with Principles 1 and 2).
> (3) To ensure **consistency with real-world user** perception (Principle 3), annotators are encouraged to refer to the official interpretation of the original video author and user consensus (highly praised comments in the comment section) when writing or verifying the correct answer. This helps mitigate annotator bias and ensures that the reasoning task reflects the understanding of a wider audience.

### C.2  CONSTRUCTION PIPELINE

In this section, we present the construction process of MMR-V Bench in a macro sense. The whole process is divided into three stages: **video collection**, **data annotation**, and **quality assurance**. For **video collection**, we designed a checklist to ensure the quality and diversity of videos in the Bench. "High recognition interpretation?" ensures that the questions raised and the annotated answers based

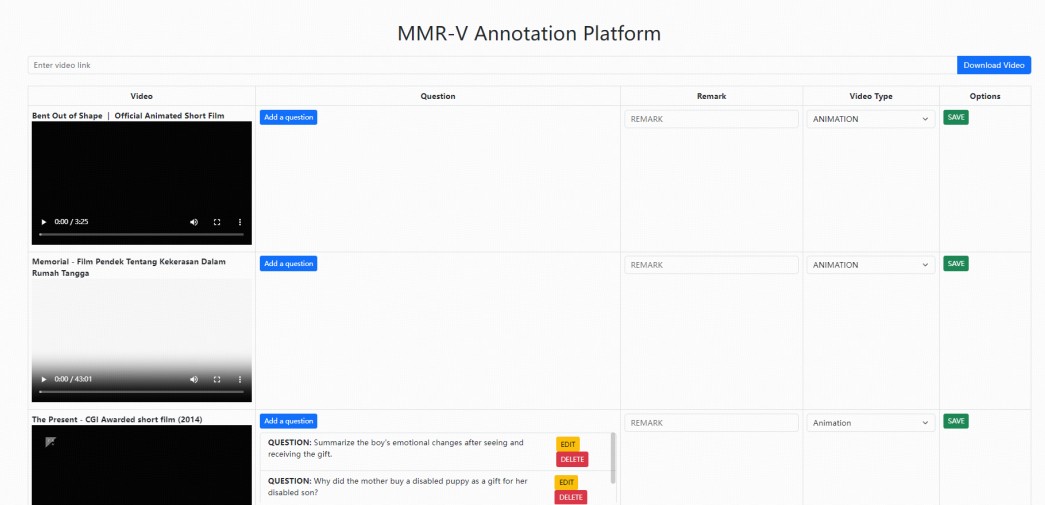

Figure 8: Annotation Platform of MMR-V.

on the video have references that are consistent with public cognition (official interpretations or highly praised comments) to alleviate the subjective bias of the annotator. "Is it Non-straightforward?" ensures that the video is not a straightforward narrative, which is conducive to increasing the reasoning difficulty of the question. For **data annotation**, as described in section 3.2 of the main text, we use gpt-4o to assist in annotation with interference options. Let the model generate the correct answer based on the question, and manually review to ensure that the correct answer generated by the model is different from the manual annotation. If they are different, the answer generated by gpt-4o is used as the interference item, otherwise the interference item is manually written. For **quality assurance**, we designed a checklist for human reviewers to check the correctness and difficulty of the tasks. The annotation platform is shown in Figure 8.

### C.3    PIPELINE OF ERROR ANALYSIS

To ensure the reliability of the error analysis presented in Section 4.5, we implemented an annotation pipeline and quality control strategy. Firstly, we categorized model failures into six distinct types based on initial analysis of the false answers. To standardize the criteria, we selected representative anchor examples for each category to distinguish boundary cases. Some examples are presented in Appendix J. Next, to mitigate author bias, we invited non-author evaluators to review the definitions and anchor examples, ensuring they align with general human understanding before formal annotation. Finally, the primary authors annotated the sampled responses based on the established guidelines. Any disagreements were discussed and resolved to ensure 100% consensus of authors on the final error distribution.

## D    DIVERSITY OF MMR-V

In this section, we show the diversity of MMR-V Bench, including video diversity and task diversity. For video, we show the six categories of videos in MMR-V in Figure 9a, including Life, Animation, Film, Art, TV, and Philosophy. At the same time, for each category, we divide it into several subcategories to better understand the classification of video categories. Secondly, in section 3.3 we show the diversity of video length, ranging from 7 seconds to 3771 seconds. For tasks, we divide them into two parts, ten categories and 33 subcategories, three levels. The division of the first and second levels, as well as the proportion of different types of tasks, can be seen in Figure 9b.

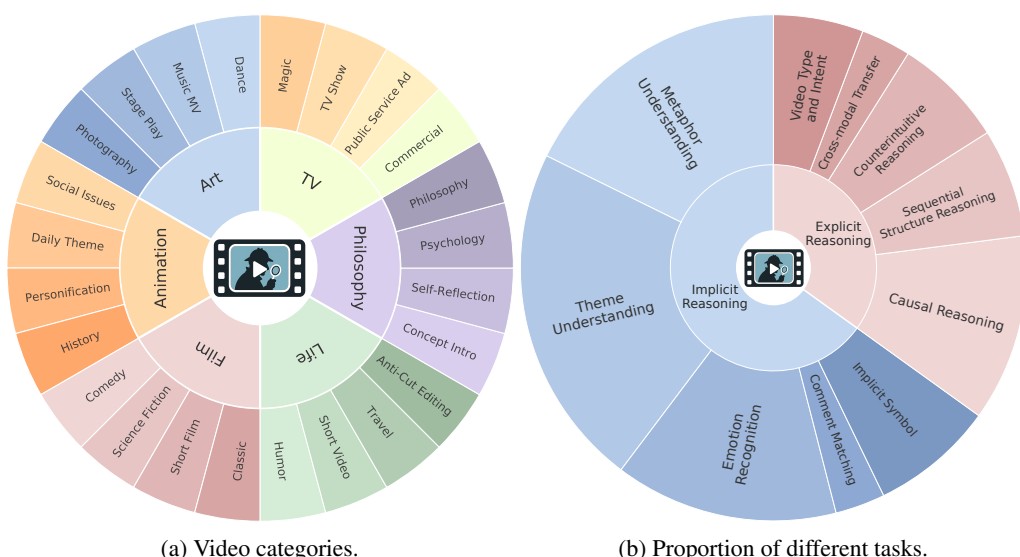

(a) Video categories.       (b) Proportion of different tasks.

Figure 9: (a) Video categories in MMR-V Bench. (b) Proportion of different tasks in MMR-V Bench.

Table 5: Comparison with existing video benchmarks. **#Dur.**: Average video duration in seconds. **QA Anno.**: Annotation method (A: Automatic, M: Manual, A&M: Both).

| Benchmark | #Videos | #Dur.(s) | #QA Pairs | Data Source | Task Categories | Multilingual | QA Anno. |
|---|---|---|---|---|---|---|---|
| MVBench | 3,641 | 16.0 | 4,000 | Open-Domain | 20 | × | A |
| EgoSchema | 5,063 | 180.0 | 5,063 | Egocentric | × | × | A |
| TempCompass | 410 | 11.4 | 7,540 | Open-Domain | 11 | × | A&M |
| Video-MME | 900 | 1017.9 | 2,700 | Open-Domain | 12 | ✓ | M |
| LongVideoBench | 3,763 | 473.0 | 6,678 | Open-Domain | 17 | × | M |
| LVBench | 103 | 4101.0 | 1,549 | Open-Domain | 26 | × | M |
| CGBench | 1,219 | 1624.4 | 12,129 | Open-Domain | 12 | × | M |
| Video-MMMU | 300 | 506.2 | 900 | Open-Domain | 6 | × | M |
| MMVU | 1,529 | 51.4 | 3,000 | Multi-Disc | × | × | M |
| VRBench | 960 | 5796.0 | 8,243 | Narrative | 7 | ✓ | A&M |
| **MMR-V (Ours)** | 317 | 277.0 | 1,257 | Open-Domain | 33 | ✓ | M |

## E  TASK DETAILS

The tasks in MMR-V can be divided into three levels. Level 1: Implicit Reasoning & Explicit Reasoning. Level 2: Contains ten task classes. Level 3: Contains 33 task subclasses. Next, we will introduce these tasks with some task examples[1].

### E.1  IMPLICIT REASONING TASKS

#### I. METAPHOR UNDERSTANDING (MU)

For the definition of subclasses of the metaphor understanding task, we mainly refer to the book Metaphors We Live (Lakoff & Johnson, 2008) By by George Lakoff and Mark Johnson, which introduces metaphor-related concepts in detail.

---

[1]The videos linked here are the original source versions. Note that the actual videos in MMR-V were processed to trim frames containing shortcuts to the answers.

Table 6: Three-level classification of tasks in MMR-V.

| Ability Type L1 | Ability Type L2 | Ability Type L3 |
|---|---|---|
| Implicit Reasoning | Metaphor Understanding (MU) | Structural Metaphor, Orientational Metaphor, Ontological Metaphor, Creative Metaphor |
| | Theme Understanding (TU) | Philosophical Concepts, Social Issues, Personal Reflection, Everyday Topics, Video Naming |
| | Emotion Recognition (ER) | Explicit Emotion, Implicit Emotion, Meta-emotion, Audience Emotion |
| | Comment Matching (CM) | Humorous, Thought-provoking, Trending |
| | Implicit Symbol (IS) | Cultural Symbols, Art Symbols, Other Symbols |
| Explicit Reasoning | Causal Reasoning (CAR) | Forward Reasoning, Backward Reasoning |
| | Sequential Structure Reasoning (SSR) | Narrative Structure, Core Connecting Elements, Inference on Editing Techniques, Hallucination |
| | Counterintuitive Reasoning (CIR) | Magic Deconstruction or Special Effects Editing, Artistic Techniques, Humor and Exaggeration |
| | Cross-modal Transfer Reasoning (CTR) | Video-to-Text, Video-to-Audio, Video-to-Video |
| | Video Type and Intent (VTI) | Video Type, Video Intent |

## I.1. STRUCTURAL METAPHOR

**Task Description**: There are structural similarities between the subject and object. For example, time can be compared to flowing water, both of which have the structure of flow and passing away.
**Example Question**:

> **Question**: What does the brown coat in the video symbolize?
> **Options**:
> (A) It is said to represent the family's long - lost fortune that they are still searching for.
> (B) The brown coat symbolizes the lost hope of the family because it was worn during a difficult time.
> (C) It refers to a coat that has been washed and taken out to dry, likely worn by the father.",
> (D) It symbolizes the father in a family, who protects his family in times of difficulty.
> (E) It represents the fear of the outside world.
> (F) The unfulfilled dreams of the children in the family as they always saw it as a sign of something unattainable.
> (G) The brown coat in the video represents a raincoat, used to protect the clothes inside from getting wet.
> (H) The bad luck that has been following the family for generations.
> **CorrectAnswer**: (D)
> **Video**: father - 1 minute emotional award winning - video_url

## I.2. ORIENTATIONAL METAPHOR

**Task Description**: There are similarities in direction or composition between the subject and the metaphor, for example, walking up a staircase is compared to ambition.
**Example Question**:

> **Question**: Why does the dance, which is filled with artistry and beauty throughout, end with a descent?
> **Options**:
> (A) There is a connection between the fall and the creation at some point.
> (B) It represents a dive to explore new depths, both literal and metaphorical.
> (C) It indicates the dancer's exhaustion, capturing a moment of fatigue.
> (D) It reflects the calmness of the ocean, evoking a sense of tranquility.
> (E) It highlights the theme of rebirth, symbolizing renewal and transformation.
> (F) It represents the beauty of underwater life, showcasing its unique allure.

(G) It symbolizes being weighed down by emotions, expressing inner turmoil.
(H) It symbolizes the end of a dream, marking a moment of conclusion.
(I) It shows the dancer's connection to water, emphasizing fluidity and grace.
(J) It symbolizes a return to nature and surrender to life's forces, embracing the natural flow.
(K) It signifies the end of the dance's energy, indicating a point of culmination.
**CorrectAnswer**: (A)
**Video**: Falling - Underwater dance - video_url

## I.3. ONTOLOGICAL METAPHOR

**Task Description**: This metaphor involves viewing an abstract concept as a concrete entity. Usually, the core concept of the entire video is turned into a concrete entity to tell the story.
**Example Question**:

**Question**: The scene around 1:00 metaphorically represents what aspect of communities?
**Options**:
(A) Communities can build their resilience to setbacks by working together and adapting to new challenges.
(B) Promoting individual success in competitive environments.
(C) Building resilience through community partnerships.
(D) Overcoming challenges for community progress.
(E) Celebrating the individual achievements of community members.
(F) Developing sustainable practices for environmental harmony.
(G) Decision-making processes of a community.
(H) The interconnectedness of global communities.
(I) Isolation of communities for self-sufficiency.
(J) The role of external aid in community development.
(K) Highlighting the diversity of cultures within a community.
**CorrectAnswer**: (A)
**Video**: Resilience: Anticipate, organise, adapt - video_url

## I.4. CREATIVE METAPHOR

**Task Description**: This metaphor is usually carefully designed by the author for a specific video and needs to be understood in the context of the video.
**Example Question**:

**Question**: What is the pink fairy ball in the film?
**Options**:
(A) It's a toy the boy picked up on the street, having no special connection to his condition.
(B) They are the microorganisms in this world, living in every corner.
(C) The pink fairy ball represents the boy's childhood dream of becoming a fairy.
(D) It's a hallucination caused by lack of sleep, not related to antidepressants at all.
(E) They are the boy's toys, which he bought to help treat his depression.
(F) It is the effect of the antidepressants the boy is taking, which helps him see many things with vitality and positive effects.
(G) It's an advertisement prop for a new product in the background of the scene.
(H) The ball is a sign of the boy's wish to escape from his daily work routine.
(I) The pink fairy ball is a symbol of the city's upcoming festival decorations
**CorrectAnswer**: (F)
**Video**: Soft Rain | Animated Short Film (2023) - video_url

## II. THEME UNDERSTANDING (TU)

### II.1. PHILOSOPHICAL CONCEPTS

**Task Description**: The themes of the videos are usually about concepts and principles related to philosophy and psychology.
**Example Question**:

> **Question**: What is the overall message that the animation aims to convey?
> **Options**:
> (A) It suggests happiness comes solely from financial achievements.
> (B) The animation emphasizes the need to avoid all responsibilities.
> (C) The animation aims to illustrate the ways to relieve stress.
> (D) It illustrates the mechanical process of water flow.
> (E) The animation encourages saving water to prevent wastage.
> (F) The animation conveys the importance of managing stress through self-care practices.
> (G) The animation highlights achieving success through hard work.
> (H) The animation suggests that ignoring stress leads to happiness.
> (I) The video underlines the significance of collective teamwork.
> (J) It depicts progress and growth through constant work.
> **CorrectAnswer**: (C)
> **Video**: The Stress Bucket - video_url

### II.2. SOCIAL ISSUES

**Task Description**: The theme of the video is usually to reflect some problems existing in today's society and express a strong appeal of the author.
**Example Question**:

> **Question**: What social reality does this video satirize?
> **Options**:
> (A) The rise of environmental awareness in urban settings.
> (B) The video represents the bystander effect in society.
> (C) The economic disparities in urban vs. rural areas.
> (D) The challenges of modern relationship dynamics.
> (E) The impact of fashion trends on daily life.
> (F) The increasing complexity of urban development planning.
> (G) The need for infrastructure improvement and road safety.
> (H) The influence of social media on public behavior.
> (I) The rapid pace of technological advancement in transportation.
> (J) The shift in societal values towards individualism.
> **CorrectAnswer**: (B)
> **Video**: Stone | 1 Minute Short Film | Hot Shot - video_url

### II.3. PERSONAL REFLECTION

**Task Description**: The author hopes that the video will inspire people to reflect on and resonate with things in their lives.
**Example Question**:

> **Question**: What is the core concept that the film aims to convey?
> **Options**:
> (A) Romantic relationships in adolescence.
> (B) The importance of education institutions.
> (C) Overcoming supernatural challenges.
> (D) The dynamics of family disagreements.
> (E) Exploration of technological advancement.
> (F) Not to judge others too quickly.

(G) Journey of a superhero in saving the city.
(H) Inter-species relations on Earth.
(I) Power struggles in political leadership.
(J) Historical recount of a famous personality.
**CorrectAnswer**: (F)
**Video**: Award Winning SHORT FILMS Don't Judge | BATTI Hindi Heart Touching Short Movies | Content Ka Keeda - video_url

## II.4. EVERYDAY TOPICS

**Task Description**: The themes expressed in the videos are usually the sublimation of the insights and themes in daily life, such as praising maternal love, friendship, etc.
**Example Question**:

**Question**: What is implied by the contrast between the scenes around 0:47 and 1:11?
**Options**:
(A) The contrast shows that the mother is indecisive and can't make up her mind in a crisis.
(B) It demonstrates the father's sense of responsibility and bravery, praising paternal love.
(C) The contrast between the beginning and the end conveys a sense of tragedy, criticizing the destruction of the ecological environment by humans.
(D) It shows that the father wants to abandon the child when facing danger.
(E) It shows the bravery of the bird in the background, facing authority head-on, and praises courage.
(F) The mother still protects her child at all costs even in the face of danger, which praises maternal love.
(G) It implies that the father is doing it for self - preservation rather than out of love for the child.
(H) It shows that even when there are many birds, they do not appear very united, and in the face of danger, they become a disorganized mess.
**CorrectAnswer**: (F)
**Video**: Mother 1 minute Sad Emotional Award Winning Iranian Short Film Animation Animated - video_url

## II.5. VIDEO NAMING

**Task Description**: Come up with a suitable title for this video or the core content of the video (dance, etc.). This tests the model's control over the overall content and whether it can get the subtleties of the title like humans.
**Example Question**:

**Question**: "Please come up with a suitable name for this dance.",
**Options**:
(A) The Dance of the Butterfly.",
(B) The Rhythm of the Phoenix.",
(C) The Grace of the Swan.",
(D) The Spirit of the Dragon.",
(E) The Charm of the Peony.",
(F) The Step of the Tiger.",
(G) The Soul of Peacock",
(H) The Beat of the Forest.",
(I) The Leap of the Deer.",
(J) The Spin of the Star.",
(K) The Waltz of the Moon."
**CorrectAnswer**: (G)
**Video**: Yang Liping - The Soul of Peacock - Peacock Dance - Traditional Dance - video_url

## III. EMOTION RECOGNITION (ER)

### III.1. EXPLICIT EMOTION

**Task Description**: Analyze the emotions of the characters in the video. Explicit emotions can usually be directly understood through facial expressions, body movements, etc.
**Example Question**:

> **Question**: Summarize the boy's emotional changes between 6:00 and 7:00.
> **options**:
> (A) Anger - Fear - Surprise and happiness
> (B) Sadness - Excitement - Helplessness
> (C) Disappointment - Let - down - Sorrow
> (D) Loneliness - Isolation - Solitude
> (E) Sadness - Grief - Mourning
> (F) Sadness - Shock - Surprise and happiness
> (G) Disappointment - Astonishment - Stupefaction
> (H) Loneliness - Isolation - Sorrow
> (I) Disappointment - Excitement - Helplessness
> **correctAnswer**: (F)
> **Video**: CGI Animated Short Film: "Crunch" by Gof Animation | CGMeetup - video_url

### III.2. IMPLICIT EMOTION

**Task Description**: Analyze the emotions of characters in the video. Implicit emotions usually need to be analyzed indirectly through the environment, style, etc.
**Example Question**:

> **Question**: What kind of emotional atmosphere does the stage lighting create?
> **options**:
> (A) Solemn and sorrowful atmosphere.
> (B) Neutral and unemotional atmosphere.
> (C) Intense and dramatic atmosphere.
> (D) Joyful and festive atmosphere.
> (E) Sadness and loss.
> (F) Confident and empowering atmosphere.
> (G) Chaotic and confusing atmosphere.
> (H) Calm and serene atmosphere.
> (I) Playful and whimsical atmosphere.
> (J) Romantic and loving atmosphere.
> **CorrectAnswer**: (E)
> **Video**: Stages of Grief- AVANTGARDE SHOW 2023 - video_url

### III.3. META-EMOTION

**Task Description**: This part refers to the high-level emotions in the video, such as the emotions expressed by the author through the video, and the emotions expressed by the entire video.
**Example Question**:

> **Question**: Summarize the meaning of this short film in one word.
> **Options**: [
> (A) Creation
> (B) Transformation
> (C) Stress
> (D) Mutation
> (E) Metamorphosis
> (F) Growth
> (G) Rebirth"

(H) Destruction
(I) Erosion
(J) Development
(K) Isolation
(L) Conversion
**CorrectAnswer**: (C)
**Stress - Shortfilm** - video_url

### III.4. AUDIENCE EMOTION

**Task Description**: Analyze the emotions that viewers are most likely to feel after watching the video. This is more advanced and relatively easy for humans to sense. Including the perception of humor.
**Example Question**:

**Question**: What are the reasons for the high number of views on this video?
**Options**:
(A) The video features a well-known celebrity who has a large fan base, drawing a lot of attention.
(B) The dance style is extremely unique and has never been seen before, sparking curiosity.
(C) People are under a lot of stress and need videos that can help them unwind.
(D) The background music is a popular hit song that many people recognize and enjoy.
(E) The video was released during a major holiday season when people are more likely to watch videos.
(F) The choreography is incredibly complex and impressive, showcasing the dancers' skills.
(G) The video has a strong and inspiring message that resonates with a wide audience.
(H) The video was featured on a popular TV show or news segment, driving more views.
(I) The video was shared by a large number of dance schools and communities, spreading its reach.
(J) The video was part of a viral challenge that encouraged people to share it.
(K) The video has high-quality production values that make it stand out from other content.
**CorrectAnswer**: "(C)
**Satisfying and Relaxing Kinetic Sand ASMR shorts** - video_url

## IV. COMMENT MATCHING (CM)

### IV.1. HUMOROUS

**Task Description**: The video will spark laughter because of certain comments, making the audience feel funny, testing whether the model can match it correctly.
**Example Question**:

**Question**: Based on this video, which of the following comments is likely to make people laugh?
**Options**:
(A) Did he just audition for a water ballet?
(B) How many fish does it take to catch a man?
(C) Is there a Walmart beneath the river?
(D) The fish are holding a grudge, watch out!
(E) Now that's what I call a splash of creativity.
(F) I came for the fishing tips and stayed for the synchronized swimming.
(G) That water has more personality than my neighbor!
(H) I'm starting to think he's part fish.
(I) I think the fish caught him instead.
(J) That's definitely a land fish champion.
(K) That fish will never trust humans again.
**CorrectAnswer**: "(C)",
**He DI Lao** - video_url

## IV.2. THOUGHT-PROVOKING

**Task Description**: Some comments under the video will enhance people's thinking and test whether the model can accurately understand.
**Example Question**:

**Question**: Which of the following statements can better explain the social reality expressed in this animation?
**Options**:
(A) The animation showcases an idealized view of advancement within a corporate ladder.
(B) The depiction highlights the dehumanization and mechanization of individuals in a powerful social system.
(C) It portrays the joy of discovering one's true passions through societal pressures.
(D) The scenes show a man achieving happiness through daily routine.
(E) It represents personal ambition and the drive for success in individual careers.
(F) The animation indicates the triumph of an individual's spirit in the face of adversity.
(G) It reflects the disintegration of traditional family roles.
(H) The animation shows the importance of family support in work-life balance.
(I) It emphasizes the challenge of maintaining personal identity in urban settings.
(J) We are all working for others without realizing it due to our own needs.
(K) The animation illustrates the struggle with contemporary health issues.
**CorrectAnswer**: (J)
**EL EMPLEO** - video_url

## IV.3. TRENDING

**Task Description**: It is relatively difficult to test whether the model can accurately infer and analyze the most popular comments under the video.
**Example Question**:

**Question**: Which of the following comments best summarizes the content conveyed by this film?
**Options**:
(A) Material possessions define one's value.
(B) Selfless acts lead to rewards that surpass material wealth.
(C) Loneliness is a desirable state.
(D) Personal gains are the ultimate goal of helping others.
(E) Isolation is the path to personal growth.
(F) True happiness is found through wealth accumulation.
(G) Success comes from competitive behavior.
(H) Sharing leads to financial prosperity.
(I) He receives what money can't buy.
(J) Adversity breeds stronger individuals.
**CorrectAnswer**: (I)
**Unsung Hero** - video_url

# V. IMPLICIT SYMBOL (IS)

## V.1. CULTURAL SYMBOLS

**Task Description**: Test whether the model can infer and analyze the cultural characteristics hidden under the surface visual elements of the video (such as nationality, festivals, customs, religion, etc.).
**Example Question**:

**Question**: The plaque inscribed with "Dominating Three Continents" that appears in the video is most likely to be found in the architecture of which of the following religions?
**Options**:
(A) Taoism

(B) Shinto
(C) Sikhism
(D) Judaism
(E) Islam
(F) Christianity
(G) Buddhism
(H) Hinduism
(I) Jainism
(J) Zoroastrianism
**CorrectAnswer**: (G)
**[4K] Hangzhou 2024 in the misty rain | West Lake, Lingyin Temple, Night walking in Hefang Street** - video_url

## V.2. ART SYMBOLS

**Task Description**: Test whether the model can infer and analyze the art-related characteristics hidden under the surface visual elements of the video (such as dance style, artistic skills, imitation, etc.).
**Example Question**:

**Question**: What is the shadow that appears in our view at 1:40 imitating?
**Options**:
(A) The shadow is imitating a pole dancer.
(B) The shadow is imitating a person washing a dog.
(C) The shadow is imitating a person brushing their hair.
(D) The shadow is imitating someone playing a violin.
(E) The shadow is imitating two people engaged in a conversation.
(F) The shadow is imitating someone painting a wall.
(G) The shadow is imitating a person feeding a horse.
(H) The shadow is imitating a person washing their car.
(I) The shadow is imitating a dog barking at a person.
(J) The shadow is imitating someone performing a magic trick.
(K) The shadow is imitating a person holding an umbrella.
(L) The shadow is imitating someone walking a large dog.
**CorrectAnswer**: (A)
**LEAKED! Hilarious Shadow Puppets - AGT 2023 Early Release** - video_url

## V.3. OTHER SYMBOLS

**Task Description**: Test whether the model can infer and analyze other special symbols (such as commercial advertisements, etc.) hidden under the surface visual elements of the video.
**Example Question**:

**Question**: "What do you think the chimpanzee that appears multiple times in the film symbolizes?",
**Options**:
(A) The chimpanzee symbolizes chaos and disruption in everyday life. (B) The chimpanzee symbolizes a childhood fear. (C) The chimpanzee symbolizes technology invading personal space. (D) The chimpanzee symbolizes the unpredictability of fate. (E) The chimpanzee symbolizes a glue company. (F) The chimpanzee symbolizes lost opportunities. (G) The chimpanzee symbolizes an obsession with social status. (H) The chimpanzee symbolizes environmental degradation. (I) The chimpanzee symbolizes the desire for freedom. (J) The chimpanzee symbolizes misunderstanding between people. (K) The chimpanzee symbolizes reliability and trust in friendships.
**CorrectAnswer**: (E)
**All Gorilla glue ads** - video_url

### E.2 Explicit Reasoning Tasks

#### I. Causal Reasoning (CAR)

#### I.1. Forward Reasoning

**Task Description**: Forward reasoning can also be understood as the prediction of future events, including prediction of outcomes, prediction of content that has not yet appeared, etc.
**Example Question**:

> **Question**: What is the speculated ending of the film?
> **Options**:
> (A) The movie concludes with an unexpected twist where the flowers reveal a hidden secret.
> (B) The ending is a cliffhanger, leaving the audience uncertain about the characters' fate.
> (C) Her boyfriend passed away due to illness, leaving the girl devastated with grief.
> (D) The film wraps up with a joyous family reunion.
> (E) The film ends with a dramatic breakup as one character leaves with a heavy heart.
> (F) The movie concludes with a comedic mishap involving the flowers.
> (G) The ending shows a tragic farewell as one character moves to a new city.
> (H) The film ends with the revelation of a long-lost sibling.
> (I) The story concludes with the characters embarking on a spontaneous road trip.
> (J) The film ends on a melancholic note, reflecting on lost opportunities.
> (K) The video closes with a heartwarming reconciliation between the main characters after exchanging heartfelt notes and gestures.
> **CorrectAnswer**: (C)
> **For Milo - AWARD WINNING 1 Minute Short film (2020)** - video_url

#### I.2. Backward Reasoning

**Task Description**: Backward reasoning means finding the cause from the effect and inferring the reason why an event occurred.
**Example Question**:

> **Question**: Why was the elderly black man warned by security at the beginning of the film?
> **Options**:
> (A) Mobile phones are not allowed for recording during magic shows.
> (B) He was trying to sell unauthorized merchandise.
> (C) He was recognized as a local celebrity causing disruptions.
> (D) He was accused of stealing a bicycle.
> (E) He was creating loud music disturbing the peace.
> (F) He was believed to have lost his entrance ticket.
> (G) He was inadvertently blocking the pathway.
> (H) He was associated with another person causing trouble nearby.
> (I) He was engaged in card tricks that security found suspicious.
> (J) He was loitering without a purpose.
> **CorrectAnswer**: (A)
> **Now You See Me Official Opening Scene (2013) - Mark Ruffalo, Morgan Freeman Movie HD** - video_url

#### II. Sequential Structure Reasoning (SSR)

#### II.1. Narrative Structure

**Task Description**: Reasoning and analyzing the narrative order of the entire video, including the editing order, such as sequential, flashback, and interpolation.
**Example Question**:

**Question**: What kind of narrative sequence does the film employ?
**Options**:
(A) non-linear flashback sequence, where events are shown out of chronological order, often revealing backstory
(B) parallel overlapping sequences, showing multiple storylines happening simultaneously with some overlap
(C) cyclical narrative structure, repeating events or themes in a circular pattern
(D) linear narrative sequence, following a straightforward progression from beginning to end
(E) random jumps in the timeline, moving unpredictably between different points in time
(F) interwoven thematic structure, weaving together different themes and ideas throughout the story
(G) reverse chronological order, starting with the end and moving backwards in time
(H) fragmented narrative, presenting the story in disjointed or broken segments
(I) begins with a flashback and then proceeds in chronological order
(J) episodic progression, advancing the story through a series of distinct episodes or chapters
(K) multi-perspective narrative, telling the story from multiple characters' points of view
**CorrectAnswer**: (I)
**Identity SHORT FILM (Award Winning Inspirational Short)** - video_url

## II.2. CORE CONNECTING ELEMENTS

**Task Description**: Videos with this type of question usually have a key connecting element that runs through the entire video. It is carefully designed by the producer and tests the model's inductive reasoning of the visual information of the entire video.
**Example Question**:

**Question**: What is the recurring element in the video, summarized in one word?
**Options**:
(A) Pareidolia
(B) Smile
(C) Alarm
(D) Work
(E) Mirror
(F) Mundane
(G) Routine
(H) Suit
(I) Coffee
(J) Sleep
(K) Bedroom
(L) Portrait
**CorrectAnswer**: (B)
**PAREIDOLIA - 1 Minute Short Film | Award Winning** - video_url

## II.3. INFERENCE ON EDITING TECHNIQUES

**Task Description**: These tasks evaluate the models' deep analysis and multimodal reasoning about video editing strategies.
**Example Question**:

**Question**: "Please guess how many videos were needed to record the moment the man punched the punctured water ball at the beginning of the video?",
**Options**:
(A) At least two separate takes would be needed.
(B) At least one single take is needed.
(C) Three separate takes are needed.
(D) Four separate takes are needed.
(E) Each scene can be captured in a single continuous take.

(F) Five separate takes are needed.
(G) Six separate takes are needed.
(H) Eight separate takes are needed.
(I) Ten separate takes are required.
(J) Twenty separate takes are necessary.
(K) At least ten separate takes are needed.
**CorrectAnswer**: (C)
**Playing With Time** - video_url
**Note:** The reasoning and analysis process of this question can refer to this disassembly video .

## II.4. HALLUCINATION

**Task Description**: Evaluate whether the model perceives various types of hallucinations when perceiving video content.
**Example Question**:

**Question**: How many dancers are there in the video?
**Options**:
(A) 0
(B) 1
(C) 2
(D) 3
(E) 4
(F) 5
(G) 6
(H) 7
(I) 8
(J) 9
(K) options before are all false
**CorrectAnswer**: (B)
**Rat dance with falling body parts** - video_url

## III. COUNTERINTUITIVE REASONING (CIR)

## III.1. MAGIC DECONSTRUCTION OR SPECIAL EFFECTS EDITING

**Task Description**: This type of video usually creates some impossible magical effects, but some are magic tricks, and some are editing and special effects, which require deeply reasoning.
**Example Question**:

**Question**: Starting at 4:35, how did the man achieve this magical effect in the magic trick?
**Options**: (A) Sleight of hand technique with a hidden ring, using dexterity to conceal and reveal the ring.
(B) Utilizing a mirror to confuse the audience, creating optical illusions through reflection.
(C) A distraction technique with a smoke bomb, diverting attention with a sudden burst of smoke.
(D) A special ring that retracts into a fake thumb, using a concealed mechanism to make the ring disappear.
(E) Using a magnet hidden in the sleeve, manipulating objects with magnetic force.
(F) A camera trick with video editing, altering footage to create the illusion of magic.
(G) Sleight of hand technique with a hidden string, using a concealed thread to control objects.
(H) The bottle inside the paper bag had already been altered to leave only the outer plastic skin.
(I) Employing a twin assistant to swap the ring, using a look-alike to deceive the audience.
(J) The use of an invisible thread, employing a nearly undetectable line to move objects.
(K) A sound cue to mislead the audience's attention, using noise to distract from the real action.
**CorrectAnswer**: (H)
**Level 1 to 100 Magic Tricks Anyone Can Do** - video_url

## III.2. ARTISTIC TECHNIQUES

**Task Description**: This type of video usually creates some impossible scenes, but it is usually an artistic expression deliberately designed by the author.
**Example Question**:

> **Question**: Why is the shadow on the boy's face illuminated by sunlight at 1:06?
> **Options**: (A) Because the boy moves to a position where a strong light source is directly above him, not related to the girl.
> (B) It's just a coincidence that the angle of the sun changes suddenly at that moment, and has nothing to do with any special meaning.
> (C) The sunlight illuminates the shadow because the cameraman adjusts the lighting equipment to create a better visual effect.
> (D) The girl's appearance brings good luck, and the sunlight representing good fortune clears away the gloom of bad luck in his world.
> (E) This is because the boy has walked into a neighborhood with better weather and climate.
> (F) The sunlight lights up the shadow because there is a hidden light - emitting device in the scene that is turned on at 1:06.
> (G) It's a result of the special lens filter used during filming, which makes the shadow on the boy's face appear to be lit by sunlight.
> (H) Because the boy didn't get hurt after falling and his mood improved, the sunlight is used to represent his improved mood.
> **CorrectAnswer**: (D)
> **CGI Animated Short Film HD "Jinxy Jenkins & Lucky Lou" by Mike Bidinger & Michelle Kwon | CGMeetup** - video_url

## III.3. HUMOR AND EXAGGERATION

**Task Description**: A common technique in humorous videos is to use exaggerated expressions that seem unreasonable, but there are some clues to understand the meaning. This type of question tests the model's ability to reason about exaggerations and unusual techniques.
**Example Question**:

> **Question**: Why does the first half of the scene look sunny but also show rain?
> **Options**:
> (A) It is a sunshower, when rain falls while the sun is shining.
> (B) The character is dreaming of being both wet and warm.
> (C) There are rainclouds directly above while sunlight comes from the side.
> (D) It is snow instead of rain, reflecting the sunlight.
> (E) The effect is caused by morning fog and light refraction.
> (F) It's a visual illusion caused by mist.
> (G) The character moved to a different location quickly.
> (H) A rainbow is forming which intensifies the sunlight.
> (I) Dew drops from trees reflect sunlight.
> (J) There are two unrelated weather animations merged together.
> (K) The man wet the bed, which caused the presence of water in his dream.
> **CorrectAnswer**: (K)
> **It now makes sense** - video_url

## IV. CROSS-MODAL TRANSFER REASONING (CTR)

Evaluate the ability to transfer reasoning from video to text, audio, video or image (for example, video-to-text: the theme of a video may have the same meaning as a famous quote) **Task Description**: Evaluate the ability to transfer reasoning from video to text (for example, the theme of a video may have the same meaning as a famous quote)
**Example Question**:

**Question**: Which of the following proverbs best explains the theme of this short film?
**Options**:
(A) When one door closes, another opens.
(B) Opportunity knocks only once.
(C) Time heals all wounds.
(D) The early bird catches the worm.
(E) Never judge a book by its cover.
(F) All that glitters is not gold.
(G) The grass is always greener on the other side.
(H) Actions speak louder than words.
(I) A stitch in time saves nine.
(J) Beauty is in the eye of the beholder.
(K) Absence makes the heart grow fonder.
(L) A penny saved is a penny earned.
**CorrectAnswer**: (E)
**Video**: Award Winning SHORT FILMS Don't Judge | BATTI Hindi Heart Touching Short Movies | Content Ka Keeda - video_url

## V. VIDEO TYPE AND INTENT (VTI)

### V.1. VIDEO TYPE

**Task Description**: Evaluate the model's ability to analyze video types, such as commercials, science fiction films, comedies, etc.
**Example Question**:

**Question**: What type of video is this most likely to be?
**Options**:
(A) A documentary about airplane technology
(B) Advertisement for an ice-cream
(C) A drama set on an airplane
(D) A comedy film featuring an airline
(E) An in-flight safety demonstration video
(F) A travel vlog featuring aerial views
(G) A science fiction movie on a spaceship
(H) This is an advertisement.
(I) A video tour of an airplane factory
(J) A virtual reality experience of flying
(K) A news segment on turbulence incidents
**CorrectAnswer**: (H)
**Leo Messi vs Kobe Bryant - Legends on Board - Turkish Airlines** - video_url

### V.2. VIDEO INTENT

**Task Description**: Reasoning and analyzing the purpose and production intention of the video (e.g. what kind of product performance is promoted in a commercial advertisement, etc.)
**Example Question**:

**Question**: Which year do you think this video was most likely released?
**Options**:
(A) 2018
(B) 2017
(C) 2016
(D) 2015
(E) 2023
(F) 2019
(G) 2023
(H) 2020

(I) 2014
(J) 2013
**CorrectAnswer**: (H)
**Lockdown | One Minute Short Film Challenge | Film Riot** - video_url

## F    EVALUATION DETAILS

### F.1    BASELINES

The baselines include closed-source models: (1) GPT series: GPT-4o (OpenAI, 2024a), GPT-4o-mini (OpenAI, 2024b), GPT-4.1 (OpenAI, 2025), and GPT-5 (OpenAI, 2025a); (2) Gemini series: Gemini-2.0-flash, Gemini-2.0-flash-thinking-01-21 (Reid et al., 2024), and Gemini-2.5 (flash/pro) (Google DeepMind, 2025); (3) Claude-3-5-Sonnet-20241022 (Anthropic, 2024); (4) o3, o4-mini (OpenAI, 2025b); open-source models: (1) Qwen series: Qwen2.5-VL (7B/72B-Instruct) (Yang et al., 2024a); (2) Gemma series: Gemma-3 (12B/27B) (Kamath et al., 2025); (3) InternVL series: Intern3-VL (8B/38B) (Zhu et al., 2025); (4) LLava series: LLava-Onevision-7B (Li et al., 2024a), Video-LLava-7B (Lin et al., 2023); (5) Phi-4-multimodal-Instruct (Abouelenin et al., 2025); (6) Other video models: Cogvlm2-video-llama3-chat (Hong et al., 2024), NVILA-8B-Video (Liu et al., 2024b). All local experiments are conducted on 4×A100 80GB GPUs.

### F.2    FRAME SELECTION

We followed the official configurations of models that support multi-image input, as well as settings in previous works (Fu et al., 2024; Wang et al., 2024b), to define the number of input frames for each model. Specifically, we fixed the number of frames per model and sampled them evenly across the video duration. We sampled 8 frames for LLaVA-OneVision, Video-LLaVA, and NVILA-8B-Video, Phi-4-multimodal-instruct; 16 frames were sampled for Qwen2.5-VL-7B, Qwen2.5-Omni-7B, CogVLM2-Video-LLaMA3-Chat, InternVL-8B, Gemma-3-it-12B and Gemini-2.0-Flash-Thinking; 32 frames were sampled for Qwen2.5-VL-72B, InternVL-38B, Gemma-3-it-27B, GPT-4o-Mini, o4-mini, o3, GPT-5, Gemini-2.5-Flash-preview and Claude-3.5-Sonnet; 300 frames for GPT-4o. Exceptionally, since Gemini-2.0-Flash supports long video and multimodal context inputs, we sampled one frame per second across each video, with a maximum cap of 512 frames to ensure API stability. For Gemini-2.5-pro, we also tested the official setting of 1fps sampling frequency. Additionally, to enable a fair comparison with Gemini-2.0-Flash-Thinking, we also tested a version of Gemini-2.0-Flash with 16 frames.

## G    FURTHER RESULTS

### G.1    IMPACT OF VIDEO LENGTH ON MODEL PERFORMANCE

To provide a more comprehensive analysis, we evaluated model performance on videos of varying lengths. Following the categorization standards used in prior work such as Video-MME (Fu et al., 2024) and based on the video length distribution of MMR-V, we divided the dataset into three groups: **Short** ($< 2$ minutes), **Medium** (2–6 minutes), and **Long** ($> 6$ minutes). These categories comprise 329, 593, and 335 tasks, respectively. The detailed evaluation results are presented in Table 7. We observe a general trend where the performance of most models decreases as the video length increases. However, it is worth noting that this performance gap becomes **less pronounced for models equipped with long-context capabilities**. For instance, *Gemini-2.0-Flash* (processing 512 frames) and *Gemini-2.5-Pro* (using 1 fps sampling) demonstrate remarkable stability across all duration categories.

### G.2    BENEFIT OF AUDIO INPUT

In this section, we illustrate through several examples how incorporating audio as an additional modality provides extra reasoning cues that enhance video reasoning. In the following examples, green text highlights the model's analysis of audio cues, which contributes to reaching the final answer.

Table 7: Evaluation results (%) across different video length. **Bold** and underlined values indicate the best performance among proprietary and open-source models, respectively.

| Model | Overall | Video Length | | |
| --- | --- | --- | --- | --- |
| | | Short Video | Medium Video | Long Video |
| *Open-source models* | | | | |
| LLaVA-Video | 18.4 | 20.5 | 16.7 | 10.4 |
| NVILA-8B-Video | 25.5 | 27.1 | 25.9 | 22.6 |
| Phi-4-multimodal-instruct | 26.7 | 30.1 | 24.5 | 27.2 |
| Cogvlm2-video-llama3 | 25.6 | 27.4 | 25.9 | 22.6 |
| Qwen2.5-VL-7B | 30.1 | 34.7 | 28.0 | 29.3 |
| InternVL3-8B | 33.6 | 34.9 | 33.0 | 32.8 |
| Gemma-3-12b-it | 34.0 | 35.7 | 33.5 | 31.1 |
| InternVL2.5-38B | 39.9 | 43.9 | 39.1 | 38.2 |
| Qwen2.5-VL-72B | 39.1 | 43.8 | 40.4 | 34.5 |
| Gemma-3-27b-it | 42.0 | 43.8 | 42.0 | 40.3 |
| *Proprietary models* | | | | |
| GPT-4o-mini-2024-07-18 | 34.8 | 39.2 | 34.1 | 31.6 |
| Gemini-2.0-Flash (16 frames) | 42.6 | 46.2 | 42.3 | 39.7 |
| Claude-3-5-Sonnet-20241022 | 43.3 | 47.7 | 43.5 | 38.5 |
| Gemini-2.0-Flash-thinking | 45.0 | 48.9 | 45.0 | 40.9 |
| GPT-4.1-2025-04-14 | 46.6 | 49.9 | 46.2 | 44.2 |
| Gemini-2.0-Flash (512 frames) | 48.8 | 48.9 | 48.6 | 49.0 |
| Gemini-2.5-Flash | 51.2 | 54.1 | 51.9 | 47.2 |
| o4-mini-2025-04-16 | 52.5 | 57.1 | 53.0 | 47.2 |
| GPT-4o-2024-11-20 | 52.8 | 57.4 | 53.1 | 48.1 |
| o3-2025-04-16 | 59.1 | 62.9 | 59.7 | 54.3 |
| GPT-5-2025-08-07 | 60.9 | 62.3 | 61.6 | 56.2 |
| Gemini-2.5-pro (1fps) | **64.3** | **64.7** | **64.1** | **63.9** |

In the first example (Figure 10), audio reveals that the library is actually noisy. When the protagonist puts on headphones, the video turns silent. Without audio, the model might mistakenly assume the library is quiet based solely on visuals. The original video link is: Spec commercial - SONY.

In the second example (Figure 11) the audio can better understand that 'parachute' and 'pair of shoes' are homophones. The original video link is: Hasta la vista.

Moreover, audio can supply cues from other perspectives. For instance, background music can convey the overall mood of a video, while magic performance videos often include voice narration.

## Benefit of Additional Audio Input

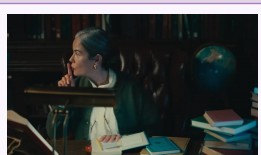 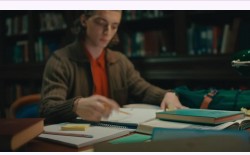 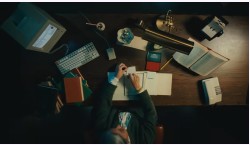 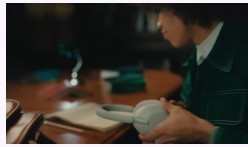

**Question:** Which of the following statements can better summarize the core message of this video?

**Options:**

(A) The video emphasizes the importance of athletic competition.
(B) It portrays a medical laboratory's dedication to scientific experiments.
(C) The primary focus of the video is the pursuit of knowledge in a quiet library setting.
(D) It depicts a cafe's lively social interactions as its core message.
(E) The main theme is the artistic creation in a studio space.
(F) The main theme is the chaos and bustle of a city?
(G) The video is centered around teamwork in a corporate office.
(H) It captures the essence of outdoor adventure and exploration.
(I) It highlights a museum's exploration and discovery process.
(J) Find your focus. ✓

**Gemini-2.0-Flash-with-audio:** The user wants to identify the core message of the video. I need to analyze the video content and compare it to the given options. The video takes place entirely in a library. People are shown studying. There are several instances where the librarian shushes people, and sounds of distraction are highlighted (like chewing gum or flipping pages loudly), which are then silenced. The last frame has text, but it's not visible in the provided images. Let's evaluate the options: ...\n\n(C) 'The primary focus of the video is the pursuit of knowledge in a quiet library setting.' - This fits the visuals of the library, but the sound in the first half of the video is not quiet - Incorrect. ... (J) 'Find your focus.' - This option directly captures the recurring theme of individuals trying to concentrate amidst distractions in a study environment. The cues (shushing, noise being amplified and then silenced, people trying to study) strongly support this. \n\nFinal Answer: [[J]]

**Gemini-2.0-Flash-without-audio:** In the video, students are studying and doing activities in the library. The whole library environment looks very quiet. Some people are looking through materials, some are looking for books they need, and a librarian is using gestures to maintain a quiet environment. Some people are wearing headphones and concentrating on studying. Let's analyze the options: ...(C) 'The primary focus of the video is the pursuit of knowledge in a quiet library setting.' This aligns well with the visual of a library, people studying, and the emphasis on quiet. This is correct. ... \nChoice (C) is more appropriate because the scene is a quiet library and the protagonist seems to be studying.\n\nFinal Answer: [[C]]:**\n\n

Figure 10: Benefit of Additional Audio Input. The green text highlights the analysis of audio cues.

Figure 11: Benefit of Additional Audio Input. The green text highlights the analysis of audio cues.

## H  DETAILS OF COT ANALYSIS EXPERIMENT

This section presents the CoT analysis experiments discussed in Section 4.5. We use a representative model CoT to illustrate 4 categories of analysis in Figure 12. Specifically, Text Analysis refers to the examination of textual information such as the question and options; Video Analysis focuses on the content of the video; Question Frame targets the specific frame referenced in the question—for instance, the frame where the magician controls two flames; and Other Frame pertains to frames outside the scope of the question. In Figure 12, yellow, red, and blue represent text, question frame, and other frame analysis respectively. Red, blue, and green all represent video analysis.

Table 8: CoT analysis prompt.

---

**Prompt for CoT Annotation**

You will be given a model's textual reply to a video-based question along with the video frames. Your task is to determine four boolean labels for each chunk of the reply:
1. 'other frame desc': Does this chunk describe visual information from frames other than question frame?
2. 'question frame desc': Does this chunk correctly describe visual information from the question frame specified in the question?
3. 'video analysis': Does this chunk perform analysis of the video content?
4. 'text analysis': Does this chunk perform analysis of the text (e.g., question text, options) rather than visual content
- The question frame refers to the specific frame(s) referenced by the question prompt.
- Other-frame descriptions are visual details not present in the question frame but from other frames.
- Video analysis includes describing trends, motions, or visual inference beyond plain description.
- Text analysis includes reasoning over question text, options, or external text context.
Respond strictly in JSON:
{
"other frame desc": true or false,
"question frame desc": true or false,
"video analysis": true or false,
"text analysis": true or false
}
Question:  {question}

Reply Chunk:  {chunk}

Whole CoT Reply:  {CoT}

---

## CoT Analysis Example

**Question:** How does the man at the beginning of the video pick up and casually control the flame on the lighter?

**Options:**
(A) He used a holographic projector to simulate the flame.
(B) He used a special flame-retardant chemical on his hand to create the illusion.
(C) He possessed an innate immunity to fire.
(D) He practiced yoga meditation to withstand any flame heat.
(E) A quick extinguishing spray was applied that halted the flame.
(F) He surrounded the flame with an invisible film.
(G) He mastered the art of fire manipulation.
(H) The flame was made of non-flammable gas.
(I) He applied a hidden cooling technology under his sleeve.
(J) The flame was actually an LED light.
(K) A hidden lighter in his hand, a sleight of hand trick.

My thinking process to arrive at the answer is as follows:\n\n1. **Understand the Question:** The question is specifically asking about *how* the man at the beginning of the video picks up and *casually controls* the flame on a lighter. The key words are \"how\" and \"casually controls,\" implying a method or technique rather than a supernatural ability or complex technology.\n\n2. **Analyze the Video** I noticed that the \u201cfloating\u201d flame always stayed just above his palm in a way that perfectly hid the source, and that when he distracted us he slipped a small butane lighter into position. In the close\u2010up reveal later, you can see how the second lighter remains concealed until he produces the flame on his fingers. It\u2019s all done by classic palm concealment and sleight of hand with a hidden lighter. \n\n2. **Analyze the Options:** I reviewed each option to see which one is most plausible and aligns with the context of a magic trick video. I considered the practicality and likelihood of each explanation.\n\n  *  **(A) Holographic Projector, (J) LED Light, (H) Non-flammable gas:** These options suggest the flame isn't real. While possible for illusions, they are less likely for a *casual* control and might be visually detectable as fake in a video focusing on close-up magic.\n\n  *  **(C) Innate immunity to fire, (D) Yoga meditation, (G) Mastered fire manipulation:** These options are fantastical or mystical and don't explain a practical method.  They are less likely to be the intended explanation in a video about magic *tricks*.\n\n  4. **Prioritize Simplicity and Plausibility:** In the context of magic tricks, simpler explanations are often better. Option (K) - sleight of hand - is the simplest and most direct explanation that fits the description of \"casual control.\"  It's a well-known method for creating illusions with small objects like lighters.\n\n5. **Eliminate Less Likely Options:** I eliminated the options that were:\n  *  Too complex or technologically advanced (A, I).\n  *  Unrealistic or fantastical (C, D, G, H).\n  *  Less direct or slightly less probable for \"casual control\" in a close-up magic context compared to sleight of hand (B, E, F, J).\n\n6. **Select the Best Fit:** Option (K) - \"A hidden lighter in his hand, a sleight of hand trick\" - emerges as the most likely and best-fitting answer because it is:\n  *  A common and effective magic technique.\n  *  A simple and direct explanation.\n  *  Consistent with the idea of \"casual control.\"\n  *  Plausible in the context of a magic trick video.\n\n\nTherefore, based on the question, the options, and the likely context of a magic trick video, option (K) is the most accurate answer.\n\n**Final Answer:** [[K]]

Figure 12: CoT example of experiments in Section 4.5. Yellow, red, and blue represent text, question frame, and other frame analysis respectively. Red, blue, and green all represent video analysis.

## I  EVALUATION PROMPT

We evaluated two settings in the main experiment, zero shot and zero shot + CoT. The prompts used are as follows.

Table 9: Evaluation prompt for the Zero-Shot Setting.

---

**Prompt for Zero-Shot Setting**

[[INSTRUCTION]]
Please select the best answer to the following multiple-choice question based on the video.
Only one option is the most accurate answer in relation to the question and the video.
What is the correct answer to this question
{Question}
Options:
{Options}
[[END OF INSTRUCTION]]
[[OUTPUT FORMAT]]
Format your answer as follows:
Please directly output the answer letter without thinking and explanation.
If the correct option letters (A, B, C, D... ) for the multiple-choice question is X, give the final correct option number in the following format: "[[X]]"
[[END OF OUTPUT FORMAT]]

---

Table 10: Evaluation prompt for the Zero-Shot + CoT Setting.

---

**Prompt for Zero-Shot + CoT Setting**

[[INSTRUCTION]]
Please select the best answer to the following multiple-choice question based on the video.
Only one option is the most accurate answer in relation to the question and the video.
What is the correct answer to this Question:
{Question}
Options:
{Options}
Let's think step by step.
[[END OF INSTRUCTION]]
[[OUTPUT FORMAT]]
Format your answer as follows:
Your thinking process.
If the correct option letters (A, B, C, D... ) for the multiple-choice question is X, give the final correct option number in the following format: "[[X]]"
[[END OF OUTPUT FORMAT]]

---

## J  CASE STUDY

In this section, we present reasoning processes and results from selected models on the MMR-V benchmark. Through these case studies, we aim to better illustrate the current shortcomings of models in multimodal reasoning tasks and provide insights that may inspire future research and advancements in this area.

Firstly, there is a comparison between a good CoT and a poor CoT in 13. Yellow highlights indicate text-based analysis, while green highlights denote video-based analysis. As shown, the upper two CoTs engage in deep multimodal reasoning grounded in the video, ultimately reaching the correct answer. In contrast, the bottom CoT relies mainly on textual reasoning with only shallow video perception, leading to an incorrect outcome.

**Comparative Case Study of MCoTs**

**Question:** What does the object being chased by the people refer to?

**Options:**

(A) Difficulties in life.
(B) His fully automatic house that can walk autonomously.
(C) The house that this person is saving money to buy.
(D) The old house that the family decided to renovate.
(E) The villa that the rich man bought as a vacation home.
(F) The mansion that the celebrity owns in the city center.
(G) The cottage that the couple wants to rent for their honeymoon.
(H) The enchanted house.

**GPT-5-CoT:** The animation depicts people in relentless pursuit of a walking house—quite literally chasing after a "home." Throughout the film, the house slips away through rain and snow, over seas and across canyons, never within reach. In the final scene, it stands on the opposite side of a vast divide, facing a crowd of people separated from a cluster of buildings. The imagery serves as a metaphor for the unending struggle to afford and finally secure a home. The pursued house symbolizes the one people dream of saving enough to buy.\n\n [[C]]

**o3-CoT:** The animated short shows a small house with legs and arms sprinting away, while more and more people join in the chase. Nothing in the story suggests that anyone already owns the house, or that it is just a futuristic "automatic" home. Instead, it comes across as something everyone desperately wants but cannot yet have. The running house clearly represents the dream of owning a home that ordinary people are always pursuing—the place they hope to buy someday. In this sense, the object being chased is best understood as the house that a person, and by extension everyone else, is saving money to purchase.\n\n [[C]]

**Gemini-2.0-Flash-Thinking-CoT:** The**Thinking Process:**\n\n1. **Identify the question:** The question asks about the meaning of \"the object being chased by the people\".\n\n2. **Watch the video and observe the object:** The video clearly depicts a house with legs walking and being chased by people.\n\n3. **Analyze the options and compare them to the video:**\n    * **(A) Difficulties in life:** While metaphorical interpretations are possible, the video literally shows a house being chased. This option is too abstract and doesn't directly relate to the visual content.\n    * **(B) His fully automatic house that can walk autonomously:** This option directly describes what is seen in the video \u2013 a house that can walk on its own. This is a strong contender.\n    * **(C) The house that this person is saving money to buy:** There is no visual or narrative element in the short video clip about saving money or buying a house. This option is not supported by the video.\n    ......* **(F) The mansion that the celebrity owns in the city center:** No celebrity, mansion, or city center is depicted. This option is irrelevant.\n    * **(G) The cottage that the couple wants to rent for their honeymoon:** There is no mention of a couple, honeymoon, or cottage rental in the video clip. This option is irrelevant.\n    * **(H) The enchanted house:** While the walking house could be considered \"enchanted,\" option (B) is more precise and descriptive of the *mechanism* of the house's unusual behavior (autonomously walking). \"Enchanted\" is a more general and less accurate description compared to \"fully automatic house that can walk autonomously.\"\n\n4. **Select the best option:** Option (B) is the most direct, accurate, and visually supported answer based on the video. It literally describes the object being chased.\n\n**Final Answer:**\n\n[[B]]

Figure 13: A comparison of CoTs from two models on the same task. Yellow and green indicate text and video analysis, respectively. As shown, GPT-5 and o3's reasoning paradigm demonstrates a deeper analysis of the video content.

## Lack of Visual Reasoning

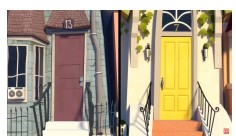 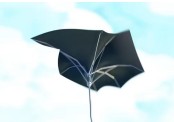 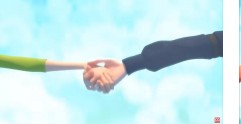 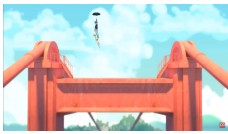

**Question**: Why did the broken umbrella in the boy's hand, damaged by the wind, suddenly get fixed while the boy and the girl were in the air?

**Options**:
(A) Because the boy initially used too much force, causing the umbrella to be blown apart, and it was repaired after he adjusted his effort.
(B) Because the boy held the girl's hand, gained good luck, and as a result, the umbrella was restored.
(C) Because the wind in the air happened to die down, the umbrella returned to normal.
(D) Because the girl held the boy's hand, making him feel extremely happy, which in turn allowed the umbrella to be repaired smoothly.
(E) It's just a visual illusion in the video, and the umbrella wasn't actually blown apart in the first place.
(F) The umbrella was designed to be a self - repairing model, and it activated its repair function automatically.
(G) It was a pre - arranged magic trick, and the umbrella was never truly damaged.
(H) The video was reversed during that part, making it look like the broken umbrella was getting fixed.
(I) Aerodynamics, changes in wind strength and direction contributed to the umbrella being repaired.
(J) It was a computer - generated special effect added during post - production to create a magical look."

**Video Type: Animation**
**Task Type L1: Explicit Reasoning.**
**Task Type L2: Counterintuitive Reasoning**
**Task Type L3: Artistic Techniques**

**Gemini-2.0-Flash-Thinking**: [[Thinking Process]]\nThe user is asking for the best answer to the question \"Why did the broken umbrella in the boy's hand, damaged by the wind, suddenly get fixed while the boy and the girl were in the air?\" based on the provided video and options.\n\nAfter watching the video, it's clear that:\n1. The umbrella is initially broken by the wind on the ground.\n2. The boy and girl are lifted into the air by pianos attached to balloons.\n3. While in the air, the broken umbrella appears to become fixed.\n\nNow let's evaluate each option against what's observed in the video and consider the most logical explanation within the context of the animation:\n\n* ...... **(B) Because the boy held the girl's hand, gained good luck, and as a result, the umbrella was restored.** - This is a whimsical, magical explanation. While the video has a fantastical element, \"good luck\" causing physical repair is not the most direct or likely intended answer.\n\n ......(I) Aerodynamics, changes in wind strength and direction contributed to the umbrella being repaired.** - This is a more detailed and scientifically grounded explanation. As they ascend, changes in wind strength, direction, and the aerodynamic forces acting on the umbrella could allow it to reshape itself. This is very plausible.\n\n* - While the video *is* computer-generated animation, this option describes *how* it's shown, not *why* it happens within the video's narrative. It doesn't explain the in-universe reason for the repair.\n\n\nComparing options (C) and (I), option (I) is more comprehensive as it includes \"aerodynamics\" in addition to changes in wind, which is a more complete physics-based explanation relevant to an umbrella in motion in the air. Option (C) is a simplified version of (I).\n\nOption (I) is the most scientifically plausible and best explanation from the given options for why the umbrella appears to get fixed within the context of the animation.\n\n[[I]]

**Human Error Annotation**:The model only analyzes visual information in the first small orange part of the analysis, and only analyzes the frame corresponding to the question. After that, most of the CoT is doing text analysis (comparing the pros and cons of each option)

Figure 14: Error Case: Lack of Visual Reasoning.

## Lack of Visual Reasoning

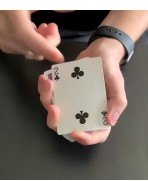 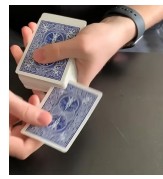 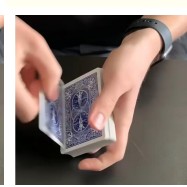 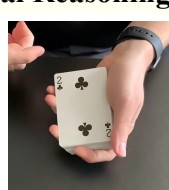 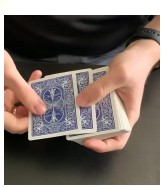 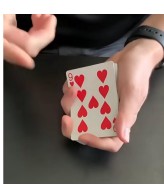

**Question**: How did the man manage to insert the top card into the deck and then reveal the top card again, only to find it back in its original position?

**Options**:
"(A) The card deck was equipped with a hidden compartment.",
(B) He had two identical cards, and he switched them.",
(C) The card deck was magnetic, allowing precise control.",
(D) The cards were fitted with a special adhesive.",
(E) The man used camera tricks to manipulate what was shown.",
(F) He used invisible threads to control the card's position.",
(G) The cards had sensors that automatically rearranged them.",
(H) He inserted the card in a fake cut, not the real deck.",
(I) The man used sleight of hand to secretly return the card to the top.",
(J) The deck was rigged with springs to pop the card back up.",
(K) When showed the cards, he actually turned over two cards."

**Video Type: TV**
**Task Type L1: Explicit Reasoning.**
**Task Type L2: Counterintuitive Reasoning**
**Task Type L3: Magic Deconstruction**

**Gemini-2.0-Flash-Thinking**: My thinking process to arrive at the correct answer, option (I), is as follows:\n\n1. **Understand the Question:** The question is asking for the *method* used by the man to make the top card reappear on top of the deck after being inserted into the deck. This is a classic magic trick scenario.\n\n2. **Watch the Video Carefully:** I watched the video multiple times, paying close attention to the man's hand movements, especially when he \"inserts\" the card and then reveals it again.\n\n3. **Analyze the Actions in the Video:**\n    * He shows the top card (2 of Clubs).\n    * He *appears* to insert the 2 of Clubs into the deck.\n    * He immediately reveals the top card again, and it's still the 2 of Clubs.**\n\n4. **Evaluate Each Option Against the Video and Common Magic Principles:** I went through each option, considering if it's plausible based on what I saw and general knowledge of magic tricks.\n\n    * **(A) Hidden Compartment:** Possible in theory, but decks with hidden compartments are less common for simple card tricks. It's also not visually indicated in the video. Less likely for this type of quick trick.\n ......***(I) Sleight of Hand:** This is the most probable and general explanation for card magic. Sleight of hand encompasses various techniques to manipulate cards secretly. This option directly addresses the core of card magic and fits the observed actions. *This is the most likely answer.*\n    * **(J) Rigged Springs:** Mechanically complex and unnecessary. Springs would be hard to conceal and control for this specific effect. Unlikely.\n    * **(K) Turned Over Two Cards:** Doesn't explain how the card returns to the top after insertion. Also, visually, it looks like he's only turning over one card.  Unlikely and irrelevant to the question.\n\n5. **Select the Best Fit:** After evaluating all options, option (I) \"The man used sleight of hand to secretly return the card to the top\" stands out as the most general, accurate, and likely explanation.  It encompasses the core principle of card magic and directly addresses the question of how the card returned to the top. Option (B) is a *type* of sleight of hand, but (I) is broader and more encompassing as a general explanation.  Option (H) is also related to sleight of hand but option (I) is more direct to the question.\n\n6. **Final Answer:** Therefore, option (I) is the best answer.\n\n[[I]]

**Human Error Annotation**: The model only analyzes the actions in the frames adjacent to the question frame in the video. In fact, at the end of the video, the magician explains that he turned over two cards at the beginning and showed the second card. The **model did not observe the evidence frame of the decryption part through visual reasoning**.

Figure 15: Error Case: Lack of Visual Reasoning.

## Implicit Misinterpretation

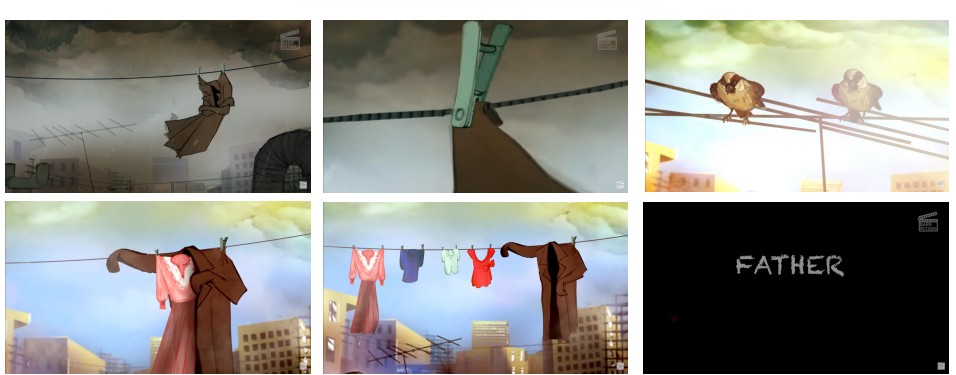

**Question**: What is the main message that this video wants to convey?

**Options**:
(A) The video is mainly about showing how easy it is to solve family problems without much effort.
(B) It praises the mother's strength and kindness, and how she teaches her children to interact with others.
(C) It praises the sacrifice of the brown coat, ensuring that the other clothes do not get wet.
(D) It aims to illustrate that family members should always rely on external help when in trouble.
(E) The video tries to convey that family is not important compared to personal success.
(F) The video praises the father's sense of responsibility and commitment.
(G) The main point is that fathers always make mistakes and children have to fix them.
(H) It aims to show that fathers should focus more on their careers than on family.

**Video Type: Animation**
**Task Type L1: Implicit Reasoning.**
**Task Type L2: Theme Understanding**
**Task Type L3: Everyday Topics**

**GPT-4o**: [Analyze the best option for question] \nThe video showcases a brown coat protecting other clothes from rain on a clothesline. **The actions and symbolism suggest the brown coat's sacrifice, highlighting the theme of protection or sacrifice for others**, akin to a familial role. \n\n[Justification for your final choice based on the thinking process.]\nThrough the coat's protective measures, the video illustrates the theme of sacrifice by the brown coat to keep other clothes dry. This aligns closely with option C, where the sacrifice of the brown coat ensures that the other clothes do not get wet.\n\nGive the final correct option number in the following format: \"[[C]]\"

**Human Error Annotation**: The model only analyzes the actions in the frames adjacent to the question frame in the video. In fact, at the end of the video, the magician explains that he turned over two cards at the beginning and showed the second card. The **model did not observe the evidence frame of the decryption part through visual reasoning**.

Figure 16: Error Case: Implicit Misinterpretation.

## Reasoning Error

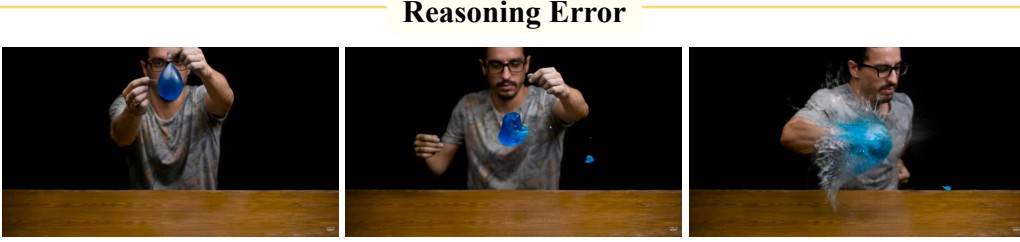

**Question**: Please guess how many videos were needed to record the moment the man punched the punctured water ball at the beginning of the video?

**Options**:
(A) At least two separate takes would be needed.
(B) At least one single take is needed.
(C) Three separate takes are needed.
(D) Four separate takes are needed.
(E) Each scene can be captured in a single continuous take.
(F) Five separate takes are needed.
(G) Six separate takes are needed.
(H) Eight separate takes are needed.
(I) Ten separate takes are required.
(J) Twenty separate takes are necessary.
(K) At least ten separate takes are needed.

**Video Type: TV**
**Task Type L1: Explicit Reasoning.**
**Task Type L2: Counterintuitive Reasoning**
**Task Type L3: Magic Deconstruction**

**Gemini-2.0-Flash-Thinking**: [[Thinking Process]]\nThe question asks to guess the number of videos needed to record the water balloon punch scene at the beginning of the video. This is a question about the *process* of video making, specifically related to achieving a good take.\n\nBased on common video production knowledge and the nature of capturing a dynamic event like a water balloon bursting in slow motion, it is highly unlikely that such a scene would be captured perfectly in a single take. **There are many variables such as timing of the punch, the balloon's burst pattern, camera focus, and framing.**\n\nConsidering the options:\n* **Options suggesting a single take (B) and (E)** are less likely. Capturing a dynamic and potentially messy action like a water balloon burst perfectly on the first try is improbable, especially for a visually appealing slow-motion shot.\n* **Options suggesting multiple takes (A), (C), (D), (F), (G), (H), (I), (J), (K)** are more reasonable. Filmmaking often involves multiple takes to ensure the best possible shot.\n* **Option (A) 'At least two separate takes would be needed.'** **This is the most conservative option indicating that more than one take is probable. It's a very reasonable minimum guess.**\n* **Options with higher numbers of takes (C), (D), (F), (G), (H), (I), (J), (K)** are also possible, but they are more specific guesses. Without explicit information from the video, these are harder to justify as the *best* guess compared to the minimal \"at least two\".\n* **Option (K) 'At least ten separate takes are needed.'** and **(J) 'Twenty separate takes are necessary.'** are likely overestimations for a relatively simple shot like punching a water balloon. While possible, they are less probable as the *best* guess without more information.\n\nGiven that we are asked to *guess* and choose the *best* answer, the most reasonable and safe guess, based on general video production practices and the nature of the scene, is that **at least two takes would be needed**. This accounts for the likelihood of needing to adjust something or re-perform the action for a better result.\n\nTherefore, Option (A) is the most accurate and reasonable answer from the provided options.\n\n[[Correct Option]]\n[[A]]

**Human Error Annotation**: The model accurately found the frame corresponding to the problem and **perceived the evidence frame required for reasoning, so it is not a Lack of Visual Reasoning. However, the model made mistakes in the orange part during reasoning.**

Figure 17: Error Case: Reasoning Error. The reasoning and analysis process of this example can refer to the disassembly video.

## Correct Case

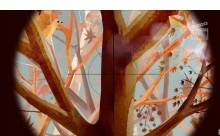 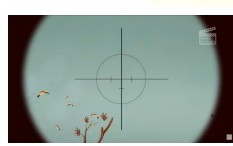 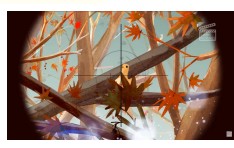 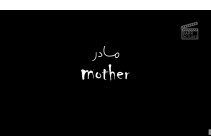

**Question**: The entire video is viewed through a black circle. What is this?

**Options**:
(A) It's a circular hole in a piece of paper the viewer is looking through.
(B) It's a decorative frame added for aesthetic purposes.
(C) It is the scope of a gun.
(D) It is the view through a telescope.
(E) It is a special pair of observation glasses used to observe the behavior of wild animals.
(F) It is an astronomical telescope.
(G) It is the lens of a telephoto DSLR camera.
(H) It is the view through a kaleidoscope.
(I) It's the lens of a vintage camera.
(J) It's a magic hoop that gives special vision.

**Video Type: Animation**
**Task Type L1: Explicit Reasoning.**
**Task Type L2: Causal Reasoning**
**Task Type L3": Backward Reasoning**

**claude-3-5-sonnet**: Let me think about this step by step:\n\n**1) Looking at the video, I notice the view is consistently shown through a circular frame with crosshairs in it.\n\n2) The crosshairs (reticle) are a key identifying feature - they have the characteristic \"+\" shape with a circle.\n\n3) Throughout the video, the view is used to target/aim at birds in the trees.\n\n4) The combination of the circular view, crosshairs, and targeting function strongly indicates this is a scope used for aiming.\n\n**5) Among the options, C mentions \"the scope of a gun\" - this matches perfectly with all the visual evidence.\n\n6) While other options like telescopes or cameras might have circular views, they typically don't include targeting crosshairs like we see here.\n\n7) The clear targeting/hunting context and reticle design confirms this is specifically a gun scope rather than any other optical device.\n\n[[C]]

**Human Analysis**:This is a relatively good answer. The model's thinking process before answering the question is mainly centered around the visual information of the video, rather than the entire CoT being spent analyzing the textual reasoning of the options.

Figure 18: Correct Case.

