# OpenReview forum: "MMR-V: What's Left Unsaid? A Benchmark for Multimodal Deep Reasoning in Videos"
_ICLR.cc/2026/Conference — ICLR 2026 Poster_

### Official Review · Reviewer_NiAg · 2025-10-16

**Soundness:** 3
**Presentation:** 2
**Contribution:** 3
**Rating:** 4
**Confidence:** 4

**Summary:**

This paper introduces MMR-V, a new benchmark designed to evaluate the deep multimodal reasoning capabilities of large language models in the context of videos. The authors argue that existing video benchmarks primarily test perceptual understanding over short temporal windows, failing to assess more complex reasoning. MMR-V addresses this gap by featuring tasks that require long-range, multi-frame reasoning to infer information that is not directly visible. A key contribution is the categorization of tasks into "implicit reasoning" (understanding subtext, metaphors, themes) and "explicit reasoning" (logical deduction from perceivable evidence). The benchmark consists of 317 videos and 1,257 manually annotated multiple-choice questions with carefully designed distractors to challenge current models. The experimental results, covering a wide range of state-of-the-art models, demonstrate that even top-performing models like Gemini-2.5-pro struggle significantly, revealing a substantial gap between model and human performance and highlighting the limitations of current reasoning-enhancement techniques like CoT.

**Strengths:**

1、The paper introduces a benchmark that addresses a critical gap in video understanding evaluation. It introduces a novel task structure focused on deep, multi-frame reasoning. This requires models to synthesize information across long temporal spans and infer hidden context, a much-needed step up in complexity that better reflects real-world reasoning challenges.
2、The quality and rigor of the benchmark construction are a major strength. The reliance on manual annotation, guided by real-world user consensus, ensures high-quality, nuanced data that is difficult to create automatically. Furthermore, the innovative strategy of using model-generated errors to create plausible and challenging distractors is a key feature that effectively tests the robustness of models and mitigates the risk of them relying on superficial cues.
3、The paper provides a clear, comprehensive, and valuable evaluation of a wide range of state-of-the-art models. This extensive testing not only establishes a strong and reliable baseline for future research but also offers insightful analysis into the current limitations of MLLMs in video reasoning. The results clearly highlight the significant performance gap between models and humans, underscoring the benchmark's importance and difficulty.

**Weaknesses:**

(1) The benchmark’s 317 videos primarily cover six categories but underrepresent real-world genres like mystery, puzzle-solving, or gaming (acknowledged in Section B as a limitation). Additionally, most videos are in English, with only a small proportion in other languages, restricting MMR-V’s applicability to multilingual video reasoning tasks.
(2) While the paper notes that models perform better on implicit tasks (+7.9% average gain) due to dispersed visual cues, it does not explore why explicit tasks are harder. For example, whether it is due to fewer evidence frames or stricter logical requirements. This limits insights into how to improve model performance on specific reasoning types.
(3) While the paper tests frame count impact (Figure 4) for Gemini-2.0-Flash, it does not report efficiency metrics (e.g., inference time, memory usage) for models processing long videos. This is relevant for real-world deployment, as long-video reasoning often requires balancing accuracy and efficiency.

**Questions:**

1、The paper acknowledges that video categories like mystery and gaming are underrepresented. Could you elaborate on why these categories were excluded (e.g., difficulty in annotating reasoning tasks, lack of popular videos) and whether there are plans to expand MMR-V to include them in future updates?
2、For implicit reasoning tasks, the paper attributes models’ better performance to "abundant visual cues dispersed throughout the video." Have you explored whether models’ pre-trained world knowledge (e.g., understanding that "room number 7 symbolizes good luck") also contributes to this performance gap?
3、The human experiment uses 200 tasks (100 correct/100 incorrect for GPT-4o). Were these tasks balanced across implicit/explicit reasoning and video categories? If not, could this imbalance affect the measured human-model gap?

---

> ### Author Response · Authors · 2025-11-19
> **Rebuttal 1**
>
> Reviewer 4 NiAg,
> Thank you for your careful and insightful review. We are greatly encouraged by your strong support and are particularly grateful for your detailed acknowledgment of our key strengths: the novelty of our deep reasoning benchmark, the high quality and rigor of its construction, and the comprehensive evaluation that provides valuable insights into current model limitations. We will now address the points you raised and clarify some misunderstandings.
>
> ## **Weakness 1**
>
> > The benchmark’s 317 videos primarily cover six categories but underrepresent real-world genres like mystery, puzzle-solving, or gaming (acknowledged in Section B as a limitation). Additionally, most videos are in English, with only a small proportion in other languages, restricting MMR-V’s applicability to multilingual video reasoning tasks.
>
> First, as you mentioned, these two points are from our **Limitations section**, and we consider these to be **valuable ideas for future extensions of MMR-V**. They are meant to guide subsequent work and inspire other research. They **do not affect the core contribution or data quality of the current version of MMR-V** and are only presented as potential directions for expansion. Regarding the issues you raised, we would like to make the following clarifications.
> - **Regarding the issue of video categories**, we want to clarify the following:
>   - Our choice of six primary categories was informed by previous influential video benchmarks (e.g., Video-MME[1], MLVU[2]). These six categories are **coarse-grained classification**. In Figure 9 of Appendix D, we expand on these six categories, outlining **24 fine-grained subcategories**, which already provide comprehensive coverage of common, high-quality videos in the real world. In fact, the "mystery" genre mentioned in the Limitations belongs to the "**Short Film**" subcategory. The reason it is not extensively covered is that excellent detective-style videos are usually found in longer films, which often have **copyright issues**. Therefore, we only included some detective videos from short films. As for the "puzzle-solving" and "gaming" types we mentioned, we are mainly referring to **synthetic game videos in game environment**. These typically lack high popularity and do not meet our criteria for **real-world authenticity**, which would be in violation of the principles we established in Chapter 3.
>   - MMR-V already covers approximately **24 subcategories, demonstrating good diversity and a strong representation of various real-world videos**. Also, due to **resource and cost limitations**, it is difficult for us to **exhaustively include every type** of video that appears in the real world.
> - **Regarding the issue of language variety**, we would like to make the following clarifications:
>   - First, the focus of our benchmark is on **video reasoning**, not **multilingual reasoning**. Furthermore, *most of the clues come from the **visual information***. This can be seen in Section 4.4 of the main text, where the performance gain from adding audio is not significant.
>   - Second, the main purpose of a reasoning benchmark should be to test a model's **thinking and reasoning abilities**. For example, current mainstream reasoning benchmarks like MATH500, AIME, and GPQA are **all entirely in English**. Mainstream video benchmarks (e.g., LVBench[3], Video-MMMU[4]) are also **all in English**.

---

> > ### Author Response · Authors · 2025-11-19
> > **Rebuttal 2**
> >
> > ## **Weakness 3**
> >
> > > While the paper tests frame count impact (Figure 4) for Gemini-2.0-Flash, it does not report efficiency metrics (e.g., inference time, memory usage) for models processing long videos.
> >
> > We will clarify this question from the following two points.
> > - First, we want to clarify that we used **Google's API** to test Gemini. **To the best of our knowledge, we were unable to obtain the "memory usage" metric you requested via the Google API**. As the model cannot be deployed locally, preventing us from monitoring this metric. Similarly, because we were using an API, we also **could not rigorously calculate the inference time**. The time we can actually measure is closely related not only to our video **upload speed** but also to the **network speed** of our server at the time. Furthermore, in practice, evaluating models via **API is not always stable**; if the upstream API is unstable, it might require multiple retries to get an inference. All these factors made it impossible for us to accurately calculate the true inference time.
> > - Second, we want to clarify that the reason we did not calculate efficiency metrics (e.g., inference time, memory usage) is **based on the precedent of previous video benchmarks**. Almost all previous video benchmarks (e.g., Video-MME[1], LongVideoBench[5]) have **not provided inference time and memory usage metrics** when evaluating Gemini for the same reasons.
> >
> > ## **Question 1**
> >
> > > The paper acknowledges that video categories like mystery and gaming are underrepresented. Could you elaborate on why these categories were excluded (e.g., difficulty in annotating reasoning tasks, lack of popular videos) and whether there are plans to expand MMR-V to include them in future updates?
> >
> > Similar to the response above, we will explain the reasons as follows.
> > - As mentioned above, the "Short Film" subcategory **does include some mystery videos**. However, many high-quality mystery videos are long films, which have **copyright restrictions**. As for the gaming category we mentioned, it mostly refers to synthetic videos, meaning videos generated within a game environment. These videos are **not very common on YouTube**, our collection source, and often lack high engagement and popularity. This feature does not align well with the principle mentioned in Chapter 3.
> > - Second, as shown in Figure 9, our MMR-V already includes **6 major categories and 24 subcategories** of videos. These categories already cover **most of the popular genres** on YouTube and demonstrate good diversity. Including all real-world video types would be very difficult given our resource and manpower limitations. So at this stage, we have chosen these 24 subcategories as they are sufficiently **representative**.
> > - Finally, we **do plan to create an enhanced version of MMR-V** based on community feedback after its release. It will incorporate more video types, task types, and so on.

---

> > > ### Author Response · Authors · 2025-11-19
> > > **Rebuttal 3**
> > >
> > > ## **Question 2**
> > >
> > > > For implicit reasoning tasks, the paper attributes models’ better performance to "abundant visual cues dispersed throughout the video." Have you explored whether models’ pre-trained world knowledge (e.g., understanding that "room number 7 symbolizes good luck") also contributes to this performance gap?
> > >
> > > Thank you for your question. As you said, some problems in Implicit Reasoning do require sufficient world knowledge. However, we want to clarify that this is **not the main reason** for the performance gap between implicit and explicit reasoning. We offer the following analysis and explanation:
> > > - **First, implicit reasoning requires an extra step compared to explicit reasoning: using world knowledge to perform metaphorical analysis.** As in the example from Figure 1, the explicit reasoning task only requires the model to discover the "lighter in the man's hand" to reach the final conclusion. However, the implicit reasoning task not only requires the model to find that "the girl's room number is 7" but also to infer from its world knowledge that "the number 7 symbolizes good luck" before it can make the final inference. This example shows that **using world knowledge for implicit reasoning is actually a more complex step** than what is required for explicit reasoning. Therefore, if a model finds the key clues for both tasks, the implicit one should be **harder** due to the additional step of metaphorical analysis.
> > > - **Second, the key factor for implicit reasoning being easier is the number of clue frames.** As mentioned in the first point, implicit reasoning has an extra step of "using world knowledge to understand a metaphor." However, implicit reasoning tasks **have more clues**. For example, in the implicit reasoning case from Figure 1, the sunny weather at the girl's door, the gloomy weather at the boy's door, and the blooming flowers where the girl walks **all hint at her good luck**. The model only needs to understand one of these hints to make the subsequent inference. In the explicit reasoning example, **only a few frames** show the lighter in the man's hand. *This difference in the number of clue frames **compensates for the added difficulty of the "using world knowledge" step** in implicit reasoning, leading to its superior performance.*
> > > - **Third, does the sufficiency of a model's world knowledge affect implicit reasoning performance?** In fact, the world knowledge required in MMR-V tends to be **common sense** (lucky number 7, Christmas, the COVID-19 pandemic, etc.). Powerful models generally possess this basic knowledge; for example, models like o4-mini and Gemini-2.5-pro all answered the example you mentioned correctly. To prove this, we **conducted a RAG experiment during the rebuttal phase**. RAG can be used to supplement external knowledge that a model may lack, allowing us to observe if the amount of a model's world knowledge affects implicit reasoning performance. Specifically, we conducted preliminary RAG experiments using both the open-source Qwen2.5-VL-7B and the proprietary Gemini-2.5-Flash, with Wikipedia as the knowledge base. The naive RAG approach retrieves passages based directly on the question and options, while the ReAct[6] RAG encourages the model to reason and plan whether to retrieve information and what to retrieve. The results are shown in the table below. It can be observed that neither naive RAG nor ReAct RAG provided a significant performance improvement. This result also indicates that the models generally possess the world knowledge required for MMR-V. **The amount of knowledge reserve does not have a significant impact on improving model performance.**
> > >
> > > |                              | Overall | Implicit | Explicit | Art  | Life | TV   | Film | Ani. | Phi. |
> > > |------------------------------|---------|----------|----------|------|------|------|------|------|------|
> > > | Qwen2.5-VL-7B                | 30.1    | 33.7     | 20.8     | 20.9 | 18.1 | 29.6 | 21.2 | 48.4 | 19.8 |
> > > | Qwen2.5-VL-7B (naive RAG)    | 28.0    | 31.7     | 18.5     | 20.1 | 16.4 | 28.6 | 19.1 | 46.0 | 18.6 |
> > > | Qwen2.5-VL-7B (ReAct RAG)    | 30.9    | 33.9     | 23.4     | 23.0 | 15.8 | 32.3 | 22.2 | 49.2 | 20.9 |
> > > | Gemini-2.5-Flash             | 51.2    | 52.9     | 46.9     | 45.3 | 39.5 | 50.3 | 47.9 | 65.6 | 34.9 |
> > > | Gemini-2.5-Flash (naive RAG) | 49.8    | 51.0     | 46.9     | 42.4 | 38.4 | 46.0 | 50.3 | 62.9 | 34.8 |
> > > | Gemini-2.5-Flash (ReAct RAG) | 52.1    | 54.8     | 45.1     | 45.3 | 44.1 | 50.3 | 49.0 | 66.4 | 31.4 |

---

> > > > ### Author Response · Authors · 2025-11-19
> > > > **Rebuttal 4**
> > > >
> > > > ## **Question 3**
> > > >
> > > > > The human experiment uses 200 tasks (100 correct/100 incorrect for GPT-4o). Were these tasks balanced across implicit/explicit reasoning and video categories?
> > > >
> > > > We can address this concern by detailing both our sampling methodology and the empirical results of that process.
> > > >
> > > > - **First, our sampling strategy did, in fact, account for the balance of various data categories**. For the human experiment, we randomly sampled 100 questions that GPT-4o answered correctly and 100 that it answered incorrectly. We then checked to **ensure that every task type** (i.e., the Ability Types mentioned in Table 5) and **video type** had sufficient data coverage. This was to ensure the overall balance of the data.
> > > > - Second, to address your specific question, **we have compiled the data distribution** for the 200 questions in the human experiment during rebuttal phase. The results is shown in the tables below. The results show that the distribution of **implicit/explicit reasoning and video types was roughly balanced**.
> > > >
> > > > **Video Type Distribution**
> > > >
> > > > | Video Type | Count |
> > > > | :--- | :--- |
> > > > | Animation | 54 |
> > > > | Movie | 39 |
> > > > | Life | 39 |
> > > > | TV | 32 |
> > > > | Art | 26 |
> > > > | Philosophy | 10 |
> > > >
> > > > **Reasoning Type Distribution**
> > > >
> > > > | Reasoning Type | Count |
> > > > | :--- | :--- |
> > > > | Explicit Reasoning | 102 |
> > > > | Implicit Reasoning | 98 |
> > > >
> > > > [1] Fu C, Dai Y, Luo Y, et al. Video-mme: The first-ever comprehensive evaluation benchmark of multi-modal llms in video analysis[C]//Proceedings of the Computer Vision and Pattern Recognition Conference. 2025: 24108-24118.
> > > >
> > > > [2] Zhou J, Shu Y, Zhao B, et al. Mlvu: A comprehensive benchmark for multi-task long video understanding[J]. arXiv e-prints, 2024: arXiv: 2406.04264.
> > > >
> > > > [3] Wang W, He Z, Hong W, et al. Lvbench: An extreme long video understanding benchmark[C]//Proceedings of the IEEE/CVF International Conference on Computer Vision. 2025: 22958-22967.
> > > >
> > > > [4] Hu K, Wu P, Pu F, et al. Video-mmmu: Evaluating knowledge acquisition from multi-discipline professional videos[J]. arXiv preprint arXiv:2501.13826, 2025.
> > > >
> > > > [5] Wu H, Li D, Chen B, et al. Longvideobench: A benchmark for long-context interleaved video-language understanding[J]. Advances in Neural Information Processing Systems, 2024, 37: 28828-28857.
> > > >
> > > > [6] Yao S, Zhao J, Yu D, et al. React: Synergizing reasoning and acting in language models[C]//International Conference on Learning Representations (ICLR). 2023.
> > > >
> > > > **[Final Remark]** Thank you again for your supportive and constructive feedback. We hope our explanations have addressed your concerns. We appreciate your positive assessment of our work and would be happy to provide any further clarification needed.

---

> > > > > ### Author Response · Authors · 2025-11-25
> > > > >
> > > > > Dear Reviewer NiAg,
> > > > >
> > > > > Thank you for your recognition and helpful feedback. As the discussion phase has now begun, we would greatly appreciate knowing whether our rebuttal has adequately addressed your concerns. Your insights are extremely valuable to us, and we are very willing to address any remaining issues to further improve the paper.

---

> ### Author Response · Authors · 2025-11-27
> **Looking Forward to Your Feedback**
>
> Dear Reviewer NiAg,
>
> We would like to express our appreciation for your time and the comments you have shared with us.
> In response to your review, we have submitted further explanations and clarifications to address the issues you highlighted. Since the discussion phase has already begun, we are keen to hear your thoughts on these updates and would appreciate it if you could verify whether our explanations have resolved your concerns. Should you have any further questions or require additional clarifications, we remain very willing to address them.
>
> Best regards,
>
> Authors

---

### Official Review · Reviewer_qv7x · 2025-10-30

**Soundness:** 2
**Presentation:** 3
**Contribution:** 3
**Rating:** 6
**Confidence:** 5

**Summary:**

This paper introduces MMR-V, a new benchmark designed to evaluate the deep multimodal reasoning capabilities of large language models in videos. MMR-V requires models to locate and analyze evidence across multiple, long-range frames to answer questions that involve hidden information. Through extensive testing on 21 different models, the study reveals that even the best-performing model, Gemini-2.5-pro, only achieves 64.3% accuracy, indicating a significant gap between current AI capabilities and complex video reasoning.

**Strengths:**

1.	It is an interesting and novel step to divide the reasoning tasks to be implicit and explicit. Evaluating the model’s capability to combine the precepted information with previously learned world knowledge to understand the metaphors is important.
2.	The entire benchmark is human-labeled, and several approaches are conducted from the video collection to data annotation to guarantee the overall quality to make the benchmark more reliable.
3.	Experiments are comprehensive, the authors evaluate enough frontier open-source and proprietary models, and compare with human performance. The authors also conducted a series of ablation studies regarding the input frames and modalities.

**Weaknesses:**

1.	The questions designed for reasoning are not as deep as the authors claim. First, all question-answer pairs are single-step and do not require reasoning chains. This makes the proposed benchmark less challenging compared with some widely adopted text or image reasoning benchmarks, which include some mathematical or scientific problems that require multi-step reasoning. Besides, for many reasoning question types, such as personal reflection, video naming, and meta-emotion, it is more likely to be visual perception tasks. What reasoning capabilities are needed for these tasks? Readers may think the authors overclaim some perception problems as “deep reasoning” tasks.
2.	The authors have detailed the generation process of the wrong answers, yet the description of the human annotation process is vague and unclear. Are the annotators specifically trained for this labeling task? Is there any human review process for the human annotations? How to guarantee the overall quality of the human labeling process? More clarifications are needed.
3.	The literature review only includes some video perception benchmarks, but more recent video reasoning benchmarks are not involved and compared, such as VideoEspresso, VRBench, VideoReasonBench, CG-Bench, and MMVU. I believe some discussions are needed to highlight the contribution.

**Questions:**

1.	Why do the authors only select the multiple-choice question as the evaluation protocol? Are open-ended questions essential for the reasoning tasks?
2.	For the annotation process, are there any time interval limitations for the questions? The checklist requires long distance, how long is it, and how to guarantee it?
3.	For the implicit reasoning question type, how can to ensure that all annotators can obtain all world knowledge to understand the metaphors in the videos?
4.	Is there the possibility that the discrepancy in the annotators’ cultural background leads to different reasoning results?

---

> ### Author Response · Authors · 2025-11-19
> **Rebuttal 1**
>
> Dear Reviewer qv7x,
>
> We wish to express our sincere gratitude for your detailed evaluation and constructive feedback. We are particularly grateful for your recognition of our work's novelty, the quality of our benchmark, and the comprehensiveness of our experiments. We are happy to address your concerns below, while also hoping to clarify a few potential misunderstandings.
>
> ## **Weakness 1**
>
> > The questions designed for reasoning are not as deep as the authors claim.
>
> We need to clarify that our benchmark is designed to test a model's **multi-modal reasoning** ability in videos. This includes how to find keyframes through reasoning and how to reason about the relationships between them. Your view that it does not require deep reasoning might be from a human perspective. For a human observer, some tasks might seem simple. Our ability to process an entire video and effortlessly synthesize relationships between disparate frames makes the task feel like mere "perception." For models, however, the reality is starkly different; they struggle to find the **crucial keyframes** through reasoning. Experiments show that they tend to focus more **on question frames**.
>
> In fact, many videos in our benchmark are not simple, even for humans. Here, I can provide a few examples:
>
> 1. **Example 1**: [Video](https://www.youtube.com/watch?v=gooWdc6kb80). We ask the model to guess what is the minimum number of video shots that had to be filmed to edit together the opening scene of the man breaking the water balloon? Ability Type: Sequential Structure Reasoning. [Reference Answer](https://www.youtube.com/shorts/WDvjtRh90OI)
> 2. Example 2: [Video](https://www.youtube.com/watch?v=D-eDNDfU3oY). We require the model to infer whether and why this video is being played forwards or in reverse. Ability Type: Counterintuitive Reasoning
>
> > All question-answer pairs are single-step and do not require reasoning chains. This makes the proposed benchmark less challenging compared with some widely adopted text or image reasoning benchmarks, which include some mathematical or scientific problems that require multi-step reasoning.
>
> Here, we want to clarify that the **"multi-modal reasoning" and "text-based reasoning"** are two different, challenging types of reasoning, as stated in our paper (L136). The mathematical and scientific problems you refer to are indeed the text-based reasoning, requiring long chains of thought that involve hypothesis, verification, computation, and formal proof. However, we argue that the definition of ***"multi-step reasoning"** should not be exclusively confined to tasks with explicit, sequential deductive chains*. **Many real-world applications**, such as intelligent surveillance[1], actually require a model to act **like a Sherlock Holmes** detective, finding clues distributed across vast amounts of multimodal information. Given the intractable complexity of the real world, we employ the video modality as a simplified environment to evaluate this capability. For example, the "judging if a video is in reverse" task we mentioned above is a simplified version of real-world detective tasks. While this reasoning process lacks the interlocking calculations, deductions, and proofs, it equally demands deep planning and reasoning.
>
> Secondly, **these reasoning tasks that humans find easy but on which models perform poorly are a very meaningful and interesting research direction**. In established reasoning domains like *mathematics, models have already achieved superhuman performance* on benchmarks such as AIME and MATH500. However, many researchers point out that these same models cannot be deployed in many real-world scenarios (e.g., embodied intelligence), *as their analysis and reasoning in real-world situations are far less "smart" than what they demonstrate on math problems*. MMR-V was specifically designed to highlight this discrepancy by presenting tasks that are easy for humans but prove difficult for models that excel in other domains. Our results show that even **the best model is 21.7% less accurate than humans. We believe this gap is worthy of investigation**.

---

> > ### Author Response · Authors · 2025-11-19
> > **Rebuttal 2**
> >
> > > For many reasoning question types, such as personal reflection, video naming, and meta-emotion, it is more likely to be visual perception tasks.
> >
> > We believe there may be a **misunderstanding** regarding these three tasks. The three tasks you mentioned are all forms of **implicit reasoning**. They require implicit reasoning more than just perception; they involve abstracting the main theme or metaphor from information within the video. The examples for these tasks provided in Appendix E, and none are solvable through simple perception.
> > Additionally, we speculate that the issue might stem from a **misunderstanding of the examples in Appendix E**. Examples in Appendix E shows links to the original videos, which, in their raw form, may contain explicit answers in their opening or closing segments (e.g., via a title card or a concluding thematic statement). We must clarify that all videos in the final MMR-V dataset have been **carefully processed to trim the frames at the beginning and end that contain the answer**. We ensure that the model cannot directly see information like the video name where the answer might be revealed. We guarantee that the final videos in MMR-V **do not contain these kinds of shortcuts** that can be solved by direct perception; this has been **carefully reviewed by the paper's main authors**. The primary reason for showing video links is that the video files are too large to upload, so this is how we present the examples.
> > If this was the source of the misunderstanding, we apologize for causing it. We will add a disclaimer in the appendix to state that the videos in the dataset have been processed to remove any direct answers.
> >
> > ## **Weakness 2**
> >
> > > The authors have detailed the generation process of the wrong answers, yet the description of the human annotation process is vague and unclear.
> >
> > Thank you for your careful review and for providing such detailed suggestions. In fact, we do have a training and annotation process to ensure the quality of our error analysis. I will clarify this concern as follows:
> > - First, we have a **strategy to ensure annotator consensus on the cause of errors**. We began by **roughly classifying** the model's error, which resulted in the six major error categories introduced in Section 4.5, for which we provided definitions. At the same time, for each error type, we selected **some representative tasks and model responses** to better distinguish between the different causes in a practical way. Some of these examples are shown in Appendix J. We invited non-authors to review these definitions and examples to reach a consensus on them, thus avoiding bias. Finally, the **main authors of the paper** (who have a **deep understanding of the dataset**) performed the error cause annotation based on these **definitions and examples and unified their results**. We did not detail this entire process in paper, but we will add it to the new version.
> > - **Second, during the rebuttal stage, we invited an individual who did not participate in the original error annotation to perform a check**. The checker is trained using the aforementioned definitions and error response examples. This was done to provide a quantitative metric for the quality of our error analysis. The results showed that disagreements arose on **only 5%** of the samples. *This does **not affect the conclusions** in the paper and serves to demonstrate the quality of our error analysis.*

---

> > > ### Author Response · Authors · 2025-11-19
> > > **Rebuttal 3**
> > >
> > > ## **Weakness 3**
> > >
> > > > More recent video reasoning benchmarks are not involved and compared, such as VideoEspresso, VRBench, VideoReasonBench, CG-Bench, and MMVU. I believe some discussions are needed to highlight the contribution.
> > >
> > > We sincerely thank the reviewer for highlighting these excellent video benchmarks. It was our oversight not to have discussed all of them. However, there are also some misunderstandings we need to clarify. Below, we will discuss these benchmarks and the differences with our MMR-V.
> > > - VideoEspresso introduces a **fully automated pipeline for generating video reasoning data with Chain-of-Thought (CoT)**. Its strength lies in its **scalability**, enabling the creation of a large-scale dataset (200k+ training, 1.4k test) by having GPT-4o annotate keyframes. In contrast, our MMR-V focuses more on the **difficulty and quality** of the video reasoning benchmark through pure manual QA annotation, with **fine-grained control over task categories, quality, reliability, and authenticity**.
> > > - CG-Bench is a valuable benchmark for **long video understanding**, using a "clue-grounded" methodology to ensure questions are directly tied to video content. We want to clarify that CG-Bench is a video understanding rather than a video reasoning benchmark. For example, a task from it like, "In the video, during a dinner in Italy, what did the protagonist and her husband do after a man finished singing?" leans more towards **perception and understanding**.
> > > - MMVU and Video-MMMU are focused on evaluating a model's **specialized knowledge reasoning(more like text-based reasoning) within multi-disciplinary videos** (e.g., physics, economics). In contrast, MMR-V utilizes **open-domain** videos to assess the general multi-modal reasoning capabilities across various real-world videos and different reasoning tasks.
> > > - Finally, we want to clarify that the reason VRBench and VideoReasonBench were not introduced in the current paper is that they were **concurrent works with ours**. However, we are very willing to discuss their outstanding contributions.
> > >   - VRBench focuses on reasoning in **Long Narrative Videos**, with a greater emphasis on reasoning about the overall narrative plot of the video. Our MMR-V, on the other hand, focuses on in-depth reasoning about various aspects such as video plot, structure, editing strategies, and themes, and is not concerned with whether the video is a long narrative one. The **detailed differences** between VRBench and MMR-V in terms of annotation strategy and question difficulty can be found in our response to **Reviewer Xppq**.
> > >   - VideoReasonBench shares a similar core idea with ours, testing a model's **deep reasoning of video information and its Vision-Centric reasoning capabilities**.  A key feature is that the videos in their dataset are **all synthesized using an engine**, which challenges a model's in-depth reasoning and memory of visual information in the video, making it very challenging. For example, a task might be, "There are some upside-down cups, some of which initially have balls under them. After shuffling the cups, the model is asked which cups have balls in the end." In these tasks, every frame of the video is a keyframe, and missing the details of a single frame will lead to an incorrect result. The key difference is that the tasks in our MMR-V are all from **real-world, open-domain, and diverse types of videos, making them closer to reality**.

---

> > > > ### Author Response · Authors · 2025-11-19
> > > > **Rebuttal 4**
> > > >
> > > > ## **Question 1**
> > > >
> > > > > Why do the authors only select the multiple-choice question as the evaluation protocol? Are open-ended questions essential for the reasoning tasks?
> > > >
> > > > The fundamental reason for choosing multiple-choice questions is to make the evaluation **more scientific and reliable**. This can be considered from the following points:
> > > > - Currently, the **main open-ended reasoning tasks**, such as math reasoning (MATH500 and AIME), have **a single standard answer** (or a variation) that allows for checking the accuracy of the response. The answers to our tasks are mostly a sentence rather than a few numbers, making them **difficult to judge accurately with a judge function**. For answers that cannot be judged accurately by a function, most works use **an LLM to score** them against the ground truth. However, the *reliability of LLM scoring and its alignment with human values remain questionable*. *As a **reasoning** dataset, we prefer to provide a **strict and reliable** performance metric.* Furthermore, while we can have a strict answer for Explicit Reasoning tasks, it would be **very difficult for Implicit Reasoning tasks** to judge how similar a model's understanding of a metaphor or theme is to the official interpretation, and to what extent the model has grasped the video's meaning, as these tasks involve abstract understanding of metaphors and hidden meanings.
> > > > - Second, most of **the excellent video benchmarks** of the past, such as Video-MME, LongVideoBench, LVBench, MLVU, and Video-MMMU, have **all adopted a multiple-choice format**. This also served as a basis for our choice.
> > > > - Finally, **a key factor is that performance is poor with an open-ended format**. As we state in our main text (Section 3.2, L263), during the initial data annotation phase, we tested GPT-4o with open-ended setting on 100 samples. The accuracy, as checked by humans, was **only 17%**. GPT-4o often struggles to provide on-point answers, which could lead to *a compressed performance distribution across models in the dataset, reducing its ability to differentiate them*. Therefore, we did not consider an open-ended task format during our initial planning.
> > > >
> > > > ## **Question 2**
> > > >
> > > > > Are there any time interval limitations for the questions? The checklist requires long distance, how long is it, and how to guarantee it?
> > > >
> > > > - First, our original checklist had some explicit requirements, as per the checklist in Appendix C.1. We required annotators to watch the entire video before starting. This helps the annotator understand the full content and propose more global and long-span questions.
> > > > - Second, we **initially had a strict quantitative** definition for "long-distance multi-frames."  Our original checklist here required that the video clue frames needed for a question must **cover at least 20% of the video duration or exceed 1 minute**, and require at least **5 clue frames**. However, during the annotation process, we found that *some tasks, even without meeting these two requirements, were still very challenging for deep reasoning of multimodal information (such as the Example 1 shown in Weakness 1)*. These tasks also test the deep reasoning abilities and **align with our paper's core idea** of "Multimodal Deep Reasoning in Videos." Therefore, we included these tasks in the final dataset. For the sake of rigor, we ultimately removed the quantitative data requirements from the checklist in Appendix. However, such data is not numerous and does not affect the overall long-range, multi-frame nature of the dataset.
> > > >
> > > > To provide a **concrete, quantitative validation** of this characteristic, we also sampled and checked its quality after the dataset was completed. The results are stated in the Introduction: "**Tasks typically require reasoning over an average of 12 video frames, covering about 60% of video duration.**" This result can prove that, even with some exceptional cases, our dataset as a whole has the long-range, multi-frame property and provides a quantitative metric for "how long" the span is.

---

> > > > > ### Author Response · Authors · 2025-11-19
> > > > > **Rebuttal 5**
> > > > >
> > > > > ## **Question 3**
> > > > >
> > > > > > For the implicit reasoning question type, how can to ensure that all annotators can obtain all world knowledge to understand the metaphors in the videos?
> > > > >
> > > > > During the process of annotating implicit reasoning tasks, we use **reliable, real-world references** to help annotators **understand the metaphors** and provide answers **without subjective bias**. Specifically, as mentioned in the *principles in Chapter 3 and the annotation method in the Appendix C.1 CHECKLIST*, Specifically, as outlined in Appendix B.2, **we rely on official comments from the video creators or highly upvoted and widely recognized interpretations from the comment section of original video**. This helps to better align our annotations with the common understanding of a wide range of real-world users. This approach effectively helps annotators acquire the necessary *"world knowledge" to understand the metaphors, while also helping to mitigate the inherent biases of a limited group of annotators*, allowing us to better model the understanding that is endorsed by the majority of real users.
> > > > >
> > > > > ## **Question 4**
> > > > >
> > > > > > Is there the possibility that the discrepancy in the annotators’ cultural background leads to different reasoning results?
> > > > >
> > > > > - If you are referring to the human experiment part, then people from different cultural backgrounds might have slight deviations. Due to resource limitations, the human experimenters in this paper are from a single cultural background. However, we can assure you that **this impact is not significant**, because this factor mainly affects the "Cultural Symbols" task sub-category, and this type of question only accounts for **4.6%** of the total.
> > > > > - If you are referring to the annotation process, our annotators are required to ground their work in **external resources**, such as official explanations from the video creators. Furthermore, all annotations are subject to a review process to ensure consistency and objectivity. Therefore, there will **not be a significant impact from bias**.
> > > > >
> > > > > [1] Ibrahim S W. A comprehensive review on intelligent surveillance systems[J]. Communications in science and technology, 2016, 1(1).
> > > > >
> > > > > **[Final Remark]** Thank you again for your insightful feedback on our paper. We hope that our responses and clarifications have addressed your questions and concerns. We would be happy to engage in any further discussion and welcome any additional questions you may have.

---

> > > > > > ### Author Response · Authors · 2025-11-25
> > > > > >
> > > > > > Dear Reviewer qv7x,
> > > > > >
> > > > > > We sincerely thank you for your recognition of our work and insightful comments. Following your suggestions, we have updated the revised paper to include a discussion on recent video benchmarks. Regarding your concerns, we have provided corresponding clarifications and new quality verification results in the rebuttal response above. Your insightful comments are valuable to us. We remain fully open to addressing any further questions you may have.

---

### Official Review · Reviewer_9zhX · 2025-10-30

**Soundness:** 3
**Presentation:** 3
**Contribution:** 2
**Rating:** 4
**Confidence:** 3

**Summary:**

In this work, the authors propose MMR-V, a benchmark for multimodal reasoning in videos. MMR-V consists of 317 videos and 1,257 tasks, for  long-range, multi-frame reasoning in the real world. The authors also investigate the current MLLMs models on this benchmark.

**Strengths:**

*Clarity

The paper is well-written with good structure. Hence, the clarity is basically good.

*Significance

This paper focuses on evaluating video reasoning capacity of MLLMs, which is an important and practical problem for video understanding. Hence, the significance is basically OK for video research community.

**Weaknesses:**

* Reference

1) The recent work [VRBench: A Benchmark for Multi-Step Reasoning in Long Narrative Videos, ICCV 2025] proposes the similar topic for video understanding. Please clarify the key difference.
2) Small suggestion: It would be clearer to include a table to show key statistics difference between this bench and the existing ones such as Video-MME,  LongVideoBench, LVBench, Video-MMMU, MMVU, etc.

* Method Insight

1) It woule be more interesting to investigate or indicate how to design MLLMs to tackle the tasks in this benchmark.
2) Actually, there exists some agentic mechanism for long video reasoning such as VideoAgent, VideoTree, etc. These methods are suitable for reasoning via tool usage with multi-round clue discovery. Hence, it would be great to include these works for evaluating the benchmark.

* Small Size

The authors collected only 317 original videos. The small number of videos would restrict the generalization of this benchmark.

**Questions:**

Please see the weakness section.

---

> ### Author Response · Authors · 2025-11-19
> **Rebuttal 1**
>
> Dear Reviewer 9zhX,
>
> Thank you for dedicating your time and effort to reviewing our work. We sincerely appreciate your recognition of the clarity of our paper and the significance of our contributions. We hope that our response satisfactorily addresses your concerns.
>
> ## **Weakness 1**
>
> > The recent work [VRBench] proposes a similar topic for video understanding. Please clarify the key difference.
>
> We appreciate you for pointing us to the excellent work VRBench. Since the works **were conducted concurrently**, the current version of our paper does not provide a discussion of VRBench, and we apologize for this oversight. We now offer a clear description of its contributions and explicitly highlight the key differences between VRBench and our MMR-V.
>
> VRBench is designed to evaluate a model’s reasoning capabilities over **long narrative videos**. It adopts a scalable human–model collaborative annotation pipeline and provides **stepwise** reasoning annotations. Below, we compare VRBench and our MMR-V across **four key dimensions to clarify the major distinctions** between the two benchmarks.
>
> **1. Differences in motivation and core idea.**
>   - VRBench primarily focuses on **long narrative videos**, with an average duration of 1.6 hours and genres centered on narrative content (e.g., film, travel, sports). Its goal is to evaluate a model’s ability to reason about **plot lines and understand story development**, rather than to test fine-grained visual cue reasoning. A typical question would be: “Why was the woman crying?”
>   - In contrast, the core idea of MMR-V is to evaluate a model’s ability to perform **deep reasoning over fine-grained visual clues** within videos. Our benchmark is not only about assessing a model’s understanding of global narrative structure; instead, it emphasizes the analysis and extraction of multimodal micro-level clues. As illustrated in our logo, we aim for models to **“think with video”**, just like *Sherlock Holmes examining some evidences*. Moreover, our dataset contains a large number of **non-narrative videos** (such as magic performances, videos with sophisticated editing, or artistic content). These videos are non-narrative structure but contain meticulously designed clues. This aligns with our data collection principle stated in Section 3.1.
>   - Here we provide two comparative examples to clearly show the difference between the **deep reasoning** over video required by MMR-V and the focus on **narrative plot reasoning** in VRBench (difference in the core idea).
>     - **VRBench: Focus on Narrative Comprehension.**
>
>       Example1. Question: "Why did the male protagonist arrange for his friend to rob the female protagonist?" (We couldn't find the original video link. However, this is the example shown in Figure 2 of the original VRBench[2] paper, which you can refer to to learn about the plot.) This is actually an understanding of the overall plot of the video.
>     - **MMR-V: Focus on Deep, Multimodal Reasoning Beyond Narrative.**
>
>       Example 1. [Video](https://www.youtube.com/watch?v=gooWdc6kb80). We ask the model to guess what is the minimum number of video shots that had to be filmed to edit together the opening scene of the man breaking the water balloon? Ability Type: Sequential Structure Reasoning. [Reference Answer](https://www.youtube.com/watch?v=WDvjtRh90OI&t=5s)
>
>       Example 2. [Video](https://www.youtube.com/watch?v=D-eDNDfU3oY). We require the model to infer whether and why this video is being played forwards or in reverse. Ability Type: Counterintuitive Reasoning
>
>       Our appendix contains examples of all 33 subcategories of tasks, with more examples demonstrating the differences between our tasks and those for plot reasoning tasks.

---

> > ### Author Response · Authors · 2025-11-19
> > **Rebuttal 2**
> >
> > **2. Differences in annotation strategies.**
> >   - VRBench employs a semi-automated human–AI collaborative annotation pipeline. (1) Videos are segmented using AutoShot, and **captions** are generated via VideoChat2. Whisper-large-v3 and DeepL are then used to extract and translate **audio** into English. (2) Captions and transcripts are fed into **GPT-4o to generate QA pairs and multi-step reasoning chains**. (3) **Human annotators** then refine the QA pairs based on GPT-4o outputs, videos, and subtitles. (4) False options for multiple-choice questions **are generated by DeepSeek-V3** based on the annotated answers.
> >   - In MMR-V, **all QA pairs are carefully curated by humans**, allowing us to better control question **quality and difficulty**. (1) Unlike VRBench, we instead provide annotators with a **detailed checklist and curated annotation examples**, enabling trained annotators to generate high-quality QA pairs. During annotation, annotators are also encouraged to record potential distractor candidates. When determining QAs, annotators are encouraged to refer to official explanations by video creators and to top-voted comments, ensuring alignment with real-world user interpretation and reducing annotator bias. (2) **For distractor options**, as described in Section 3.2, we investigate **three different strategies**. The approach of VRBench corresponds roughly to our **strategy 2**. We conducted a preliminary inspection of the distractors produced by all three strategies and found that directly relying on models to generate incorrect options (strategy 2) leads to **low-quality** distractors with poor diversity. The results in Table 1 further support this observation.
> >
> > **3. Differences in task categories.**
> >   - Because VRBench focuses on narrative video understanding, its task taxonomy consists of **7 categories** reflecting different aspects of narrative reasoning, and GPT-4o is prompted to generate tasks based on these descriptions.
> >   - MMR-V, however, targets a much broader range of **multimodal reasoning abilities**. The tasks involve a wide spectrum of abilities beyond narrative comprehension. Thus, we categorize reasoning tasks into Implicit Reasoning and Explicit Reasoning, which are further divided into **10 categories and 33 subcategories**. This richer set of task categories can provide a more comprehensive analysis of the reasoning abilities required for deep multimodal video reasoning.
> >
> > **4. Evidence of task difficulty.**
> >   - From a results perspective, we show that models struggle more with **fine-grained clue reasoning** than with reasoning about narrative plots. This is one of the central motivations behind MMR-V.
> >   - For example, although the average length of VRBench videos is **1.6 hours**, GPT-4o achieves **81.23%** accuracy on MCQ tasks with only *64 frames*. In contrast, on MMR-V, with videos averaging **277 seconds**, GPT-4o achieves only **55.0%** accuracy with *300 frames*. Similarly, Phi-3.5-Vision, using 8 frames, reaches **58.03%** accuracy on VRBench, whereas Phi-4-multimodal-instruct, using 8 frames, attains only **27.6%** accuracy on MMR-V.
> >   - These results indicate that, even with significantly shorter videos, models struggle to deeply mine and reason over fine-grained visual clues. These results can illustrate our core ideas and contributions.
> >
> > Finally, we will **explicitly acknowledge the valuable contributions of VRBench in the revised version of our paper.**
> >
> > > Small suggestion: It would be clearer to include a table to show key statistics difference between this bench and the existing ones.
> >
> > Thank you for your valuable suggestion. In response, we have compiled the following statistics from previous benchmarks and will include them in the revised version of our paper.
> >
> > | Benchmark | #Videos | #Dur.(s) | #QA Pairs | Data Source | Task Categories | Multilingual |  QA Anno. |
> > |---|---|---|---|---|---|---|---|
> > | MVBench | 3,641 | 16.0 | 4,000 | Open-Domain | 20 | X | A |
> > | EgoSchema | 5,063 | 180.0 | 5,063 | Egocentric | X | X | A |
> > | TempCompass | 410 | 11.4 | 7,540 | Open-domain | 11 | X | A&M |
> > | Video-MME | 900 | 1017.9 | 2,700 |  Open-domain | 12 | ✓ | M |
> > | LongVideoBench | 3,763 | 473.0 | 6,678 | Open-Domain | 17 | X | M |
> > | LVBench | 103 | 4101.0 | 1,549 | Open-Domain | 26 | X | M |
> > | CGBench | 1,219 | 1624.4 | 12,129 | Open-Domain | 12 | X | M |
> > | Video-MMMU | 300 | 506.2 | 900 | Open-Domain | 6 | X | M |
> > | MMVU | 1,529 | 51.4 | 3,000 | Multi-Disc | X | X | M |
> > | VRBench | 960 | 5796.0 | 8,243 | Narrative | 7 | ✓ | A&M |
> > | MMR-V(Ours) | 317 | 277.0 | 1,257 | Open-Domain | 33 | ✓ | M |

---

> > > ### Author Response · Authors · 2025-11-19
> > > **Rebuttal 3**
> > >
> > > ## **Weakness 2**
> > >
> > > > There exists some agentic mechanism for long video reasoning such as VideoAgent, VideoTree, etc. ... It would be great to include these works for evaluating the benchmark.
> > >
> > > We sincerely thank the reviewer for pointing out two excellent agent frameworks for long video understanding, VideoAgent [1] and VideoTree [2]. Their overall ideas are similar: both use a proposed Agent framework to extract the missing video frames, which are then described and handed over to an LLM for reasoning and planning. **The main difference lies in their acquisition strategies for missing frames**. VideoAgent uses a **multi-turn** process where the LLM describes missing information and then is provided with the relevant video frames to make its next decision. In contrast, VideoTree first clusters the video frames, **then builds a tree by expanding based on relevance**, and filters from coarse to fine granularity according to the question. The selected frames are then provided to the LLM **all at once** for reasoning. Since their core ideas are similar, we have chosen **VideoAgent** for our evaluation.
> > >
> > > The original backbone of VideoAgent is an LLM, so we chose **Deepseek R1**, a LLM that has demonstrated strong reasoning capabilities recently, as the foundation. However, in the process of reproducing VideoAgent, we discovered that *feeding only the caption descriptions* of the corresponding video frames to the model for reasoning causes a significant loss of information. For example, in the implicit reasoning video sample from Figure 1 of our paper, a frame is described as "there are two pictures of a house with a yellow door and a pink door," which fails to mention the key information that "the numbers on the doors are 7 and 13, respectively, implying the metaphors of good luck and bad luck." Therefore, we slightly modified the VideoAgent framework to use **MLLMs** with strong multimodal reasoning capabilities as the foundation (**o4-mini, GPT-5, and Gemini-2.5-Pro**). In this setup, we directly input the retrieved missing frames from each round into the MLLM without an intermediate captioning step. We set the initial frames for all models to be 32 uniformly sampled frames. The test results are shown in the table below.
> > >
> > >
> > > | Model | Overall | Short | Medium | Long | Implicit | Explicit | Art | Life | TV | Film | Ani. | Phi. |
> > > | :--- | :--- | :--- | :--- | :--- | :--- | :--- | :--- | :--- | :--- | :--- | :--- | :--- |
> > > | VideoAgent Deepseek-R1 | 37.2 | 36.5 | 37.1 | 35.8 | 39.0 | 32.5 | 38.6 | 30.9 | 34.0 | 36.3 | 43.1 | 38.5 |
> > > | o4-mini-2025-04-16 | 52.5 | 57.1 | 53.0 | 47.2 | 54.6 | 47.1 | 48.2 | 40.1 | 54.0 | 51.7 | 65.3 | 27.9 |
> > > | VideoAgent o4-mini-2025-04-16 | 53.7 | 56.5 | 53.7 | 50.7 | 55.2 | 49.7 | 49.6 | 40.1 | 60.8 | 51.7 | 65.3 | 27.9 |
> > > | Gemini-2.5-pro (1fps) | 64.3 | 64.7 | 64.1 | 63.9 | 65.9 | 60.2 | 57.6 | 58.1 | 65.6 | 62.5 | 73.0 | 54.6 |
> > > | VideoAgent Gemini-2.5-pro | 61.3 | 62.3 | 61.4 | 60.3 | 61.4 | 60.9 | 56.1 | 41.8 | 60.3 | 60.4 | 77.0 | 45.3 |
> > > | GPT-5-2025-08-07 | 60.6 | 62.3 | 61.6 | 56.2 | 61.0 | 59.7 | 56.8 | 41.8 | 56.6 | 60.4 | 76.5 | 45.4 |
> > > | VideoAgent GPT-5-2025-08-07 | 62.2 | 62.0 | 63.2 | 60.6 | 63.6 | 58.6 | 61.2 | 52.5 | 63.0 | 57.6 | 74.1 | 45.3 |
> > >
> > >
> > > To intuitively demonstrate the benefits of the VideoAgent framework, we provided the direct performance of three MLLMs on MMR-V. The results show that o4-mini and GPT-5 achieved **performance gains of +1.2% and +1.6%**, respectively. In contrast, Gemini-2.5-pro's performance degraded by **-2.0%**. We provide the following analysis and explanation for these results:

---

> > > > ### Author Response · Authors · 2025-11-19
> > > > **Rebuttal 4**
> > > >
> > > > - We segmented the tasks based on video duration into Short (<2 min), Medium (2-6 min), and Long (>6 min) and reported scores for each subset. Interestingly, when integrated with VideoAgent, the performance of o4-mini and GPT-5 **on Short and Medium videos remained largely unchanged. However, their performance on Long Videos improved**.
> > > > - As articulated in the VideoAgent paper, the framework is designed to address the challenges of **long video understanding**. Its core mechanism, retrieving missing frames, typically serves to supplement information that is ***mentioned in a query but is absent from the initial visual input***. For instance, the case study in the VideoAgent paper shows that when asked, "What is the color of the stairs surrounded by green plants?", the model generates a query for more information like, "I can't provide details about these images as they don't contain any stairs surrounded by green plants." Then it uses semantic similarity to retrieve the relevant frames. In our experiments, the limited 32-frame input for o4-mini and GPT-5 might lead to the **omission of question frames in some long videos**. VideoAgent rectifies this by retrieving these frames, thereby boosting performance. Conversely, for a model like Gemini, which can natively process a high volume of frames (e.g., at 1 fps), the overhead of the VideoAgent framework is less effective than simply providing all available frames at once.
> > > > - *Why does VideoAgent fail to enhance reasoning performance on **short and medium-length videos?*** As the VideoAgent case studys illustrate, the model primarily **requests retrieval of information that is explicitly mentioned in the text of the question**. These frames correspond precisely to what we define as **"Question Frames" in our work**. The successful retrieval of Question Frames is largely a test of a model's image-text matching capability. However, a core objective of MMR-V, as stated in our abstract, is to evaluate: "Long-range, multi-frame reasoning: Models are required to infer and analyze **evidence frames that may be far from the question frame**." Our benchmark is designed to probe the ability of model to conduct deep reasoning and uncover "evidence frames" located elsewhere in the video, rather than just analyzing the frames directly referenced in the question. This conclusion is further supported by our analysis in Section 4.5 and Figure 6, which shows that top-performing models on MMR-V **dedicate more analysis to non-question frames**.
> > > > - In summary, VideoAgent is an exceptional contribution to **long video understanding**. The problem they solve focuses on the difficulty that a **model cannot directly input all frames from a long video**, thus requiring a tool to retrieve and supplement the **missing question frames**. However, our results show that for the short and medium-length videos in MMR-V, the model's initial input frames already contain the necessary question frames. VideoAgent struggles to help the model find other clue frames that are far from the question frames. In contrast, the focus of MMR-V is on whether the model can **perform deep reasoning on the video to find these clues**.

---

> > > > > ### Author Response · Authors · 2025-11-19
> > > > > **Rebuttal 5**
> > > > >
> > > > > ## **Weakness 3**
> > > > >
> > > > > > It woule be more interesting to investigate or indicate how to design MLLMs to tackle the tasks in this benchmark.
> > > > >
> > > > > Based on our experiments, we would like to ***offer some insights and discussion on solving the tasks in MMR-V***. According to our analysis above, while some existing works have proposed using Agents to understand video, they focus more on the problem of "models being unable to read all frames from a long video at once, thus missing some question frames." MMR-V, however, requires in-depth analysis of the video **beyond just the question frames**. We believe there are **two potential approaches** to solve this.
> > > > >
> > > > > - First, the task in MMR-V is essentially a "Think with Video" problem. Similar to the excellent results achieved by **o3 in image reasoning**, where the model is trained to use various tools to manipulate images (e.g., cropping and zooming), we believe that **training a model to use a similar toolset for video** is a potential solution. Both VideoAgent and VideoTree are Training-Free methods that prompt the model to use tools. This is *not actually reasoning about the multimodal information of the **video itself***. For example, in the implicit reasoning problem shown in Figure 1 ("Why did the broken umbrella... get fixed... in the air?"), VideoAgent can only retrieve some frames related to the umbrella based on the question. It cannot reason back to the beginning of the video to find the metaphorical clue that the girl's room number is "7". Therefore, a key direction for MLLM design and training is to **define an effective toolset for video reasoning** and to encourage models to attend more closely to the rich multimodal information present in videos.
> > > > >
> > > > > - Second, some other works are exploring exhaustive descriptions of visual information[3][4]. These methods aim to generate highly **detailed textual description of images, sometimes with controllable granularity**. We believe this is a feasible strategy. If video frames can be converted into sufficiently detailed and accurate descriptions(e.g. capturing critical clues like "the girl's room number" or "the details on the cards in magician's hand"), the task could be fundamentally transformed. It would shift from **a complex video reasoning challenge to a more tractable long-text reasoning problem**. And locating visual clues becomes a matter of locating textual information.

---

> > > > > > ### Author Response · Authors · 2025-11-19
> > > > > > **Rebuttal 6**
> > > > > >
> > > > > > ## **Weakness 4**
> > > > > >
> > > > > > > The authors collected only 317 original videos. The small number of videos would restrict the generalization of this benchmark.
> > > > > >
> > > > > > - Video reasoning benchmarks demand a **higher standard of video quality** compared to video understanding benchmarks. Our collection process was entirely manual and subject to a meticulous screening process guided by the principles outlined in Section 3.1. As MMR-V is designed to evaluate a model's capacity for in-depth video analysis, we prioritized content with **sophisticated design**, such as the example with clever editing techniques shown in Weakness1. These meticulously crafted, high-quality videos are few in number. Conversely, more linear content like sports broadcasts was excluded for being too straightforward. Secondly, to avoid introducing niche biases, we need to ensure the popularity and discussion volume of the videos. Adhering to these stringent criteria significantly limited the pool of suitable, high-quality videos. Similarly, *a recent representative academic video reasoning benchmark, VideoMMMU, also has only **300 videos** (and is used as a benchmark for new model releases by many companies, such as for Gemini-2.5-pro)*.
> > > > > > - We have **ensured our videos are both diverse and representative**. In terms of duration, *the videos span from 7 to 3,771 seconds*. *Regarding video types, the dataset covers 6 major and 24 minor categories*, as detailed in Figure 9. We also made efforts to include a *wide array of thematic subjects*. Therefore, our video diversity is excellent, can cover many real-world scenarios, and is sufficiently representative.
> > > > > > - **We primarily test reasoning models, which have high token consumption and long evaluation times**. Existing reasoning benchmarks, such as AIME and GPQA, are designed for convenient and rapid testing and thus have a very small number of questions. This also makes it easier for developers to evaluate model performance.
> > > > > > - Finally, **the high cost of manual annotation** makes scaling MMR-V challenging. Nevertheless, we **plan to develop an improved version** of MMR-V by incorporating community feedback and suggestions following its initial release. Future iterations will feature more comprehensive and difficult tasks, as well as an expanded set of videos and questions.
> > > > > >
> > > > > > [1] Wang X, Zhang Y, Zohar O, et al. Videoagent: Long-form video understanding with large language model as agent[C]//European Conference on Computer Vision. Cham: Springer Nature Switzerland, 2024: 58-76.
> > > > > >
> > > > > > [2] Wang Z, Yu S, Stengel-Eskin E, et al. Videotree: Adaptive tree-based video representation for llm reasoning on long videos[C]//Proceedings of the Computer Vision and Pattern Recognition Conference. 2025: 3272-3283.
> > > > > >
> > > > > > [3] Lian L, Ding Y, Ge Y, et al. Describe anything: Detailed localized image and video captioning[J]. arXiv preprint arXiv:2504.16072, 2025.
> > > > > >
> > > > > > [4] Dwibedi D, Jain V, Tompson J J, et al. Flexcap: Describe anything in images in controllable detail[J]. Advances in Neural Information Processing Systems, 2024, 37: 111172-111198.
> > > > > >
> > > > > > **[Final Remark]** Thank you again for reviewing our paper and for the pleased comments. We hope these new responses and clarifications can address your questions and concerns. We sincerely invite you to engage with us if you have more questions.

---

> > > > > > > ### Author Response · Authors · 2025-11-25
> > > > > > >
> > > > > > > Dear Reviewer 9zhX,
> > > > > > >
> > > > > > > Thank you for your helpful suggestions. We have updated the paper accordingly, adding a discussion of relevant video reasoning works (including VRBench) and a statistical comparison table in the revised version. Regarding your other concerns, we have provided corresponding experimental analyses and clarifications in the rebuttal above. We hope these updates adequately address your questions, and we are more than willing to address any remaining issues to further improve the work.

---

> ### Author Response · Authors · 2025-11-27
> **Looking Forward to Your Feedback**
>
> Dear Reviewer 9zhX,
>
> Thank you once again for your constructive comments and the time you have dedicated to reviewing our paper.
> We have carefully provided further explanations and clarifications based on your suggestions. As the discussion phase has already begun, we would greatly appreciate knowing whether our responses have adequately resolved your concerns. If there are any remaining issues or if further clarification is needed, we are fully willing to address them to improve the quality of our work. We look forward to your feedback.
>
> Best regards,
>
> Authors

---

### Official Review · Reviewer_Xppq · 2025-10-30

**Soundness:** 4
**Presentation:** 4
**Contribution:** 3
**Rating:** 8
**Confidence:** 4

**Summary:**

This paper introduces a novel video reasoning benchmark, MMR-V, to evaluate multimodal deep reasoning in videos. Unlike existing video understanding benchmarks, MMR-V emphasizes multi-frame reasoning, implicit and explicit reasoning. Extensive experiments show the limitation of current MLLMs in handling complex reasoning tasks.

**Strengths:**

1. The benchmark is carefully curated with human annotation, with evidenced distractors generated by GPT.

2. The authors evaluated a wide range of models (including GPT-5, Gemini, Claude, and open-source alternatives) and provide detailed error analysis, scaling trends, and modality impact (e.g. audio).

**Weaknesses:**

1. The distinction between “implicit” and “explicit” is not always clear-cut in practice. For example, in detective films, it might require both implicit and explicit clues to determine the criminals.

2. Comparsion with existing video reasoning benchmarks (e.g. VRBench) could be further discussed.

[1] VRBench: A Benchmark for Multi-Step Reasoning in Long Narrative Videos. ICCV 25.

3. To further understand MLLMs reasoning ability, it is encouraged to include the comparison based on either different video lengths (e.g. <60s, 60s-300s, etc), or different clue lengths (e.g. 1-frame, 5-10 frames).

**Questions:**

1. In L452, it is not very clear what GPT-4.1 is labeling, can you explain in detail?

---

> ### Author Response · Authors · 2025-11-19
> **Rebuttal 1**
>
> Dear Reviewer Xppq,
>
> We sincerely appreciate your careful review and your recognition of the significance of our benchmark, the quality of our annotations, and the breadth of our experiments. We are glad to address your concerns and respond to your question.
>
> ## **Weakness 1**
>
> > The distinction between “implicit” and “explicit” is not always clear-cut in practice.
>
> We acknowledge that, in practice, the distinction between implicit and explicit can sometimes be challenging. However, we can **demonstrate the reliability of this distinction in our work from two perspectives**:
>
> -  **Theoretical foundation.** In defining the boundary between implicit and explicit reasoning, we draw upon well-established research in cognitive psychology, most notably Kahneman’s Dual Process Theory[1]. This provides a solid theoretical grounding for our categorization.
>
> - **Practical strategies.** Before annotation, we provided all annotators with **unified training and detailed guidelines**, clearly explaining the definitions of the task categories. We also offered several **pre-annotated examples**. Two examples are shown in Figure 1 of the main paper. We used the examples to help annotators understand the distinction, thereby improving consensus and annotation reliability. Furthermore, during dataset construction, we invited ***additional human inspectors** to manually review the annotations to minimize individual subjective bias*. Their checks covered both the correctness of questions and the validity of the assigned task types. Finally, to more concretely *demonstrate reliability*, we sampled 200 tasks from MMR-V *during the rebuttal stage* and asked two human evaluators to judge whether each belonged to the implicit or explicit type. The results show that ambiguous cases were **approximately 3%**.
>
> > For example, in detective films, it might require both implicit and explicit clues to determine the criminals.
>
> Regarding the example you provided, we would like to clarify the following point. For any problem that **requires implicit clues** for the correct answer, we categorize it as an **implicit reasoning** task. Conversely, an explicit reasoning task is assigned *only when the question can be solved **entirely without relying on any implicit clues***. Therefore, the situation you described is an implicit reasoning task.

---

> > ### Author Response · Authors · 2025-11-19
> > **Rebuttal 2**
> >
> > ## **Weakness 2**
> >
> > > Comparsion with existing video reasoning benchmarks (e.g. VRBench) could be further discussed.
> >
> > We appreciate you for pointing us to the excellent work VRBench. Since the works **were conducted concurrently**, the current version of our paper does not provide a discussion of VRBench, and we apologize for this oversight. We now offer a clear description of its contributions and explicitly highlight the key differences between VRBench and our MMR-V.
> >
> > VRBench is designed to evaluate a model’s reasoning capabilities over **long narrative videos**. It adopts a scalable human–model collaborative annotation pipeline and provides **stepwise** reasoning annotations. Below, we compare VRBench and our MMR-V across **four key dimensions to clarify the major distinctions** between the two benchmarks.
> >
> > **1. Differences in motivation and core idea.**
> >   - VRBench primarily focuses on **long narrative videos**, with an average duration of 1.6 hours and genres centered on narrative content (e.g., film, travel, sports). Its goal is to evaluate a model’s ability to reason about **plot lines and understand story development**, rather than to test fine-grained visual cue reasoning. A typical question would be: “Why was the woman crying?”
> >   - In contrast, the core idea of MMR-V is to evaluate a model’s ability to perform **deep reasoning over fine-grained visual clues** within videos. Our benchmark is not only about assessing a model’s understanding of global narrative structure; instead, it emphasizes the analysis and extraction of multimodal micro-level clues. As illustrated in our logo, we aim for models to **“think with video”**, just like *Sherlock Holmes examining some evidences*. Moreover, our dataset contains a large number of **non-narrative videos** (such as magic performances, videos with sophisticated editing, or artistic content). These videos are non-narrative structure but contain meticulously designed clues. This aligns with our data collection principle stated in Section 3.1.
> >   - Here we provide two comparative examples to clearly show the difference between the **deep reasoning** over video required by MMR-V and the focus on **narrative plot reasoning** in VRBench (difference in the core idea).
> >     - **VRBench: Focus on Narrative Comprehension.**
> >
> >       Example1. Question: "Why did the male protagonist arrange for his friend to rob the female protagonist?" (We couldn't find the original video link. However, this is the example shown in Figure 2 of the original VRBench[2] paper, which you can refer to to learn about the plot.) This is actually an understanding of the overall plot of the video.
> >     - **MMR-V: Focus on Deep, Multimodal Reasoning Beyond Narrative.**
> >
> >       Example 1. [Video](https://www.youtube.com/watch?v=gooWdc6kb80). We ask the model to guess what is the minimum number of video shots that had to be filmed to edit together the opening scene of the man breaking the water balloon? Ability Type: Sequential Structure Reasoning. [Reference Answer](https://www.youtube.com/watch?v=WDvjtRh90OI&t=5s)
> >
> >       Example 2. [Video](https://www.youtube.com/watch?v=D-eDNDfU3oY). We require the model to infer whether and why this video is being played forwards or in reverse. Ability Type: Counterintuitive Reasoning
> >
> >       Our appendix contains examples of all 33 subcategories of tasks, with more examples demonstrating the differences between our tasks and those for plot reasoning tasks.

---

> > > ### Author Response · Authors · 2025-11-19
> > > **Rebuttal 3**
> > >
> > > **2. Differences in annotation strategies.**
> > >   - VRBench employs a semi-automated human–AI collaborative annotation pipeline. (1) Videos are segmented using AutoShot, and **captions** are generated via VideoChat2. Whisper-large-v3 and DeepL are then used to extract and translate **audio** into English. (2) Captions and transcripts are fed into **GPT-4o to generate QA pairs and multi-step reasoning chains**. (3) **Human annotators** then refine the QA pairs based on GPT-4o outputs, videos, and subtitles. (4) False options for multiple-choice questions **are generated by DeepSeek-V3** based on the annotated answers.
> > >   - In MMR-V, **all QA pairs are carefully curated by humans**, allowing us to better control question **quality and difficulty**. (1) Unlike VRBench, we instead provide annotators with a **detailed checklist and curated annotation examples**, enabling trained annotators to generate high-quality QA pairs. During annotation, annotators are also encouraged to record potential distractor candidates. When determining QAs, annotators are encouraged to refer to official explanations by video creators and to top-voted comments, ensuring alignment with real-world user interpretation and reducing annotator bias. (2) **For distractor options**, as described in Section 3.2, we investigate **three different strategies**. The approach of VRBench corresponds roughly to our **strategy 2**. We conducted a preliminary inspection of the distractors produced by all three strategies and found that directly relying on models to generate incorrect options (strategy 2) leads to **low-quality** distractors with poor diversity. The results in Table 1 further support this observation.
> > >
> > > **3. Differences in task categories.**
> > >   - Because VRBench focuses on narrative video understanding, its task taxonomy consists of **7 categories** reflecting different aspects of narrative reasoning, and GPT-4o is prompted to generate tasks based on these descriptions.
> > >   - MMR-V, however, targets a much broader range of **multimodal reasoning abilities**. The tasks involve a wide spectrum of abilities beyond narrative comprehension. Thus, we categorize reasoning tasks into Implicit Reasoning and Explicit Reasoning, which are further divided into **10 categories and 33 subcategories**. This richer set of task categories can provide a more comprehensive analysis of the reasoning abilities required for deep multimodal video reasoning.
> > >
> > > **4. Evidence of task difficulty.**
> > >   - From a results perspective, we show that models struggle more with **fine-grained clue reasoning** than with reasoning about narrative plots. This is one of the central motivations behind MMR-V.
> > >   - For example, although the average length of VRBench videos is **1.6 hours**, GPT-4o achieves **81.23%** accuracy on MCQ tasks with only *64 frames*. In contrast, on MMR-V, with videos averaging **277 seconds**, GPT-4o achieves only **55.0%** accuracy with *300 frames*. Similarly, Phi-3.5-Vision, using 8 frames, reaches **58.03%** accuracy on VRBench, whereas Phi-4-multimodal-instruct, using 8 frames, attains only **27.6%** accuracy on MMR-V.
> > >   - These results indicate that, even with significantly shorter videos, models struggle to deeply mine and reason over fine-grained visual clues. These results can illustrate our core ideas and contributions.
> > >
> > > Finally, we will **explicitly acknowledge the valuable contributions of VRBench in the revised version of our paper.**

---

> ### Author Response · Authors · 2025-11-19
> **Rebuttal 4**
>
> ## **Weakness 3**
>
> > It is encouraged to include the comparison based on either different video lengths or different clue lengths.
>
> Thank you very much for your insightful suggestion, which indeed helps enrich our analysis of model performance. During the rebuttal stage, we computed model performance across videos of different lengths. Following prior work (e.g., Video-MME[3]) and the video-length distribution in MMR-V, we categorized videos into three groups: **short (<2 min), medium (2–6 min), and long (>6 min)**. These categories contain 329, 593, and 335 tasks, respectively.
>
> The results are shown below. We observe that model **performance generally decreases as video length increases**. Models tend to perform better on shorter videos. Interestingly, for models capable of processing a very large number of frames (e.g., Gemini-2.0-Flash with 512 frames and Gemini-2.5-Pro with 1 fps sampling), *this performance difference becomes **less pronounced***.
>
>
> | Models | Overall | Short | Medium | Long |
> |---|---|---|---|---|
> | LLaVA-Video | 18.4 | 20.5 | 16.7 | 10.4 |
> | NVILA-8B-Video | 25.5 | 27.1 | 25.9 | 22.6 |
> | Phi-4-multimodal-instruct | 26.7 | 30.1 | 24.5 | 27.2 |
> | Cogvlm2-video-llama3 | 25.6 | 27.4 | 25.9 | 22.6 |
> | Qwen2.5-VL-7B | 30.1 | 34.7 | 28.0 | 29.3 |
> | InternVL3-8B | 33.6 | 34.9 | 33.0 | 32.8 |
> | Gemma-3-12b-it | 34.0 | 35.7 | 33.5 | 31.1 |
> | InternVL2.5-38B | 39.9 | 43.9 | 39.1 | 38.2 |
> | Qwen2.5-VL-72B | 39.1 | 43.8 | 40.4 | 34.5 |
> | Gemma-3-27b-it | 42.0 | 43.8 | 42.0 | 40.3 |
> | GPT-4o-mini-2024-07-18 | 34.8 | 39.2 | 34.1 | 31.6 |
> | Gemini-2.0-Flash (16 frames) | 42.6 | 46.2 | 42.3 | 39.7 |
> | Claude-3-5-Sonnet-20241022 | 43.3 | 47.7 | 43.5 | 38.5 |
> | Gemini-2.0-Flash-thinking | 45.0 | 48.9 | 45.0 | 40.9 |
> | GPT-4.1-2025-04-14 | 46.6 | 49.9 | 46.2 | 44.2 |
> | Gemini-2.0-Flash (512 frames) | 48.8 | 48.9 | 48.6 | 49.0 |
> | Gemini-2.5-Flash | 51.2 | 54.1 | 51.9 | 47.2 |
> | o4-mini-2025-04-16 | 52.5 | 57.1 | 53.0 | 47.2 |
> | GPT-4o-2024-11-20 | 52.8 | 57.4 | 53.1 | 48.1 |
> | o3-2025-04-16 | 59.1 | 62.9 | 59.7 | 54.3 |
> | GPT-5-2025-08-07 | 60.9 | 62.3 | 61.6 | 56.2 |
> | Gemini-2.5-pro (1fps) | 64.3 | 64.7 | 64.1 | 63.9 |
>
> ## **Question 1**
>
> > In L452, it is not very clear what GPT-4.1 is labeling, can you explain in detail?
>
> Thank you for raising this question. We would like to clarify that in Line 451, we referenced **Appendix H**, which provides a comprehensive description of this analysis, including the detailed workflow, the specific annotation prompts used for GPT-4.1, and concrete labeling examples.
>
> We are certainly happy to provide a detailed explanation here. Our process was as follows. Firstly, we uniformly divided the Chain-of-Thought (CoT) sequences on the MMR-V dataset into N segments to analyze the shift in the model's reasoning focus across different stages. We categorized the information processed by the model into two main types: **text analysis** (targeting the question and options) and **video analysis** (targeting video content). **Video analysis** was further distinguished between reasoning based on the **question frame** and **other frames**. GPT-4.1 was tasked with labeling each CoT segment to identify **which of these specific information categories** the model was reasoning about.
>
> [1] Kahneman D. Thinking, fast and slow[M]. macmillan, 2011.
>
> [2] Yu J, Wu Y, Chu M, et al. VRBench: A Benchmark for Multi-Step Reasoning in Long Narrative Videos[J]. arXiv preprint arXiv:2506.10857, 2025.
>
> [3] Fu C, Dai Y, Luo Y, et al. Video-mme: The first-ever comprehensive evaluation benchmark of multi-modal llms in video analysis[C]//Proceedings of the Computer Vision and Pattern Recognition Conference. 2025: 24108-24118.
>
> **[Final Remark]** Thank you again for reviewing our paper and giving such meaningful suggestions. We hope these new additions address your concerns. We sincerely invite you to engage with us if you have more questions. We hope our work can contribute to the advancement of MLLMs and help steer them toward solving challenges that are more closely aligned with real-world demands.

---

> > ### Author Response · Authors · 2025-11-25
> >
> > Dear Reviewer Xppq,
> >
> > Thank you for your careful review and constructive suggestions. In response to your suggestion, we have **updated the paper** to include a discussion on VRBench and added an analysis of model performance across different video lengths in the revised version.

---

### Author Response · Authors · 2025-11-30
**Rebuttal Summary for Area Chair (Overview & Strengths)**

Dear AC,

Due to unexpected circumstances, we understand that you are now required to oversee our paper, reviews, and rebuttal content to draft the meta-review. We sincerely appreciate your contribution to the academic community and deeply respect the additional workload and effort this situation entails.

To assist you in efficiently grasping the key points of the reviews and our rebuttal, we have summarized the discussion process below. We have **included all the weaknesses and questions** here, but have **simplified our clarifications**. For more detailed explanations, we have **marked the corresponding locations** of our full response should you wish to consult them. Our summary is around three parts: **Strengths, Common Concerns, and Remaining Concerns**. Please feel free to refer to the specific parts at your convenience.

# **Summary of Strengths**
Firstly, we would like to express our gratitude to the reviewers for their meticulous reviews and insightful comments. We also appreciate their recognition of the strengths of our work:

**1. The Core Idea is Novel and Significant**. MMR-V proposes a novel long-range, multi-frame video reasoning task, which addresses a **significant and practical** problem in video research and better reflects real-world reasoning challenges (Reviewers Xppq, 9zhX, NiAg).

**2. High Quality and Reliability of the Benchmark**. All QA pairs were meticulously annotated by humans, ensuring high quality and reliability. Furthermore, MMR-V designed a novel annotation pipeline that considers real-world user consensus and incorporates a novel error answer annotation strategy (Reviewers Xppq, qv7x, NiAg).

**3. Comprehensive Experiments**. Extensive experiments were conducted across a wide range of models. The paper also performed detailed ablation studies and analyses, including error analysis, scaling trends, and modality impact (Reviewers Xppq, qv7x, NiAg).

**4. Novel and Interesting Task Taxonomy**. MMR-V is the first to divide reasoning tasks into implicit and explicit categories, which is highlighted as an **interesting and novel step** by reviewer, marking as a key contribution (Reviewers Xppq, qv7x, NiAg).

---

> ### Author Response · Authors · 2025-11-30
> **Rebuttal Summary for Area Chair (Common Concerns Part 1)**
>
> # **Summary of Common Concerns**
> Next, we outline the key points raised by reviewers, particularly those that can be combined, along with our responses.
>
> ### **1. Comparison with Related Works (Reviewers Xppq, 9zhX and qv7x)**.
>
> > (W2 of Xppq, W1 of 9zhX and W3 of qv7x) Lack of discussion regarding the recently released VRBench and requests for comparison.
>
> First, we would like to clarify that the initial version of the paper did not discuss VRBench because it is **concurrent work**. We apologize for this oversight.
>
> VRBench is an outstanding work that focuses on narrative reasoning in long narrative videos. It adopts a scalable human–AI collaborative annotation pipeline and provides stepwise reasoning annotations.
> Below, we summarize the four main differences between our MMR-V and VRBench:
> - **Motivation & Core Idea**. VRBench focuses on **narrative reasoning** (e.g., story development) in long videos (avg. 1.6h). Our MMR-V targets deep reasoning over **fine-grained visual clues**. We aim to test if models can "think with video" (like a detective analyzing evidence) rather than just the global story. We have provided examples in the rebuttal to intuitively demonstrate the differences between the two tasks.
> - **Annotation Strategy**. VRBench relies on a semi-automated, **human-AI collaborative pipeline** for QA annotation, whereas MMR-V employs a **fully human-curated pipeline**. Regarding distractor annotation, VRBench directly uses Deepseek-V3 to generate distractors based on the correct answer. In contrast, we designed several distractor annotation strategies and performed a comparative analysis. We found that distractors generated directly by models based on the answer often lack sufficient quality and diversity (Section 3.2).
> - **Task Categories**. VRBench utilizes **7 categories** centered on narrative reasoning abilities. Our MMR-V introduces a broader taxonomy, categorizing reasoning into **Implicit and Explicit types**, further divided into **10 categories and 33 subcategories**. Examples for the 33 subcategories are provided in the Appendix E. This allows for a more comprehensive assessment of diverse reasoning abilities.
> - **Difficulty**. Results show that models struggle significantly more with our fine-grained tasks than with narrative plots. For instance, GPT-4o achieves **81%** accuracy with **64 input frames** on VRBench (avg. **1.6h**), but only **55%** with **300 frames** on our MMR-V (avg. **277s**). Similarly, Phi-3.5-Vision reaches **58.03%** accuracy with 8 frames on VRBench, whereas Phi-4-multimodal-instruct attains only **27.6%** accuracy with 8 frames on MMR-V. This quantitative gap demonstrates that mining fine-grained visual clues is a harder challenge for current MLLMs.
>
> Finally, we have acknowledged and appreciated the outstanding contribution of VRBench in the revised paper. For details, please refer to our Rebuttal 2 and Rebuttal 3 to Reviewer Xppq.
>
> > (W1 of 9zhX) Suggestion to add a table comparing statistics with previous video benchmarks.
>
> We compiled this table during the rebuttal phase and have updated it to **Table 5 in the Appendix** of the revised paper.
>
> > (W3 of qv7x) Suggestion to discuss comparison with recent Video Reasoning Benchmarks and add them to the literature review.
>
> We discussed these works and highlighted key differences with our MMR-V in Rebuttal 3 to Reviewer qv7x. We have also incorporated these discussions into **Section 5 RELATED WORK** of the revised paper.

---

> > ### Author Response · Authors · 2025-11-30
> > **Rebuttal Summary for Area Chair (Common Concerns Part 2)**
> >
> > ### **2. Annotation Details for Explicit and Implicit Categories (Reviewers Xppq, qv7x)**.
> > > (W1 of Xppq) The distinction between “implicit” and “explicit” is not always clear-cut in practice. For example, in detective films, it might require both implicit and explicit clues to determine the criminals.
> >
> > We acknowledge that the distinction between implicit and explicit can sometimes be challenging. However, we ensured reliability through:
> > - **Theoretical Foundation**: Our definitions are grounded in established cognitive psychology (specifically Kahneman’s Dual Process Theory), which provides a robust theoretical basis.
> > - **Standardization**: We implemented rigorous annotator training with unified guidelines and examples for explicit and implicit reasoning tasks. We also conducted manual cross-checks to minimize subjective bias.
> > - **Verification: During rebuttal phase**, we verify 200 sampled tasks by independent human evaluators. The results showed that the number of ambiguous samples labeled implicit and explicit was around **3%**, demonstrating high consistency.
> > - Regarding tasks that involve both types of clues (e.g., detective films), we apply a clear rule. **Implicit:** Any task requiring **at least one implicit clue** is classified as Implicit. **Explicit:** Tasks solvable **entirely via explicit evidence**. Therefore, the task mentioned falls under the Implicit category.
> >
> > > (Q3 of qv7x ) For implicit reasoning, how to ensure that all annotators can obtain all world knowledge to understand metaphors?
> >
> > As detailed in **Section 3 and Appendices C.1 CHECKLIST**, we ensure annotators access the necessary **world knowledge** by strictly requiring them to reference official creator explanations and top-voted community comments of the original video pages. This strategy also aligns our ground truth with widely accepted **real-world consensus**, mitigating individual subjective bias.
> >
> > ### **3. Ablation Studies and Analysis Details (Reviewers Xppq, qv7x, NiAg)**
> >
> > >(Q1 of Xppq) In L452, it is not very clear what GPT-4.1 is labeling, can you explain in detail?
> >
> > Our objective in this section was to analyze which type of information the model is reasoning about in each segment of the CoT. This classification distinguishes between reasoning based on **text** (e.g., the question and options) and on **video content**. The video analysis is further categorized into reasoning regarding **Question Frames** (frames explicitly mentioned in the question text) versus **Other Frames**. GPT-4.1 was employed to assign these specific labels to each CoT segment. We would like to clarify that the detailed setup for this analysis experiment is presented in **Appendix H**.
> >
> > > (W2 of qv7x) The description of the human annotation process (of error analysis) is unclear.
> >
> > We ensured the reliability and consistency of our error analysis through two key aspects:
> >
> > - **Annotation Pipeline**: We established **clear definitions** for the six error categories (Section 4.5) using **representative examples** (some in Appendix J) and **validated** them with non-authors to ensure consensus prior to annotation.
> > - **Verification: During the rebuttal**, an independent evaluator checked the annotations and achieved a **95% agreement rate** (only 5% were ambiguous), quantitatively confirming the reliability of our analysis.
> >
> > > (Q3 of NiAg) Were these tasks (human experiment) balanced across implicit/explicit reasoning and video categories?
> >
> > We conducted a statistical analysis of the task distribution for the human experiment during the rebuttal phase. The results demonstrate that the sampled tasks are well-balanced across both task and video categories (Rebuttal 4 to Reviewer NiAg).

---

> > > ### Author Response · Authors · 2025-11-30
> > > **Rebuttal Summary for Area Chair (Common Concerns Part 3)**
> > >
> > > ### **4. Questions Regarding Task Annotation Details (Reviewer qv7x)**.
> > >
> > > > (Q1 of qv7x) Why do the authors only select the multiple-choice question? Are open-ended questions essential for reasoning tasks?
> > >
> > > We selected the MCQ format to ensure a **scientific and reliable** evaluation, based on three key factors:
> > > - Unlike math tasks with unique numeric answers, open-ended reasoning of our implicit tasks regarding abstract metaphors is **difficult to standardize**. Current **"LLM-as-a-judge" methods lack the necessary reliability** to accurately score, **especially implicit reasoning**, against ground truth.
> > > - This format **aligns with leading video benchmarks** (e.g., Video-MME, LongVideoBench, MLVU).
> > > - Our pilot study revealed that **GPT-4o achieved only 17%** accuracy in an open-ended setting (Section 3.2). Models often failed to provide on-point answers, resulting in a compressed performance distribution that makes it difficult to effectively differentiate model capabilities.
> > >
> > > > (Q2 of qv7x) The checklist requires long distance, how long is it, and how to guarantee it?
> > >
> > > We demonstrated the **long-range, multi-frame nature** of the dataset through three aspects:
> > > - Annotators must **watch the full video** before annotation to ensure a global understanding of video (Appendix C.1).
> > > - While we **initially set strict quantitative thresholds** (e.g., clue frames must cover at least 20% of duration or exceed 1 minute; >5 frames). However, we found that some tasks, while not meeting all of the criteria, still align with the core characteristics of our "Deep Reasoning" (example was in the original Rebuttal). For the sake of rigor, we removed these quantitative indicators and replaced them with qualitative descriptions in paper.
> > > - **Post-annotation analysis** confirms this characteristic: on average, tasks require reasoning over **12 frames** spanning **approximately 60%** of the video duration (**L113 of Introduction**).
> > >
> > > ### **5. Suggestions for Additional Main Experiments (Reviewers Xppq, 9zhX)**
> > > >(W3 of Xppq) It is encouraged to include the comparison based on either different video lengths (e.g. <60s, 60s-300s, etc), or different clue lengths (e.g. 1-frame, 5-10 frames).
> > >
> > > We calculated the performance of all models listed in Table 3 across different video length distributions (Short, Medium, Long). These results, along with our interesting findings, have been **added to Appendix G.1 and Table 7** of the revised paper.
> > >
> > > > (W2 of 9zhX) There exists some agentic mechanism for long video reasoning such as VideoAgent, VideoTree, etc. These methods are suitable for reasoning via tool usage with multi-round clue discovery. Hence, it would be great to include these works for evaluating the benchmark.
> > >
> > > During the rebuttal phase, we tested VideoAgent on our MMR-V. We acknowledge that these works are excellent methods for long video understanding. However, they **are not suitable with MMR-V**. The performance **improvement was not significant** (particularly on Short and Medium-length videos). Based on the examples and analysis in the original VideoAgent paper, as well as our experimental analysis, VideoAgent aims to address the issue of *limited context windows* preventing the ingestion of the majority of frames in long videos. Its multi-round clue discovery primarily focuses on *retrieving frames that are **mentioned in the question** but were not initially provided to the model*.
> > >
> > > In MMR-V, these correspond to what we define as **"Question Frames".** However, our objective is to evaluate whether models can locate **other clue frames** that may be *far from these question frames*, thereby achieving "Deep Reasoning" over the video like a detective. For a more detailed explanation, please refer to Rebuttal 3 and Rebuttal 4 to Reviewer 9zhX.

---

> > > > ### Author Response · Authors · 2025-11-30
> > > > **Rebuttal Summary for Area Chair (Remaining Concerns)**
> > > >
> > > > # **Summary of Remaining Concerns**
> > > >
> > > >
> > > > Finally, we provide a brief overview of the remaining, less overlapping concerns below.
> > > >
> > > > > (W1 of qv7x) Tasks might not require deep reasoning, unlike math or science problems which need long CoT. Some tasks appear to rely only on perception (e.g., personal reflection, video naming, and meta-emotion).
> > > >
> > > > - MMR-V aims to evaluate deep reasoning over multimodal information, similar to how OpenAI’s o3 model "thinks with images." While this differs from textual reasoning (like math), it is equally important and holds significant value for real-world model deployment.
> > > > - Current models demonstrate **superhuman** performance on math. However, on MMR-V, the best model still trails human by **21.7%**. This gap highlights a meaningful research challenge.
> > > > - The tasks mentioned are Implicit, which requires abstraction understanding beyond perception. We suspect the examples in **Appendix E caused a misunderstanding**, as the original video we show the link may contain "perception shortcuts" (e.g., a title screen with the video name) which are removed in the final video of our benchmark. We guarantee that all videos in MMR-V **have been strictly processed and reviewed to ensure no such perception shortcuts exist**.
> > > >
> > > > > (W2 of 9zhX) It would be more interesting to investigate how to design MLLMs to tackle the tasks in this benchmark. (Refer to our Rebuttal 5).
> > > >
> > > > Due to space constraints, we only briefly discussed in the CONCLUSION. However, Section 4.5 analyzes the characteristics of better-performing models, providing insights into how to address MMR-V. We provided a detailed analysis of potential solutions in the Rebuttal and are happy to incorporate this into the revised paper if needed.
> > > >
> > > > > (W3 of 9zhX) Dataset size (317 original videos) is small.
> > > > - First, videos suitable for annotating deep reasoning require high quality and are rare on the web. According to the checklist in Section 3.1, we filtered out lots of videos. Unlike general understanding, reasoning benchmarks are difficult to scale.
> > > > - Second, our video and task **diversity** is sufficient and representative (Appendix D).
> > > > - Furthermore, the time cost for evaluating reasoning models is high, which is why established reasoning benchmarks typically have smaller sample sizes (e.g., AIME, GPQA, Video-MMMU).
> > > > - Finally, manual annotations are expensive, limiting immediate scaling. But we plan to update and expand in future versions.
> > > >
> > > > >(Q4 of qv7x) Does the cultural background of different people lead to different reasoning results?
> > > >
> > > > If this refers to the human performance experiment, it might have an impact; however, since the Cultural task subcategory comprises only 4.6% of the dataset, the overall impact is minimal.
> > > >
> > > > > (W1 of NiAg) Some video types are not covered extensively, and the language is predominantly English.
> > > >
> > > > This is the future direction we mentioned in the **Limitations section**. First it is impossible to exhaust every real-world video type. We selected **24 sub-categories** to ensure diversity. We explained the reasons for omitting specific types in the Rebuttal. Second, as this is a reasoning benchmark, the dominance of English does not compromise the quality of the evaluation. **Most existing reasoning benchmarks are exclusively in English**.
> > > >
> > > > > (W2 of NiAg) The paper explains why implicit reasoning performance is higher than explicit, but does not explain why explicit is harder.
> > > >
> > > > In the paragraph starting at L353 in Section 4.2, we actually provide a **comparative analysis** of the performance difference. We point out that implicit tasks yield higher performance because the clue frames **are more numerous. **In contrast**, explicit tasks are harder because they contain fewer clue frames, requiring the model to perform more precise reasoning.
> > > >
> > > > > (W3 of NiAg) No efficiency metrics (e.g., Inference time and Memory Usage) were provided when testing Gemini with increasing frame counts.
> > > >
> > > > We rely on the API to test Gemini. To the best of our knowledge, the API does not allow for accurate measurement of these efficiency metrics. Some influencing factors (e.g. network latency and upload speeds) introduce significant variance into inference time measurements, and memory usage is unobtainable when using the API. Furthermore, previous video benchmarks typically do not report these metrics when evaluating Gemini.
> > > >
> > > > > (Q2 of NiAg) Does the volume of world knowledge affect implicit reasoning?
> > > >
> > > > MMR-V primarily relies on common sense knowledge (e.g., COVID-19), which most models possess. We conducted a RAG experiment during the rebuttal to analyze the impact of external knowledge. The results showed that the improvement brought by RAG was not significant.
> > > >
> > > > **[Final Remark]** Once again, we would like to convey our deep respect and gratitude for your significant contribution and the time dedicated to this process. We also extend our sincere thanks to the reviewers for their meticulous comments and valuable suggestions.

---

### Meta-Review · Area_Chair_TTMe · 2026-01-08

**Summary:**

The paper introduces MMR-V, a benchmark for multimodal deep reasoning in videos that emphasizes long-range and multi-frame reasoning. Reviewers broadly agree that the benchmark is novel, well-motivated, and carefully annotated, and that it exposes clear limitations of current MLLMs. Strengths include the focus on fine-grained visual clue reasoning, human-curated annotations, diverse task taxonomy, and extensive experiments across many models. Main concerns centered on comparisons with concurrent benchmarks (e.g., VRBench), clarity of task definitions, dataset size (317 videos), and how models might be designed to solve the benchmark, rather than flaws in correctness or evaluation.

**Reviewer Concerns:**

Concerns addressed by the rebuttal:
1/ Clarified distinctions between MMR-V and VRBench, including motivation, task design, annotation strategy, and difficulty.
2/ Added statistical comparison tables with existing video benchmarks.
3/ Provided analyses across video lengths and task types.
4/ Clarified implicit vs. explicit reasoning definitions and demonstrated annotation consistency.
5/ Evaluated agentic methods (VideoAgent) and explained their limited gains on MMR-V.

Remaining limitations:
1/ Dataset size remains relatively small due to manual annotation cost.
2/ Benchmark focuses on evaluation rather than proposing new ways in which the limitations could be resolved. This could be a potential future work though.
3/ Efficiency metrics for closed-source models cannot be reported for a fair comparison.

**Reviewer Scores:**

Based on discussion, AC believes most reviewers would maintain their original scores or slightly improve.

---

### Decision · Program_Chairs · 2026-01-26

Accept (Poster)